# Particle Stochastic Dual Coordinate Ascent: Exponential convergent algorithm for mean field neural network optimization

**Kazusato Oko**[1,†], **Taiji Suzuki**[1,2,‡], **Atsushi Nitanda**[3,4,2,*], **Denny Wu**[5,6,⋆]

[1]University of Tokyo,  [2]RIKEN AIP,  [3]Kyushu Institute of Technology,  [4]JST PRESTO,
[5]University of Toronto,  [6]Vector Institute

[†]`oko-kazusato@g.ecc.u-tokyo.ac.jp`, [‡]`taiji@mist.i.u-tokyo.ac.jp`,
[*]`nitanda@ai.kyutech.ac.jp`, [⋆]`dennywu@cs.toronto.edu`

## Abstract

We introduce Particle-SDCA, a gradient-based optimization algorithm for two-layer neural networks in the mean field regime that achieves exponential convergence rate in regularized empirical risk minimization. The proposed algorithm can be regarded as an infinite dimensional extension of Stochastic Dual Coordinate Ascent (SDCA) in the probability space: we exploit the convexity of the dual problem, for which the coordinate-wise proximal gradient method can be applied. Our proposed method inherits advantages of the original SDCA, including (i) exponential convergence (with respect to the outer iteration steps), and (ii) better dependency on the sample size and condition number than the full-batch gradient method. One technical challenge in implementing the SDCA update is the intractable integral over the entire parameter space at every step. To overcome this limitation, we propose a tractable *particle method* that approximately solves the dual problem, and an importance re-weighting technique to reduce the computational cost. The convergence rate of our method is verified by numerical experiments.

## 1 Introduction

A major challenge in developing an optimization theory for neural network is to handle the non-convexity of loss landscape. Recent works showed that *overparameterization* is one key property that makes global optimization possible. In particular, the effectiveness of overparameterization has been extensively investigated in the theory of *Neural Tangent Kernel* (NTK) and the *mean field analysis*.

The NTK analysis considers a linear approximation of neural networks and casts the gradient descent dynamics to that in a corresponding reproducing kernel Hilbert space (Jacot et al., 2018). Since the whole argument can be carried out essentially in a linear space, we can derive both global convergence and generalization guarantees (Du et al., 2019; Allen-Zhu et al., 2019; Zou et al., 2018; Nitanda & Suzuki, 2020). However, one of the biggest drawbacks of the NTK approach is that the fixed kernel fails to capture the feature training aspect of deep learning. Indeed, this feature learning ability of neural network is an important ingredient that separates deep learning from shallow models such as kernel methods (Suzuki, 2019; Yehudai & Shamir, 2019; Ghorbani et al., 2019).

In contrast, the mean field analysis considers a smaller scale of parameters than the NTK analysis, which allows the parameters to travel away from the initialization. However, the optimization problem is no longer convex and the analysis requires more involved mathematical tools. A typical approach is to formulate the optimization of neural networks as a partial differential equation (PDE) of the distribution of parameters (Nitanda & Suzuki, 2017; Mei et al., 2018; Chizat & Bach, 2018; Rotskoff & Vanden-Eijnden, 2018; Sirignano & Spiliopoulos, 2020). A benefit of the PDE formulation is that the objective function becomes convex in the space of measures. This being said, since the dynamics is essentially infinite dimensional, most convergence results are shown in the continuous limit (infinite-width limit). Also, many existing guarantees only handle the continuous time dynamics, due to difficulty in analyzing the time discretization of the PDE. Indeed, Nitanda & Suzuki (2017); Chizat & Bach (2018); Rotskoff & Vanden-Eijnden (2018) proved convergence to a global optimal solution for essentially infinite width networks under continuous time settings. On the other hand, global optimality may be shown under less restrictive settings if we consider the *noisy* gradient

descent algorithm; the corresponding stochastic dynamics can be formulated as McKean–Vlasov dynamics, and its convergence has been studied in Mei et al. (2018); Hu et al. (2019). However, most of these works establish convergence result for the discrete-time finite-width method by bounding the difference from its continuous-time and infinite-width limit counterpart, which usually incurs a large discretization error. Hence, it is important to develop a practical algorithm with optimization guarantees for finite width network and discrete time setting.

Recently, Nitanda et al. (2021) proposed a completely time-space discretized algorithm termed Particle Dual Averaging (PDA) that attains *polynomial time* convergence guarantee to the (regularized) global optimal solution. This method combines the dual averaging technique (Nesterov, 2005; 2009; Xiao, 2009) and the gradient Langevin dynamics (Vempala & Wibisono, 2019). More specifically, the algorithm realized time discretization by extending the dual averaging technique for finite dimensional convex optimization to the infinite dimensional setting, and realized the space discretization by sampling finite number of particles via the gradient Langevin dynamics. Eventually, the method achieves $O(1/t)$ convergence with respect to the outer-loop iteration $t$. Although $O(1/t)$ is optimal in an online optimization setting, there is still room for improvement for a finite sample size setting. Motivated by this observation, we propose a method that improves the convergence rate from the polynomial order ($O(1/t)$) to an *exponential* order ($O(\exp(-Ct))$).

**Our contribution.** We propose Particle-SDCA, a novel optimization algorithm for two-layer neural networks in the mean field regime that achieves *linear convergence (w.r.t. the outer loop steps) for regularized empirical risk minimization*[1]. We adopt the stochastic dual coordinate ascent (SDCA) technique (Shalev-Shwartz & Zhang, 2013a; Takáč et al., 2013; Shalev-Shwartz & Zhang, 2013b) to achieve linear convergence rate and much better dependency on the sample size $n$. Importantly, the integration with respect to the probability measure on the parameters is approximated by a finite number of particles. In contrast to existing methods, in which particles are sampled at each iteration, we employ a novel update rule of the weights on the particles so that the sampling procedure is only performed once in $\tilde{n}$-iterations, which provides significant computational speedup.

Furthermore, unlike many existing analyses concerning the time-space discretized version of McKean–Vlasov dynamics (Mei et al., 2018; Bou-Rabee & Schuh, 2020; Bou-Rabee & Eberle, 2021), we do not couple the discrete time update with the continuous time counterpart by taking the small step size limit. Instead, we directly analyze the convergence of the discrete time update; hence, our method converges to the optimal solution with a *fixed* (non-vanishing) step size. Our contribution can be summarized as follows (for comparison with existing works see Table 1):

- We propose a new optimization method for mean field representation of neural networks that is efficient in the finite sample setting. By utilizing the SDCA technique, the algorithm achieves exponential convergence with respect to the outer iteration. Moreover, the global convergence is guaranteed without letting the step size converge to 0.

- The integral required in each update is approximated by an average over finite number of particles. The number of particles can be of linear order with respect to a required precision, and the computational cost for re-sampling can be of poly-log order.

- Thanks to a novel re-weighting scheme for particles, we only resample particles once in $\tilde{n}$ updates, where $\tilde{n}$ can be as large as the total sample size $n$; this significantly reduces the computation cost.

| Method (authors) | Outer-iteration | Inner-iteration | Total complexity |
|---|---|---|---|
| PDE (Bou-Rabee & Schuh, 2020) | $\epsilon_P^{-1} \log(1/\epsilon_P)$ | $M_1^{*2}$ | $M_1^{*2}\epsilon_P^{-1} \log(1/\epsilon_P)$ |
| PDA (Nitanda et al., 2021) | $\epsilon_P^{-1}$ | $M_2^* G^*$ | $M_2^* G^* \epsilon_P^{-1}$ |
| P-SDCA (ours) | $\frac{1}{\tilde{n}}(n + \frac{1}{\gamma\lambda_2})\log(\frac{n}{\epsilon_P})$ | $M_3^*(\tilde{n}+G^*)$ | $M_3^*(1+\frac{G^*}{\tilde{n}})(n+\frac{1}{\gamma\lambda_2})\log(\frac{n}{\epsilon_P})$ |

Table 1: Required computational complexity to achieve $\epsilon_P$-accuracy ($P(\hat{p}) - P(p^*) \le \epsilon_P$). $\tilde{n}$ is the re-sampling interval. $M_i^*$ ($i = 1, 2, 3$) is the number of particles required to approximate the true distribution for each method: They are given as $M_1^* = \Theta(\epsilon_P^{-2})$, $M_2^* = \Theta(\epsilon_P^{-2} \log(n))$, $M_3^* = \Theta(\epsilon_P^{-1} \log(n))$. $G^*$ is the number of gradient evaluation to obtain the sampling distribution which has TV-distance $\epsilon_P$ from the true distribution; when MALA is used, this is given as $G^* = O(n \log(1/\epsilon_P)^{3/2})$.

---

[1]We remark that our goal is to propose a more efficient algorithm for optimizing mean field neural network with faster convergence guarantee, rather than to gain a better understanding on standard neural network training.

## 2 PRELIMINARIES

We first introduce the problem setting and some notations used in the following sections. We consider the empirical risk minimization problem in the supervised learning setting with a two-layer neural networks, where the input and output spaces are denoted by $\mathcal{X} \subset \mathbb{R}^d$ and $\mathcal{Y} \subset \mathbb{R}$, respectively. Let $h_\theta : \mathcal{X} \to \mathcal{Y}$ be one neuron with the parameters $\theta \in \mathbb{R}^{\tilde{d}}$. For example, we may set $h_\theta(x) = \tanh(r)\sigma(w^\top x)$ for $\theta = (r, w) \in \mathbb{R} \times \mathbb{R}^d$ and an activation function $\sigma$ (the $\tanh$ operation is merely to ensure boundedness of the output). In the mean field regime, we consider neural networks represented as an average of neurons: $f_\Theta(x) = \frac{1}{M} \sum_{m=1}^{M} h_{\theta_m}(x)$ where $\Theta = (\theta_m)_{m=1}^{M} \subset \mathbb{R}^{\tilde{d}}$ is a set of parameters. The continuous limit in the mean field regime is obtained by taking $M \to \infty$, and by an analogy to the law of large numbers, the function $f_\Theta$ converges to the following *integral representation* of a two-layer neural network:

$$f_p(x) = \int h_\theta(x)p(\theta)\mathrm{d}\theta,$$

where $p : \mathbb{R}^{\tilde{d}} \to \mathbb{R}$ is a probability density function on $\mathbb{R}^{\tilde{d}}$ representing the weight of parameters. Here, we denote by $\mathcal{P}$ the set of probability density functions on $\mathbb{R}^{\tilde{d}}$. As in the typical mean field analysis (Nitanda & Suzuki, 2017; Mei et al., 2018; Chizat & Bach, 2018), we aim to optimize the density function $p \in \mathcal{P}$ so that the neural network $f_p$ accurately predicts the output $y \in \mathcal{Y}$ from the input $x \in \mathcal{X}$. To define the empirical risk and predictive risk, we let $\ell(z, y) : \mathcal{Y} \times \mathcal{Y} \to \mathbb{R}$ be a convex loss function, such as the squared loss $\ell(z, y) = (z - y)^2/2$ for regression, and the logistic loss $\ell(z, y) = \log(1 + \exp(-yz))$ for classification. For each $(x_i, y_i)$, we use the notation $h_i(\theta)$ and $\ell_i(f(x_i))$ to indicate $h_\theta(x_i)$ and $\ell(f(x_i), y_i)$ respectively. To estimate the density function $p \in \mathcal{P}$, we minimize the regularized empirical risk defined by

$$\min_{p \in \mathcal{P}} \frac{1}{n} \sum_{i=1}^{n} \ell_i(f_p(x_i)) + \lambda \mathrm{KL}(p||N(0, \sigma^2 I)) \tag{1}$$

where $\lambda > 0$ is a regularization parameter and $\mathrm{KL}(p, N(0, \sigma^2 I))$ represents the KL divergence from the normal distribution $N(0, \sigma^2 I)$ with mean 0 and covariance $\sigma^2 I$ to the distribution with the density function $p$. This type of regularization naturally arises in training neural networks by the gradient Langevin dynamics (Hu et al., 2019; Chen et al., 2020b). In our framework, we consider directly optimizing the empirical risk with KL regularization, instead of running gradient Langevin dynamics (i.e., noisy gradient descent) on the unregularized objective. By decomposing the KL divergence in Eq. (1), we obtain the following equivalent representation of the objective:

$$\min_{p \in \mathcal{P}} P(p) := \frac{1}{n} \sum_{i=1}^{n} \ell_i(f_p(x_i)) + \lambda_1 \int p(\theta)\|\theta\|^2 \mathrm{d}\theta + \lambda_2 \int p(\theta) \log(p(\theta))\mathrm{d}\theta, \tag{2}$$

where $\lambda_1, \lambda_2 > 0$ are the regularization parameters. Importantly, the objective is convex with respect to the density $p \in \mathcal{P}$ although it is non-convex with respect to each parameter $\theta$. We make full use of this convexity in the following analysis.

The main difficulty to optimize (2) stems from the following two factors: (i) there is an integral with respect to $p$ that should be approximated by a computationally tractable scheme, and (ii) there is a summation over $n$ data points, which could lead to costly gradient evaluations. To overcome the former difficulty, we sample a finite number of particles from the density $p$. As for the later, we adopt the stochastic dual coordinate ascent (SDCA) technique, which we explain in the next section.

## 3 PROPOSED METHOD: PARTICLE-SDCA

We now explain our proposed *Particle-Stochastic Dual Coordinate Ascent (P-SDCA)* method.

### 3.1 DUAL PROBLEM AND ALGORITHM DESCRIPTION

The SDCA method relies on the dual problem. The dual of the optimization problem (2) can be derived from the following *Fenchel's duality theorem*. Note that the convexity of $P(p)$ with respect to the density function $p$ is important in the derivation.

**Proposition 1** (Fenchel's duality theorem). *Suppose that $\ell_i : \mathbb{R} \to \mathbb{R}$ is a proper convex function, $h_i : \mathbb{R}^d \to \mathbb{R}$ is bounded, and the primal problem satisfies $\inf_{p \in \mathcal{P}} P(p) > -\infty$, then the following duality holds:*

$$\inf_{p \in \mathcal{P}} P(p) = \sup_{g \in \mathbb{R}^n} D(g) := \sup_{g \in \mathbb{R}^n} \left\{ -\frac{1}{n} \sum_{i=1}^n \ell_i^*(g_i) - \lambda_2 \log \left( \int q[g](\theta) \mathrm{d}\theta \right) \right\}$$

*where $q[g](\theta) := \exp\left\{ -\frac{1}{\lambda_2}\left( \frac{1}{n}\sum_{i=1}^n h_i(\theta)g_i + \lambda_1 \|\theta\|^2 \right) \right\}$ and $\ell_i^*$ is the convex conjugate[2] of $\ell_i$. Moreover, $p^* \in \mathcal{P}$ and $g^* \in \mathbb{R}^n$ are both optimal solutions of primal and dual problems if and only if $f_{p^*}(x_i) \in \partial \ell_i^*(g_i^*)$ and $p[g^*] = p^*$ , where $p[g](\theta) := \frac{q[g](\theta)}{\int q[g](\theta')\mathrm{d}\theta'}$ and $\partial\ell_i^*$ denotes the sub-differential of $\ell_i^*$.*

This proposition can be derived from Theorems 3 and 8 of Rockafellar (1967) (its finite dimensional version can be found in Corollary 31.2.1 of Rockafellar (1970)). See Appendix A for the complete proof. By the optimality condition, we can recover the optimal primal solution $p^*$ from the optimal dual solution $g^*$. Hence, we aim to optimize the dual problem as $p^* = p[g^*]$. In the dual problem, updating one coordinate $g_i$ can be carried out by picking up one data points $(x_i, y_i)$. The main strategy of SDCA is to randomly pick the sample index $i \in [n]$ [3] and update the corresponding coordinate $g_i$ by maximizing the dual (i.e., dual coordinate ascent). Before we state the algorithm, we assume the following condition.

**Assumption 1.**

(A1) $\ell_i : \mathbb{R} \to \mathbb{R}$ is $(1/\gamma)$-smooth[4] for all $i \in [n]$.

(A2) $\sup_{\theta \in \mathbb{R}^d} |h_i(\theta)| \leq 1$ for all $i \in [n]$.

The first assumption (A1) is satisfied, for example, by the squared loss $\ell_i(u) = (y_i - u)^2/2$ with $\gamma = 1$ and the logistic loss $\ell_i(u) = \log(1 + \exp(-yu))$ with $\gamma = 1$. As for the second assumption (A2), our analysis is valid for bounded neurons, and the upper-bound 1 is chosen just for simplicity and can be replaced by another value. In fact, if we assume $\sup_{\theta \in \mathbb{R}^d} |h_i(\theta)| \leq R$ instead, the analysis can be reduced to the setting of $R = 1$ by rescaling $h_i$ and the loss function, i.e., $R$ can be absorbed into the smoothness parameter, which then becomes $R/\gamma$.

As stated above, the strategy to optimize the dual problem is to (i) randomly pick $i \in [n]$ and (ii) update $g_i$ by approximately maximizing $D(g)$ with respect to $g_i$ with other coordinates fixed. For the update of each coordinate, we employ the *proximal gradient*-type update where we optimize the lower bound of $D(g)$ given by the following lemma (the proof can be found in Appendix C).

**Lemma 1.** *Under (A2) of Assumption 1, for $g \in \mathbb{R}^n$ and $\delta g \in \mathbb{R}^n$, the second term of $D(g)$ can be bounded as*

$$\lambda_2 \log\left(\int q[g + \delta g](\theta)\mathrm{d}\theta\right) \leq -\frac{1}{n}\sum_{i=1}^n \int h_i(\theta)p[g](\theta)\mathrm{d}\theta\delta g_i + \frac{1}{2n^2\lambda_2}\|\delta g\|_1^2 + \lambda_2 \log\left(\int q[g](\theta)\mathrm{d}\theta\right).$$

Using this inequality, we can derive the update rule as a maximizer of an lower bound of the objective as follows. When we update only the $i$-th component (i.e., $\delta g_j = 0$ ($\forall j \neq i$)) for randomly chosen index $i$, the dual objective can be bounded as follows:

$$D(\check{g} + \delta g) \geq D(\check{g}) + \frac{1}{n}(\ell_i^*(\check{g}_i) - \ell_i^*(\check{g}_i + \delta g_i)) + \frac{1}{n}\int h_i(\theta)p[\check{g}](\theta)\mathrm{d}\theta\delta g_i - \frac{1}{2n^2\lambda_2}\delta g_i^2.$$

Hence, by substituting $\check{g} \leftarrow \check{g}^{(t)}$ and maximizing the right hand side with respect to $\delta g_i = g_i - \check{g}_i^{(t)}$, we obtain the following *ideal* version of the update rule of $\check{g}^{(t)}$:

$$\check{g}_i^{(t+1)} \leftarrow \underset{g_i}{\mathrm{argmax}} \left\{ -\ell_i^*(g_i) + \int h_i(\theta)p[\check{g}^{(t)}](\theta)\mathrm{d}\theta(g_i - \check{g}_i^{(t)}) - \frac{1}{2n\lambda_2}(g_i - \check{g}_i^{(t)})^2 \right\}, \quad (3)$$

while the other components are not updated: $\check{g}_j^{(t+1)} = \check{g}_j^{(t)}$ ($j \neq i$). Note that this update requires only one data point while the full gradient descent requires the whole $n$ data points. Along with the updated dual variable $\check{g}^{(t)}$, we update the primal variable as $\check{p}^{(t)} = p[\check{g}^{(t)}]$. The whole optimization procedure based on this update rule is summarized in Algorithm 1.

---

[2]The convex conjugate $f^*$ of a convex function $f : \mathbb{R} \to \mathbb{R}$ is defined by $f^*(u) := \sup_{x \in \mathbb{R}}\{ux - f(x)\}$.

[3]We write $[N] := \{1, \ldots, N\}$ for an integer $N$.

[4]$f : \mathbb{R}^D \to \mathbb{R}$ is $L$-smooth if $f$ is differentiable and $\|\nabla f(x) - \nabla f(y)\| \leq L\|x - y\|$ for all $x, y \in \mathbb{R}^D$.

---

**Algorithm 1** Dual Coordinate Descent in the continuous limit

---

**Require:** training data $\{(x_i, y_i)\}_{i=1}^n$ and number of iterations $t_{\text{end}}$

1: Choose $\check{g}_i^{(0)}$ s.t. $|\ell_i^{*\prime}(\check{g}_i^{(0)})| \leq 1$ $(i = 1, \ldots, n)$ and $\ell_i^*(\check{g}_i^{(0)}) \leq \ell_i^*(0)$
2: **for** $t = 0, 1, \ldots, t_{\text{end}}$ **do**
3:     Randomly choose $i_t$ from $\{1, 2, \ldots, n\}$
4:     $\check{g}_{i_t}^{(t+1)} \leftarrow \operatorname{argmax}_{g_{i_t} \in \mathbb{R}} \left\{ -\ell_{i_t}^*(g_{i_t}) + \int h_{i_t}(\theta) \check{p}^{(t)}(\theta) \mathrm{d}\theta (g_{i_t} - \check{g}_{i_t}^{(t)}) - \frac{1}{2n\lambda_2} |g_{i_t} - \check{g}_{i_t}^{(t)}|^2 \right\}.$
5:     Update the primal solution as $\check{p}^{(t+1)} = p[\check{g}^{(t+1)}]$.
6: **end for**
7: **return** $g^{(t_{\text{end}})}$

---

## 3.2 Our proposal: Particle update method

The update (3) requires an integration with respect to $p[\check{g}^{(t)}]$. We approximate the integral by a weighted sum over $M$ particles:

$$\int h_i(\theta) p[\check{g}^{(t)}](\theta) \mathrm{d}\theta \approx \frac{\sum_{m=1}^M r_m h_i(\theta_m)}{\sum_{m=1}^M r_m}, \tag{4}$$

where $(\theta_m)_{m=1}^M$ are particles that approximately cover the support of $p[\check{g}^{(t)}]$, and $(r_m)_{m=1}^M$ are the weights of each particle. Therefore, the key point of designing an efficient algorithm is to construct a "good" set of particles $(\theta_m)_{m=1}^M$ and their weights $(r_m)_{m=1}^M$.

The construction of particle approximation consists of two stages: (1) the sampling stage, and (2) the re-weighting stage. We perform the sampling stage once every $\tilde{n} \in [n]$ iterations, and perform the re-weighting stage in the other iterations. The detailed description of each stage is given below.

**(1) Sampling stage.** In the sampling stage, we sample $M$ i.i.d. particles $(\theta_m)_{m=1}^M$ from the density function $p[g^{(t)}]$ by using a Monte Carlo sampler, and place an even weight $r_m^{(t)} = 1/M$ ($m = 1, \ldots, M$) on the sampled particles $(\theta_m)_{m=1}^M$. As for the sampler, we may employ the *unadjusted gradient Langevin algorithm* (ULA), or the *Metropolis-adjusted Langevin algorithm* (MALA), both described in the following subsection. Note that the distribution of particles obtained from the Monte Carlo samplers as listed above may be biased away from the target distribution. We tolerate this inaccuracy by only requiring the particles to obey an approximated density function $p^{(t)}$ that is in $\epsilon_C^{(t)}$ distance from the target density $p[g^{(t)}]$ measured by the total variation distance[5]. Our convergence rate analysis will take this error into account.

**(2) Re-weighting stage.** Since the sampling procedure requires relatively heavy computation, it is desirable to perform this operation once in a while. We consider performing the sampling step *once in $\tilde{n}$ iterations*. In between the sampling stages, we instead perform the re-weighting stage, in which we iteratively update the weight of each particle, rather than resampling them. Recall that $p[g^{(t)}](\theta) \propto q[g^{(t)}](\theta) = \exp\left\{ -\frac{1}{\lambda_2} \left( \frac{1}{n} \sum_{i=1}^n h_i(\theta) g_i^{(t)} + \lambda_1 \|\theta\|^2 \right) \right\}$ which yields $q[g^{(t+1)}] = q[g^{(t)}] \exp[-h_{i_t}(\theta)(g_{i_t}^{(t+1)} - g_{i_t}^{(t)})/(n\lambda_2)]$ for the index $i_t$ chosen at the $t$-th iteration. We adapt this relation to update the weight $r_m^{(t)}$ of particles as

$$r_m^{(t+1)} = r_m^{(t)} \exp[-h_{i_t}(\theta_m)(g_{i_t}^{(t+1)} - g_{i_t}^{(t)})/(n\lambda_2)],$$

during the re-weighting stage because $r_m^{(t)}$ represents "importance" of $\theta_m$ which is proportional to $p[g^{(t)}](\theta_m)/p[g^{(\tilde{n}T)}](\theta_m)$ so that the particle approximation (4) is (nearly) unbiased. We allow the number of particles to depend on the outer-loop iteration $T$, and write $M^{(\tilde{n}T)}$ as the number of particles in the $T$-th outer-loop. The entire procedure is summarized in Algorithm 2.

Finally, we output the solution of the last iterate. We call this output as *Option (A)* and write its solution as $g_{\text{out}}^{(A)}$. Alternatively, there is another choice *Option (B)*, which randomly selects the stopping time $t'_{\text{end}}$ from the final $n$-iterations, i.e., that is randomly chosen from $\{\tilde{n}T_{\text{end}} - n + 1, \ldots, \tilde{n}T_{\text{end}}\}$, and returns the solution $g_{\text{out}}^{(B)} = g^{(t'_{\text{end}})}$. This technique improves the accuracy up to $O(n)$ factor. Therefore, we can reduce the number of particles by $O(n)$ times.

---

[5]The *total variation (TV) distance* between two probability measures with densities $p, q$ is defined as $\mathrm{TV}(p\|q) := \frac{1}{2} \int |p - q| \mathrm{d}\theta$.

---

**Algorithm 2** Dual Coordinate Descent with the particle method

---

**Require:** training data $\{(x_i, y_i)\}_{i=1}^n$ and numbers of inner-loop iterations $\tilde{n}$ and outer-loop iterations $T_{\text{end}}$,

1: Choose $g_i^{(0)}$ s.t. $|\ell_i^{*\prime}(g_i^{(0)})| \leq 1$ $(i = 1, \ldots, n)$ and $\ell_i^*(g_i^{(0)}) \leq \ell_i^*(0)$
2: **for** $T = 0, 1, \ldots, T_{\text{end}} - 1$ **do**
3:      Randomly (approximately) draw i.i.d. parameters $\theta_m$ $(m = 1, \ldots, M^{(\tilde{n}T)})$ from $p^{(\tilde{n}T)}(\theta)\mathrm{d}\theta$
     that satisfies $\mathrm{TV}(p^{(\tilde{n}T)}||p[g^{(\tilde{n}T)}]) \leq \epsilon_C^{(\tilde{n}T)}$.
4:      $r_m^{(\tilde{n}T)} \leftarrow \frac{1}{M^{(\tilde{n}T)}}$ $(m = 1, \ldots, M^{(\tilde{n}T)})$
5:      **for** $t = \tilde{n}T, \tilde{n}T + 1, \ldots, \tilde{n}T + \tilde{n} - 1$ **do**
6:          Randomly choose $i_t$ from $\{1, 2, \ldots, n\}$
7:          $g_{i_t}^{(t+1)} \leftarrow \underset{g_{i_t} \in \mathbb{R}}{\operatorname{argmax}} \left\{ -\ell_{i_t}^*(g_{i_t}) + \frac{\sum_{m=1}^{M^{(\tilde{n}T)}} r_m^{(t)} h_{i_t}(\theta_m)}{\sum_{m=1}^{M^{(\tilde{n}T)}} r_m^{(t)}} (g_{i_t} - g_{i_t}^{(t)}) - \frac{1}{2n\lambda_2}(g_{i_t} - g_{i_t}^{(t)})^2 \right\}$.
8:          $r_m^{(t+1)} \leftarrow r_m^{(t)} \exp\left(-\frac{1}{n\lambda_2} h_{i_t}(\theta_m)(g_{i_t}^{(t+1)} - g_{i_t}^{(t)})\right)$ $(m = 1, \ldots, M^{(\tilde{n}T)})$.
9:      **end for**
10: **end for**
11: **return** Option (A): $g_{\text{out}}^{(A)} = g^{(\tilde{n}T_{\text{end}})}$; Option (B): $g_{\text{out}}^{(B)} = g^{(t'_{\text{end}})}$ for $t'_{\text{end}}$ that is randomly chosen from $\tilde{n}(T_{\text{end}} - \lceil \frac{n}{\tilde{n}} \rceil) + 1, \ldots, \tilde{n}T_{\text{end}}\}$.

---

**Sampling algorithms.** Here, we present two examples that can be applied to sampling the particles $(\theta_m)_{m=1}^M$ from $p[g](\theta) \propto \exp(-U(\theta))$ where $U(\theta)$ is the potential function defined as $U(\theta) := \frac{1}{\lambda_2}\left(\frac{1}{n}\sum_{i=1}^n h_i(\theta)g_i + \lambda_1\|\theta\|^2\right)$.

**(1) ULA (unadjusted gradient Langevin algorithm):** The algorithm generates samples from the following noisy gradient descent update:

$$\theta^{k+1} = \theta^k - \eta\nabla U(\theta^k) + \sqrt{2\eta}\xi_k,$$

where $\eta > 0$ is the step-size and $\xi_k \sim N(0, I)$ is a random noise generated from the $\tilde{d}$-dimensional standard normal distribution. The sampling efficiency of ULA has been extensively studied in the literature (Dalalyan, 2017; Durmus & Moulines, 2017; Vempala & Wibisono, 2019).

**(2) MALA (Metropolis-adjusted Langevin algorithm):** The algorithm combines ULA with an accept-reject step according to the Metropolis–Hastings algorithm. That is, we generate a proposal $\tilde{\theta}^{k+1}$ by ULA, i.e., $\tilde{\theta}^{k+1} = \theta^k - \eta\nabla U(\theta^k) + \sqrt{2\eta}\xi_k$, which is accepted with probability $\alpha$ given by

$$\alpha = \min\left\{1, \frac{U(\tilde{\theta}_{k+1})q(\theta^k|\tilde{\theta}^{k+1})}{U(\theta_k)q(\tilde{\theta}^{k+1}|\theta^k)}\right\},$$

where $q(\theta'|\theta) \propto \exp(-\|\theta' - \theta - \eta\nabla U(\theta)\|^2/4\eta)$. If $\tilde{\theta}^{k+1}$ is not accepted, we set $\tilde{\theta}^{k+1} = \theta^k$. The convergence rate of MALA has been studied by Bou-Rabee & Hairer (2013); Ma et al. (2019); Chen et al. (2020a), to name a few. Due to the Metropolis-Hastings step, MALA allows for larger step size than ULA, which leads to an improved iteration complexity with respect to the desired accuracy $\epsilon_C$.

## 4 CONVERGENCE ANALYSIS

We present the convergence analysis of our method under the following assumptions.

**Assumption 2.**

**(A3)** $\inf_x \ell_i(x) > -\infty$ holds for all $i \in [n]$. There exists a constant $\gamma'$ such that $\forall i \in [n]$, $\ell_i^*$ is $\gamma'$-smooth in the interval $\{y \in \mathrm{dom}(\ell_i^*) \mid \partial\ell_i^*(y) \cap [-2, 2] \neq \emptyset\}$. $B_1 := \sup\{|\ell_i(x) - \inf_{x'}\ell_i(x')| \mid |x| \leq 1, i \in [n]\}$ and $B_2 := \sup\{|\ell_i'(x)| \mid |x| \leq 1, i \in [n]\}$ are both finite.

**(A4)** $h_i$ is $L$-smooth for all $i = 1, \ldots, n$.

It is straightforward to verify the finiteness of $B_1$, $B_2$ and $\inf_x \ell_i(x)$ in (A3) for several practical loss functions such as the squared loss and logistic loss. The smoothness assumption ($\gamma'$) on $\ell_i^*$ is required to fill in the gap between $g^{(t)}$ and $\check{g}^{(t)}$, and is also satisfied by the loss functions listed above: the squared loss satisfies it with $\gamma' = 1$ and the logistic loss satisfies it with $\gamma' = (e^2 + 1)^2/e^2$. The

second assumption (A4) is required to prove convergence of the sampling procedure, and can be omitted if we employ a particle sampler that does not require smoothness of the density function. This assumption is satisfied, for example, by a (truncated) two-layer neuron $h_\theta(x) = \tanh(r)\sigma(w^\top x)$ for $\theta = (r, w) \in \mathbb{R} \times \mathbb{R}^d$ (which also satisfies the boundedness assumption (A2)). We are now ready to present the convergence rate of our proposed method Algorithm 2. Here, we define $\tilde{s} \overset{\text{def}}{=} \frac{\tilde{n}\gamma\lambda_2}{2(1+n\gamma\lambda_2)}$.

**Theorem 1.** *Suppose that Assumptions 1 and 2 hold. Let $t_{\text{end}} = \tilde{n}T_{\text{end}}$. There exist constants $\hat{C}_1$, $\hat{C}_2$ and $\hat{C}_3$ which can depend on $\tilde{s}$ and $\lambda_2$ such that, if*

$$\epsilon_C^{(\tilde{n}T)} \leq \hat{C}_1 \exp(-\tilde{s}T/2), \quad M^{(\tilde{n}T)} \geq \frac{\hat{C}_2^2}{(\epsilon_C^{(\tilde{n}T)})^2} \log(4nT_{\text{end}}/\delta) \quad (\forall T \in [T_{\text{end}}]), \tag{5}$$

*for $0 < \delta < 1$ and $\lambda_2 \geq 2B/n$, then there exists an event $\mathcal{E}$[6] satisfying $P(\mathcal{E}) \geq 1 - \delta$ in which the conditional expectation of the duality gap can be bounded by $\mathbb{E}\left[P(p[g_{\text{out}}^{(A)}]) - D(g_{\text{out}}^{(A)}) \,\middle|\, \mathcal{E}\right] \leq \epsilon_P$ as long as the number of iterations $t_{\text{end}}$ satisfies the following conditions:*

(Option A) $\qquad t_{\text{end}} \geq 2\left(n + \frac{1}{\gamma\lambda_2}\right) \log\left(\left(n + \frac{1}{\gamma\lambda_2}\right)\frac{\hat{C}_3}{\epsilon_P}\right), \tag{6}$

(Option B) $\qquad t_{\text{end}} \geq 2\left(n + \frac{1}{\gamma\lambda_2}\right) \log\left(\left(1 + \frac{1}{n\gamma\lambda_2}\right)\frac{\hat{C}_3}{\epsilon_P}\right) + \left\lceil\frac{n}{\tilde{n}}\right\rceil. \tag{7}$

*Moreover, the constants have the following dependency on $n, \tilde{n}, \lambda_2, \tilde{s}$: $\hat{C}_2 = \exp(\frac{4B_1\tilde{n}}{\lambda_2 n})$, $\hat{C}_1^{-1} = O\left(\lambda_2^{-1/2}\hat{C}_2 \exp\left(\frac{\tilde{n}(2\hat{C}_2+1)}{n\gamma\lambda_2+1}\right)\right)$ and $\hat{C}_3 = O(\max\{\tilde{s}^{-1}, \lambda_2^{-3/2}(1 + \tilde{s}^{-1/2})\})$.*

The proof and detailed descriptions of each constant are given in Appendix C. The theorem indicates that our proposed method achieves exponential convergence with respect to the number of outer iterations. This coincides with the convergence rate of the finite dimensional version of SDCA (Shalev-Shwartz & Zhang, 2013a). In contrast, a (non-stochastic) full-batch gradient descent requires $O((n/\lambda_2\gamma)\log(1/\epsilon_P))$ total gradient evaluations to optimize a convex objective with a *condition number* $1/(\lambda_2\gamma)$. Importantly, SDCA (and our method) "decouples" the sample size $n$ and the condition number $1/(\lambda_2\gamma)$ yielding only $O((n + 1/\lambda_2\gamma)\log(1/\epsilon_P))$ iterations, which is beneficial in a setting of big data and a small regularization parameter.

The difference between options (A) and (B) is only an $O(\log(n))$ factor in terms of the number of iterations. However, this improvement affects the number of particles given by Eq. (5). Indeed, the $\log(n)$ factor improvement of $t_{\text{end}}$ allows us to take $M^{(\tilde{n}T_{\text{end}})} \geq \Omega(\hat{C}_2^2\hat{C}_3/(\hat{C}_1^2\epsilon_P))$ in the last inner loop, whereas option (A) requires $M^{(\tilde{n}T_{\text{end}})} \geq \Omega(n\hat{C}_2^2\hat{C}_3/(\hat{C}_1^2\epsilon_P))$. Either way, the number of particles can be of *linear order* in terms of the inverse of the target precision $\epsilon_P$.

**Remark 1.** *Depending on the regularization strength $\lambda_2$, the re-sampling interval $\tilde{n}$ should be chosen differently. For constant order (non-vanishing) $\lambda_2$, we may take the re-sampling interval as $\tilde{n} = n$ so that all constants are bounded. On the other hand, in the small $\lambda_2$ regime, say $\lambda_2 = O(1/n)$, we have $\hat{C}_2 = \exp(O(\tilde{n}))$, and there also appears an exponential dependency on $\hat{C}_1$ w.r.t. $\tilde{n}$. In this case, $\tilde{n} = O(1)$ is a better choice to avoid the large particle complexity. But for such choice of $\tilde{n}$, we should re-sample at almost every iteration, which is computationally demanding; this presents a trade-off. In practice, re-sampling can be executed in parallel and thus efficient training (using GPU) is possible. Importantly, to obtain the usual $O(1/\sqrt{n})$-generalization error, $\lambda_2 = 1/\sqrt{n}$ is sufficient (Nitanda et al., 2021) and thus we may take $\tilde{n} = \Omega(\sqrt{n})$ in that situation. We however conjecture that this trade-off is unavoidable without additional assumptions.*

**Sampling complexity.** In Theorem 1, it is assumed that the density $p^{(\tilde{n}t)}$ from which the particles are generated is within $\epsilon_C^{(\tilde{n}T)}$ from the target density $p[g^{(\tilde{n}T)}]$ measured by the total variation distance. The next proposition provides an upper bound of the number of iterations required to obtain the $\epsilon_C^{(\tilde{n}T)}$ accurate distribution by the Monte-Carlo samplers such as ULA and MALA. The key property of the target density that determines the convergence rate of these methods is the *log-Sobolev inequality* with a constant $c_{\text{LS}}$ (see Definition 2 for its precise definition).

---

[6]Intuitively, the event $\mathcal{E}$ represents an event in which "good" particles are sampled from the sampler to approximate the integral.

**Proposition 2.** *Under Assumptions 1 and 2, the target density $p[g^{(\tilde{n}T)}]$ at each outer iteration satisfies the log-Sobolev inequality with constant $c_{\mathrm{LS}} = \frac{2\lambda_1}{\lambda_2} \exp(-4B_2/\lambda_2)$. Accordingly, the number of iterations $K_*^{(T)}$ to obtain the $\epsilon_C^{(\tilde{n}T)}$-accurate distribution can be evaluated as follows:*

(ULA)
$$K_*^{(T)} \le \frac{4L^2}{c_{\mathrm{LS}}^2} \max\left\{1, \frac{4\tilde{d}}{(\epsilon_C^{(\tilde{n}T)})^2}\right\} \log\left[4B_2/(\lambda_2 \epsilon_C^{(\tilde{n}T)})\right],$$

(MALA)
$$K_*^{(T)} \le O\left[\frac{1}{c_{\mathrm{LS}}^{5/2}}\left(\frac{B_2(1+L)+\lambda_1}{\lambda_2} + \log(1/\epsilon_C^{(\tilde{n}T)})\right)^{3/2} \tilde{d}\right].$$

The proof and a more detailed statement can be found in Appendix B. Roughly speaking, this proposition states that the number of iterations for ULA can be simplified as $K_*^{(T)} \ge \Omega\big((c_{\mathrm{LS}}\epsilon_C^{(\tilde{n}T)})^{-2}\log(n)\big)$. On the other hand, MALA requires only poly-log order complexity with respect to $\epsilon_C^{(\tilde{n}T)}$ as $K_*^{(T)} = \Omega\big(c_{\mathrm{LS}}^{-5/2}\log(1/\epsilon_C^{(\tilde{n}T)})^{3/2}\big)$, which is therefore more preferable to obtain higher accuracy. Combining the required number of particles $M^{(\tilde{n}T)}$ and the number of iterations $K_*^{(T)}$ for the sampler, we see that the total computational complexity is still of polynomial order.

**Remark 2.** *Note that the log-Sobolev constant $c_{\mathrm{LS}}$ also depends exponentially on $1/\lambda_2$. While the exponential factor can be avoided in the outer loop (Theorem 1) by appropriately setting $\tilde{n}$, such dependence is known to be unavoidable for $c_{\mathrm{LS}}$ in the most general setting (Menz & Schlichting, 2014). An interesting direction is to explore additional assumptions that remove this dependence.*

**Total complexity.** By summarizing the results we obtained above, the total computational complexity of our algorithm (Algorithm 2) with the MALA sampler can be roughly bounded as

$$\sum_{T=1}^{T_{\mathrm{end}}} M^{(\tilde{n}T)}(1 + nK_*^{(T)}/\tilde{n})\tilde{n} = O\left(\frac{n\log(n)}{\tilde{n}\epsilon_P}\left(n + \frac{1}{\lambda_2\gamma}\right)\left[\left(\left(1 + \frac{1}{\lambda_2}\right)\frac{\log(\epsilon_P^{-1})}{c_{\mathrm{LS}}}\right)^{5/2}\right]\right),$$

where we assumed $\hat{C}_1^{-1}, \hat{C}_2 = O(1)$ (see Remark 1 on how this is achieved). Detailed evaluation of the computational complexity with the ULA sampler can be found in Appendix B.2.

## 5 ADDITIONAL RELATED WORKS

Hu et al. (2019); Sirignano & Spiliopoulos (2020) showed a linear convergence rate of the McKean-Vlasov dynamics. However, the theory only handles the infinite width limit and continuous time dynamics, and it is from trivial to covert their results to a discrete time and finite width setting. Mei et al. (2018) also analyzed space-time discretization of a similar dynamics. However, a quantitative convergence rate is not provided, and the time discretization error grows exponentially with the time horizon. Compared to these results, we deal with general $(1/\gamma)$-smooth loss functions and made the dependency on the parameters $\lambda_2$ and $\gamma$ explicit in our convergence rate. Bou-Rabee & Schuh (2020); Bou-Rabee & Eberle (2021) showed an exponential convergence of gradient Langevin dynamics with an interaction potential, which includes certain mean field models. When specialized to mean field neural networks, our formulation covers a much wider range of objectives and loss functions. Furthermore, they considered only the Hamiltonian Monte Carlo sampler, and the analysis relied on the coupling between the discrete time dynamics and the continuous time counterpart. Consequently, their outer-loop iteration complexity is larger than $O(\log(1/\epsilon_P)/\epsilon_P)$, whereas our bound requires only poly-log order cost due to the flexible choice of sampling algorithm. Jabir et al. (2021) also considered spatial-time discretization of mean field dynamics of deep neural network. However, it contains the time discretization lag $h$ in their convergence rate, while our method is purely discrete time and is not affected by such a time discretization error. Moreover, their analysis requires a strong regularization (which corresponds to $\lambda_1$ in our setting) so that the objective becomes convex-like with respect to each particle to show the geometric ergodicity; this is not required in our analysis.

## 6 NUMERICAL EXPERIMENTS

We verify our convergence rate analysis on regression problems with the squared loss. We consider teacher-student setup in which teacher and student are both two-layer neural networks of different width. The training dataset is generated by the teacher model: $y_i = \sigma(w^{*\top}x_i + b^*) + \epsilon_i$, where $x_i$ and $\epsilon_i$ are independently drawn from Gaussian distributions. We optimize a tanh neural network $(h_\theta(x) = \tanh(w^\top x + b)$ with $\theta = (w, b))$ of width $M_s$ as a student using Algorithm 2.

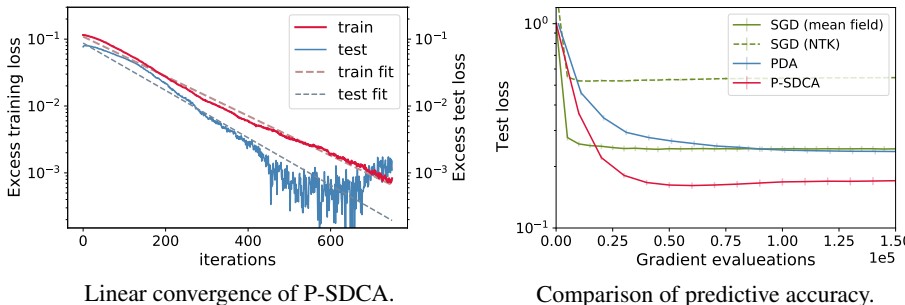

Figure 1: (Left) Linear convergence of PSDCA to the optimal value of the regularized objective, with test loss as for the supplementary. (Right) Comparison of predictive accuray of P-SDCA, PDA, and SGD. The error-bar indicates the standard error over 10 independent repetitions.

**Convergence rate of P-SDCA.** We verify the linear convergence rate of P-SDCA as suggested by our theoretical analysis in a simple setting. To verify the convergence rate, we run the method with 1000-steps of ULA. As for the teacher model, we use the ReLU activation for $\sigma$ and (note that even though this is a simple setting, the teacher model is "not" included in the student model). The number of examples and dimensionality of the input space are $n = 500$ and $d = 50$, respectively. The other hyperparameters are set to $\lambda_1 = 10^{-4}$, $\lambda_2 = 10^{-4}$, and $M_s = 200$. The left side of Figure 1 depicts the convergence of the primal objective to the optimal value. The $x$-axis is the number of outer-iteration and the $y$-axis is the excess primal objective $P(\cdot) - \inf_p P(p)$ including the regularization terms on a training set and a test set in log-scale. The integral with respect to the primal solution was computed by the particle average and the entropy is estimated by using the $k$-NN entropy estimator (Kozachenko & Leonenko, 1987; Brodersen, 2020). Since the $y$-axis is log-scaled, we can clearly observe the linear convergence of P-SDCA.

**Comparison of predictive accuracy with SGD and PDA.** We employ a sign activation for the teacher model ($\sigma(\cdot) = \text{sign}(\cdot)$), and the student is an overparameterized tanh network of width $M_s = 200$. The number of examples and input dimensionality are $n = 1000$ and $d = 50$, and we set $\lambda_1 = 10^{-3}$, $\lambda_2 = 10^{-4}$. As for SGD, we cannot apply the entropic regularization and thus the term is omitted in the optimization. We also used the neural tangent kernel (NTK) scaling $(f_\Theta(x) = \frac{1}{\sqrt{M}} \sum_{m=1}^{M} h_{\theta_m}(x))$ for SGD optimization in addition to the mean field scaling. The test error with respect to the number of gradient evaluations is displayed in the right side of Figure 1. We make the following observations: (i) P-SDCA achieves a better test error than other methods; (ii) SGD achieves minimal test error at the early stage but does not perform well afterwards; (iii) the mean field parameterization outperforms the NTK parameterization in terms of test loss; one possible explanation of this observed advantage is the presence of *feature learning*, which we empirically demonstrate in Appendix D.

## 7 CONCLUSION

In this paper, we proposed a new stochastic optimization technique for two-layer neural networks with the mean field representation. Our algorithm can be seen as an infinite dimensional extension of SDCA that maintains its advantage, namely, that it converges linearly with respect to the outer-iteration, and is much more efficient in terms of the sample size than non-stochastic methods. In the algorithm, we approximate the integral required in each update by an average over finite number of particles. Remarkably, the number of particles can be of linear order with respect to the required precision, and its sampling cost can be of poly-log order. This is significantly different from existing numerical approximation approaches of McKean-Vlasov equations, which typically require either vanishing step size or exponentially large particle size. We validated the linear convergence of P-SDCA in numerical experiments, and demonstrated that it outperforms other existing methods.

In the convex optimization literature, it is known that the SDCA algorithm considered in this work is not rate optimal – the optimal algorithm can be obtained by combining Nesterov's acceleration with stochastic methods (Allen-Zhu, 2017; Murata & Suzuki, 2017). Extending our algorithm to such an optimal accelerated method is an interesting future direction. In addition, while the exponential dependence on $1/\lambda_2$ in the log-Sobolev constant may be unavoidable in the general setting, it is worth investigating whether such dependency can be improved under structural assumptions.

ACKNOWLEDGMENT

TS and KO were partially supported by JSPS KAKENHI (18H03201), Japan Digital Design and JST CREST. AN was partially supported by JSPS Kakenhi (19K20337) and JST-PRESTO (JPMJPR1928).

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

## TABLE OF CONTENTS

# ——Appendix——

## A FENCHEL'S DUALITY THEOREM (PROOF OF PROPOSITION 1)

To prove Proposition 1, we utilize the infinite dimensional Fenchel's duality theorem by Rockafellar (1967). Two real vector spaces $E$ and $E^*$ with locally convex Hausdorff topology are called topologically paired vector spaces if all the elements of each space can be identified with all the continuous linear functionals on the other. Therefore, for a continuous linear functional $y : X \to \mathbb{R}$, there exists $y \in E^*$ such that we can write $y(x) = \langle y, x \rangle_{E^*,E}$ as a bilinear form $\langle \cdot, \cdot \rangle_{E^*,E}$.

**Definition 1** (Fenchel-Legendre convex conjugate). *Let $X, X^*$ be topologically paired vector spaces. For $f : X \to \mathbb{R} \cup \{\infty\}$, its Fenchel-Legendre convex conjugate $f^* : X^* \to \mathbb{R} \cup \{\infty\}$ is defined by*

$$f^*(y) = \sup\{\langle y, x \rangle_{X^*,X} - f(x) \mid x \in X\}.$$

**Remark 3.** *When $X = X^* = \mathbb{R}^d$ equipped with the usual inner product, the definition of the Fenchel-Legendre convex conjugate coincides with the definition introduced in the main text.*

**Lemma 2** (Theorems 3 and 8 of Rockafellar (1967)). *Let $(X, X^*)$ and $(Y, Y^*)$ be two pairs of topologically paired vector spaces. Let $f : X \to \mathbb{R} \cup \{\infty\}$ and $g : Y \to \mathbb{R} \cup \{\infty\}$ are proper lower semi-continuous convex functions, and let $A : X \to Y$ be a bounded linear operator. Assume that $\exists x \in X$ such that $f(x)$ is finite and $g$ is finite and continuous at $Ax$. Then,*

$$\inf\{g(Ax) + f(x) \mid x \in X\} = \sup\{-g^*(y) - f^*(-A^*y) \mid y \in Y^*\},$$

*where $A^*$ is the adjoint operator of $A$ such that $\langle y, Ax \rangle_{Y^*,Y} = \langle A^*y, x \rangle_{X^*,X}$. Moreover, $\hat{x}$ and $\hat{y}$ are the optimal solutions of the primal and dual problems respectively if and only if $A\hat{x} \in \partial g^*(\hat{y})$ and $A^*\hat{y} \in \partial f(\hat{x})$.*

Now, we are ready to prove Proposition 1.

*Proof of Proposition 1.* We utilize Lemma 2 to show the assertion. We let $Y = \mathbb{R}^n$. Then, we know that $Y$ is self-dual, that is, its topological dual $Y^*$ is $Y$ itself where the linear functional is given by the standard inner product $\langle u, v \rangle_{Y^*,Y} = \sum_{i=1}^n u_i v_i$ for $u = (u_i)_{i=1}^n \in Y$ and $v = (v_i)_{i=1}^n \in Y^*$. We let $X$ be the set of $L^1$-integral functions with respect to the Lebesgue measure, i.e., $X = L^1(\mathbb{R}^{\tilde{d}}) = L^1(\mathbb{R}^{\tilde{d}}; \mu)$, where $\mu$ is the Lebesgue measure on $(\mathbb{R}^{\tilde{d}}, \mathcal{B}(\mathbb{R}^{\tilde{d}}))$. It is known that its topological dual is the set of essentially bounded functions equipped with $L^\infty$-norm, i.e., $X^* = L^\infty(\mathbb{R}^{\tilde{d}})(= L^\infty(\mathbb{R}^{\tilde{d}}; \mu)) = \{f : \mathbb{R}^{\tilde{d}} \to \mathbb{R} \mid f \text{ is } \mathcal{B}(\mathbb{R}^{\tilde{d}})\text{-measurable}, \|f\|_\infty < \infty\}$, where $\|f\|_\infty = \inf\{M > 0 \mid \mu(\{x \mid |f(x)| \geq M\}) = 0\}$ for the Lebesgue measure $\mu$. The bilinear functional $\langle \cdot, \cdot \rangle_{X^*,X}$ is given by $\langle g, f \rangle_{X^*,X} = \int f g \mathrm{d}\mu$. Then, $X, X^*$ become topologically paired vector spaces by equipping $X^*$ with its weak* topology with respect to the inner product $\langle \cdot, \cdot \rangle_{X^*,X}$.

We let $A : L^1(\mathbb{R}^{\tilde{d}}) \to \mathbb{R}^n$ be $Af = (\int h_i(\theta)f(\theta)\mathrm{d}\theta)_{i=1}^n \in \mathbb{R}^n$ for $f \in L^1(\mathbb{R}^{\tilde{d}})$. Then, we can see that $A$ is a bounded linear functional by the boundedness of $h_i(\cdot)$.

It is also known that the KL divergence is jointly lower semi-continuous with respect to the weak convergence topology (Posner, 1975, Theorem 1). Since the $L^1$-norm between probability density functions is equivalent to the total variation norm between the corresponding probability distributions, and the total variation norm gives a stronger topology than the weak topology, KL divergence is lower semi-continuous with respect to the $L^1$-norm of the corresponding density functions. It is also well known that KL divergence is jointly convex and it is easy to check that $p \in \mathcal{P} \mapsto \mathrm{KL}(p, N(0, \sigma^2 I))$ is proper, i.e., $\exists p \in \mathcal{P}$ such that $\mathrm{KL}(p, N(0, \sigma^2 I)) < \infty$.

Under these settings, if we set $g(u) = \frac{1}{n}\sum_{i=1}^n \ell_i(u_i)$ for $u = (u_i)_{i=1}^n \in Y = \mathbb{R}^{\tilde{d}}$ and

$$f(p) = \begin{cases} \lambda_2 \mathrm{KL}(p \| N(0, \lambda_1/\lambda_2 I)) & (p \in \mathcal{P}), \\ \infty & (\text{otherwise}), \end{cases}$$

for $p \in L^1(\mathbb{R}^{\tilde{d}})$, then we can see that $P(p) = g(Ap) + f(p)$ for $p \in \mathcal{P}$ and $\inf_{p \in \mathcal{P}} P(p) = \inf_{h \in X}\{g(Ah) + f(h)\}$. We can also verify the condition in Lemma 2. Indeed, if we let $p_0 \in \mathcal{P}$ be the density function of $N(0, \lambda_1/\lambda_2 I)$, then $f(p_0) = 0 < \infty$ and $g$ is continuous at $Ap_0$ because a

convex function is continuous when its effective domain is the whole space $\mathbb{R}^n$ (Rockafellar, 1976, Theorem 10.1), and the domain of $g$ is the whole space $\mathbb{R}^n$ (remember that the range of $\ell_i$ is $\mathbb{R}$).

Therefore, by applying Lemma 2 and going through a simple calculation to obtain $f^*$, we obtain the desired assertion. $\qquad \square$

## B   COMPUTATIONAL COMPLEXITY OF SAMPLING ALGORITHMS (PROOF OF PROPOSITION 2)

### B.1   LOG-SOBOLEV INEQUALITY OF THE TARGET FUNCTION

In each outer iteration, we sample $M$ particles from the density $p[g^{(\tilde{n}T)}]$ so that the particles are sampled from an approximated density $p^{(\tilde{n}T)}$ with $\mathrm{TV}(p^{(\tilde{n}T)}||p[g^{(\tilde{n}T)}]) \leq \epsilon_S^{(T)}$. To estimate the convergence of Monte-Carlo methods such as ULA and MALA, the log-Sobolev inequality associated with the target density plays the central role.

**Definition 2** (Log-Sobolev inequality). *Let $p(\theta)$ be a smooth probability density function on $\mathbb{R}^{\tilde{d}}$. Then, we say $p(\theta)$ (or its corresponding probability measure on $(\mathbb{R}, \mathcal{B}(\mathbb{R}^{\tilde{d}}))$) satisfies the log-Sobolev inequality (LSI) with a constant $c_{\mathrm{LS}} > 0$ if and only if, for any smooth function $\phi : \mathbb{R}^p \to \mathbb{R}$ with $\mathbb{E}_p[\phi^2] < \infty$, it holds that*

$$\mathbb{E}_p[\phi^2 \log(\phi^2)] - \mathbb{E}_p[\phi^2] \log(\mathbb{E}_p[\phi^2]) \leq \frac{2}{c_{\mathrm{LS}}} \mathbb{E}_p[\|\nabla \phi\|_2^2].$$

The following lemma shows that the target density $p[g^{(\tilde{n}T)}]$ satisfies the LSI.

**Lemma 3.** *Suppose $\max_i |g_i^{(\tilde{n}T)}| \leq B$ (this is true by Lemma 6 below with $B = B_2$) and assume (A2) of Assumption 1, then $p[g^{(\tilde{n}T)}]$ satisfies the LSI with the constant*

$$c_{\mathrm{LS}} = \frac{2\lambda_1}{\lambda_2} \exp(-4B/\lambda_2).$$

*Proof.* Remember that $p[g^{(\tilde{n}T)}]$ is given as

$$p[g^{(\tilde{n}T)}](\theta) \propto \exp\left(-\frac{1}{\lambda_2} \frac{1}{n} \sum_{i=1}^n h_i(\theta) g_i^{(\tilde{n}T)} - \frac{1}{2} \frac{2\lambda_1}{\lambda_2} \|\theta\|^2 \right).$$

Therefore, $p[g^{(\tilde{n}T)}](\theta)$ can be written as

$$p[g^{(\tilde{n}T)}](\theta) = \frac{\exp(f(\theta))q(\theta)}{\mathbb{E}_q[\exp(f(\cdot))]}, \tag{8}$$

where $f(\theta) = -\frac{1}{\lambda_2} \frac{1}{n} \sum_{i=1}^n h_i(\theta) g_i^{(\tilde{n}T)}$ and $q(\theta)$ is the probability density function of a Gaussian measure $N(0, \frac{\lambda_2}{2\lambda_1} I)$. Note that $\|f\|_\infty \leq B/\lambda_2$ by the assumption (A2) and $\max_i |g_i^{(\tilde{n}T)}| \leq B$. Then, it is well known that $q$ satisfies the LSI with a LSI constant $c_{\mathrm{LS}}(q) = \frac{2\lambda_1}{\lambda_2}$ and Holley & Stroock (1987) showed that the bounded perturbation as in Eq. (8) preserves the LSI condition where the LSI constant of $p[g^{(\tilde{n}T)}]$ is bounded by $c_{\mathrm{LS}} = c_{\mathrm{LS}}(q)/\exp(4\|f\|_\infty) \geq \frac{2\lambda_1}{\lambda_2} \exp(-4B/\lambda_2)$. This concludes the proof. $\qquad \square$

By Lemma 3, we show that existing convergence rate analyses of ULA and MALA such as Vempala & Wibisono (2019); Ma et al. (2019) can be applied to our target density $p[g^{(\tilde{n}T)}]$. From now on, we denote our target density $p[g^{(\tilde{n}T)}]$ by $p^*$, and the negative log density of $p[g^{(\tilde{n}T)}]$ by $U$, that is, $p^*(\theta) = \exp(-U(\theta))$. In addition, let $p_k$ be the marginal distribution of $\theta_k$ (sample in the $k$-th iteration of ULA or MALA).

### B.2 Sampling complexity of ULA

First, we apply the convergence rate analysis of ULA developed by Vempala & Wibisono (2019) to our setting. We utilize the following proposition given by Vempala & Wibisono (2019).

**Proposition 3** (Theorem 1 of Vempala & Wibisono (2019))**.** *Assume that $p^*(\theta)\mathrm{d}\theta$ satisfies the log-Sobolev inequality with a constant $c_{\mathrm{LS}}$ and $U$ is $L$-smooth. Under these assumptions, for the step size $\eta$ satisfying $\eta \leq \frac{c_{\mathrm{LS}}}{4L^2}$, we have that*

$$\mathrm{KL}\left(p_k \parallel p^*\right) \leq \exp(-c_{\mathrm{LS}}\eta k)\mathrm{KL}\left(p_1 \parallel p^*\right) + \frac{8\eta\tilde{d}L^2}{c_{\mathrm{LS}}}.$$

Proposition 3 implies that if the step size satisfies $\eta \leq \frac{c_{\mathrm{LS}}}{4L^2}\min\left\{1, \frac{\epsilon}{4\tilde{d}}\right\}$ and the number of iterations satisfies

$$k \geq \frac{1}{c_{\mathrm{LS}}\eta}\log\frac{2\mathrm{KL}\left(p_0 \parallel p^*\right)}{\epsilon},$$

then we obtain an $\epsilon$-accurate distribution $p_k$ in terms of the KL-divergence.

If we employ $p_0(\theta) \propto \exp(-\frac{\lambda_1}{\lambda_2}\|\theta\|^2)$ (a Gaussian measure), then $2\mathrm{KL}\left(p_0 \parallel p^*\right)$ is bounded by $\frac{4B_2}{\lambda_2}$, from Lemma 6:

$$\begin{aligned}
\mathrm{KL}\left(p_0 \parallel p^*\right) &= \int p_0(\theta)\log\frac{p_0(\theta)}{p^*(\theta)}\mathrm{d}\theta \\
&= -\int p_0(\theta)\left[\log p^*(\theta) - \log p_0(\theta)\right]\mathrm{d}\theta \\
&= -\int p_0(\theta)\left[\log\frac{\exp\left(\frac{-\sum_{i=1}^n h_i(\theta)g_i - \lambda_1\|\theta\|_2^2}{\lambda_2}\right)}{\int \exp\left(\frac{-\sum_{i=1}^n h_i(\theta')g_i - \lambda_1\|\theta'\|_2^2}{\lambda_2}\right)\mathrm{d}\theta'} - \log p_0(\theta)\right]\mathrm{d}\theta \\
&\leq -\int p_0(\theta)\left[\log\frac{\exp\left(-\frac{\sup_i|g_i|}{\lambda_2}\right)\exp\left(-\frac{\lambda_1\|\theta\|_2^2}{\lambda_2}\right)}{\exp\left(\frac{\sup_i|g_i|}{\lambda_2}\right)\int \exp\left(-\frac{\lambda_1\|\theta\|_2^2}{\lambda_2}\right)\mathrm{d}\theta} - \log p_0(\theta)\right]\mathrm{d}\theta \\
&\leq -\int p_0(\theta)\left[\log\left\{\exp\left(-\frac{2\cdot B_2}{\lambda_2}\right)\cdot p_0(\theta)\right\} - \log p_0(\theta)\right]\mathrm{d}\theta \\
&\leq \frac{2B_2}{\lambda_2}
\end{aligned}$$

Moreover, using Pinsker's inequality bounds total variation norm by KL-divergence:

$$\|p_k - p^*\|_{\mathrm{TV}} \leq \sqrt{2\mathrm{KL}\left(p_k \parallel p^*\right)}.$$

Therefore, in order to have an $\epsilon_C$-accurate distribution in total variation, it suffices to let the step size satisfy $\eta \leq \frac{c_{\mathrm{LS}}}{4L^2}\min\left\{1, \frac{\epsilon_C^2}{8\tilde{d}}\right\}$ and the number of iterations satisfy

$$k \geq \frac{2}{c_{\mathrm{LS}}\eta}\log\frac{4B_2}{\lambda_2\epsilon_C}.$$

Hence, the total computational complexity can be bounded by $O\left(\frac{\tilde{d}L^2}{c_{\mathrm{LS}}^2\epsilon_C^2}\log\frac{B_2}{\lambda_2\epsilon_C}\right)$.

**Total computational complexity of sampling by ULA** According to Theorem 1, sampling accuracy in total variation $\epsilon_C^{(\tilde{n}T)}$ should be smaller than $O(\exp\left(-\frac{\tilde{s}T}{2}\right))$ in each sampling step $T$. Therefore, the total computational complexity for sampling can be calculated as follows:

**Algorithm 2 with Option A:** Suppose we run outer-loop up to $T_{\mathrm{end}} \geq \frac{1}{\tilde{s}}\log\left(\left(n + \frac{1}{\gamma\lambda_2}\right)\frac{(B_1 + B_2 + \frac{C_0}{1-\exp(-\tilde{s})})}{\epsilon_P}\right)$ times. Indeed it is sufficient to have an $\epsilon_P$-approximated solution, according to Theorem 3, which gives a tighter and more detailed

bound than in Theorem 1. Then, the total required computational complexity for sampling is

$$\sum_{T=1}^{T_{\text{end}}} O\left(\frac{\tilde{d}L^2}{c_{\text{LS}}^2 (\epsilon_C^{(\tilde{n}T)})^2} \log \frac{B_2}{\lambda_2 \epsilon_C^{(\tilde{n}T)}}\right)$$

$$\leq O\left(\frac{\tilde{d}L^2}{\hat{C}_1^2 c_{\text{LS}}^2} \log \frac{B_2}{\lambda_2 \epsilon_C^{(\tilde{n}T_{\text{end}})}}\right) \sum_{T=1}^{T_{\text{end}}} \exp\left(\tilde{s}T\right)$$

$$= O\left(\frac{\tilde{d}L^2}{\hat{C}_1^2 c_{\text{LS}}^2} \log \frac{B_2}{\lambda_2 \epsilon_C^{(\tilde{n}T_{\text{end}})}}\right) \frac{\exp\left(\tilde{s}T_{\text{end}}\right)}{1 - e^{-\tilde{s}}}$$

$$= \tilde{O}\left(\frac{\left(n + \frac{1}{\gamma\lambda_2}\right)^2}{\tilde{n}\epsilon_P} \frac{\tilde{d}L^2}{c_{\text{LS}}^2} \log \frac{B_2(B_1 + B_2 + C_0)}{\lambda_2 + \frac{1}{n\gamma_2}}\right),$$

where $\tilde{O}$ hides $\log\log$ terms.

**Algorithm 2 with Option B:** According to Theorem 1, it is sufficient to run outer-loop up to $T_{\text{end}} \geq \frac{1}{\tilde{s}} \log\left(\left(1 + \frac{1}{n\gamma\lambda_2}\right)\frac{\hat{C}_3}{\epsilon_P}\right)$ times to acquire an $\epsilon_P$-approximated solution. You can see the detailed definition of $\hat{C}_3 = O(\max\{\tilde{s}^{-1}, \lambda_2^{-3/2}\tilde{s}^{-1/2}\})$ in Eqs. (52a)-(52c) of Theorem 3 and in *Proof of Theorem 1*. Therefore, the total required computational complexity for sampling is

$$\sum_{T=1}^{T_{\text{end}}} O\left(\frac{\tilde{d}L^2}{c_{\text{LS}}^2 \left(\epsilon_C^{(\tilde{n}T)}\right)^2} \log \frac{B_2}{\lambda_2 \epsilon_C^{(\tilde{n}T)}}\right) \exp\left(\tilde{s}T\right)$$

$$\leq O\left(\frac{\tilde{d}L^2}{\hat{C}_1^2 c_{\text{LS}}^2} \log \frac{B_2}{\lambda_2 \epsilon_C^{(\tilde{n}T_{\text{end}})}}\right) \sum_{T=1}^{T_{\text{end}}} \exp\left(\tilde{s}T\right)$$

$$= O\left(\frac{\tilde{d}L^2}{\hat{C}_1^2 c_{\text{LS}}^2} \log \frac{B_2}{\lambda_2 \epsilon_C^{(\tilde{n}T_{\text{end}})}}\right) \frac{\exp\left(\tilde{s}T_{\text{end}}\right)}{1 - e^{-\tilde{s}}}$$

$$= \tilde{O}\left(\max\left\{\lambda_2^{-3/2}, \left(\frac{n\gamma\lambda_2 + 1}{\tilde{n}\gamma\lambda_2}\right)^{1/2}\right\} \frac{n^{3/2}\left(1 + \frac{1}{n\gamma\lambda_2}\right)^{5/2}}{\tilde{n}^{3/2}\epsilon_P} \frac{\tilde{d}L^2}{c_{\text{LS}}^2} \log \frac{B_2(B_1 + B_2 + C_0)}{\lambda_2 + \frac{1}{n\gamma_2}}\right),$$

where $\tilde{O}$ hides $\log\log$ terms as well.

## B.3 SAMPLING COMPLEXITY OF MALA

Here, we adopt the convergence rate analysis of MALA obtained by Ma et al. (2019). They showed the convergence under the setting where $U$ is $\mu$-strongly convex outside of a compact domain and is $L$-smooth. Although our situation does not necessarily satisfy this condition, the proof of Ma et al. (2019) essentially utilizes it to show the LSI of the target density. Fortunately, we have already seen that the target density $p^*$ satisfies the LSI in Lemma 3. Thus, the analysis of Ma et al. (2019) can be applied to our setting.

From now on, we denote $p[g^{(\tilde{n}T)}]$ by $p^*$. Let $R(s)$ be a positive real such that $P^*(\|\theta\| \geq R(s)) \leq s$ for $0 < s < 1$ where $P^*$ is the probability measure corresponding to $p^*$ and $\tilde{L} > 0$ be the smoothness parameter of $\log(p^*)$: $\|\nabla \log(p^*(\theta)) - \nabla \log(p^*(\theta'))\| \leq \tilde{L}\|\theta - \theta'\|$. Then, Ma et al. (2019) showed the following proposition.

**Proposition 4** (Lemma 9 of Ma et al. (2019) with modification)**.** *Let the initial density $p_0$ satisfy*

$$\sup_{\theta \in \mathbb{R}^p} \frac{p_0(\theta)}{p^*(\theta)} \leq \beta$$

*for $\beta > 0$. Then, for the step size $\eta > 0$ satisfying*

$$\eta = O\left(\min\left\{\frac{1}{\tilde{L}^{3/2}R(\epsilon/2\beta)}, \frac{1}{\tilde{L}\tilde{d}}, \frac{1}{\tilde{L}^{4/3}R(\epsilon/2\beta)^{2/3}\tilde{d}^{1/3}}\right\}\right),$$

*and the number $k$ of iterations satisfying*

$$
\begin{aligned}
k &= \Omega\left(\frac{1}{c_{\mathrm{LS}}^2\eta}\log\left(\frac{2\beta}{\epsilon}\right)\right) \\
&\geq \Omega\left(\frac{1}{c_{\mathrm{LS}}^2}\log\left(\frac{2\beta}{\epsilon}\right)\max\{\tilde{L}^{3/2}R(\epsilon/2\beta), \tilde{L}\tilde{d}, \tilde{L}^{4/3}R(\epsilon/2\beta)^{2/3}\tilde{d}^{1/3}\}\right),
\end{aligned}
\tag{9}
$$

*it holds that*

$$\|p^* - p_k\|_{\mathrm{TV}} \leq \epsilon.$$

We apply this proposition to our setting. From now on, we assume Assumptions 1 and 2 hold. Then, it can be easily seen that

$$\tilde{L} = \frac{LB_2 + 2\lambda_1}{\lambda_2}.$$

If we employ $p_0(\theta) \propto \exp(-\frac{\lambda_1}{\lambda_2}\|\theta\|^2)$ (a Gaussian measure), then we may set $\beta \leq \exp(2B_2/\lambda_2)$ because

$$
\begin{aligned}
\frac{p_0(\theta)}{p_*(\theta)} &= \left(\frac{2\lambda_1/\lambda_2}{2\pi}\right)^{p/2}\frac{\int \exp\left(-\frac{1}{\lambda_2 n}\sum_{i=1}^n h_i(\theta')g_i^{(\tilde{n}T)} - \frac{\lambda_1}{\lambda_2}\|\theta'\|^2\right)\mathrm{d}\theta'}{\exp\left(-\frac{1}{\lambda_2 n}\sum_{i=1}^n h_i(\theta)g_i^{(\tilde{n}T)}\right)} \\
&\leq \left(\frac{2\lambda_1/\lambda_2}{2\pi}\right)^{p/2}\exp(2B_2/\lambda_2)\int \exp\left(-\frac{\lambda_1}{\lambda_2}\|\theta'\|^2\right)\mathrm{d}\theta' \\
&= \exp(2B_2/\lambda_2)\left(\frac{2\lambda_1/\lambda_2}{2\pi}\right)^{p/2}\int \exp\left(-\frac{\lambda_1}{\lambda_2}\|\theta'\|^2\right)\mathrm{d}\theta' \\
&= \exp(2B_2/\lambda_2).
\end{aligned}
\tag{10}
$$

Moreover, the proof of Lemma 16 of Ma et al. (2019) showed that we may set

$$R(s) = \sqrt{2\log\left(\frac{\tilde{d}}{s}\right)\frac{\tilde{d}}{c_{\mathrm{LS}}}} + \sqrt{\mathbb{E}_{p^*}[\|\theta\|^2]}.$$

The second term in the right hand side can be evaluated as

$$
\begin{aligned}
\mathbb{E}_{p^*}[\|\theta\|^2] &= \int \|\theta\|^2 p^*(\theta)\mathrm{d}\theta = \int \|\theta\|^2 p_0(\theta)\frac{p^*(\theta)}{p_0(\theta)}\mathrm{d}\theta \\
&\leq \exp(2B_2/\lambda_2)\int \|\theta\|^2 p_0(\theta)\mathrm{d}\theta \\
&= \exp(2B_2/\lambda_2)\frac{\lambda_2}{2\lambda_1}\tilde{d},
\end{aligned}
$$

where the inequality in the second line can be shown in the same manner as Eq. (10). Then, we obtain

$$R(s) = \sqrt{2\log\left(\frac{\tilde{d}}{s}\right)\frac{\tilde{d}}{c_{\mathrm{LS}}}} + \sqrt{\exp\left(\frac{2B_2}{\lambda_2}\right)\frac{\lambda_2}{2\lambda_1}\tilde{d}}.$$

Therefore, substituting these evaluations into Eq. (9) yields that

$$
\begin{aligned}
k \geq &\Omega\left(\frac{1}{c_{\mathrm{LS}}^2}\left(\frac{2B_2}{\lambda_2} + \log\left(\frac{1}{\epsilon}\right)\right) \times \right. \\
&\left. \max\left\{\tilde{L}^{3/2}\sqrt{\left(\log\left(\frac{2\tilde{d}}{\epsilon}\right) + \frac{2B_2}{\lambda_2}\right)\frac{\tilde{d}}{c_{\mathrm{LS}}}}, \tilde{L}\tilde{d}, \tilde{L}^{4/3}\left(\log\left(\frac{2\tilde{d}}{\epsilon}\right) + \frac{2B_2}{\lambda_2}\right)^{1/3}\frac{\tilde{d}^{2/3}}{c_{\mathrm{LS}}^{1/3}}\right\}\right)
\end{aligned}
$$

---

**Algorithm 3** Definition of $\tilde{g}^{(t)}$

1: $\tilde{p}^{(\tilde{n}T)} \leftarrow p[g^{(\tilde{n}T)}]$

2: $\tilde{g}_{i_{\tilde{n}T}}^{(\tilde{n}T+1)} \leftarrow \underset{\tilde{g}_{i_{\tilde{n}T}}}{\operatorname{argmax}} -l_{i_{\tilde{n}T}}^*(\tilde{g}_{i_{\tilde{n}T}}) + \int h_{i_{\tilde{n}T}}(\theta)\tilde{p}^{(\tilde{n}T)}(\theta)\mathrm{d}\theta(\tilde{g}_{i_{\tilde{n}T}} - g_{i_{\tilde{n}T}}^{(\tilde{n}T)}) - \frac{1}{2n\lambda_2}|\tilde{g}_{i_{\tilde{n}T}} - g_{i_{\tilde{n}T}}^{(\tilde{n}T)}|^2$

3: **for** $t = \tilde{n}T+1, \ldots, \tilde{n}(T+1)-1$ **do**

4: $\quad \tilde{p}^{(t)} \leftarrow p[\tilde{g}^{(t)}]$

5: $\quad \tilde{g}_{i_t}^{(t+1)} \leftarrow \underset{\tilde{g}_{i_t}}{\operatorname{argmax}} -l_{i_t}^*(\tilde{g}_{i_t}) + \int h_{i_t}(\theta)\tilde{p}^{(t)}(\theta)\mathrm{d}\theta(\tilde{g}_{i_t} - \tilde{g}_{i_t}^{(t)}) - \frac{1}{2n\lambda_2}|\tilde{g}_{i_t} - \tilde{g}_{i_t}^{(t)}|^2$

6: **end for**

---

is sufficient to ensure $\|p^* - p_k\|_{\mathrm{TV}} \le \epsilon$, where $c_{\mathrm{LS}} = \frac{2\lambda_1}{\lambda_2}\exp(-4B_2/\lambda_2)$ and $\tilde{L} = \frac{LB_2+2\lambda_1}{\lambda_2}$. We can see that the right hand side can be bounded by

$$O\left[\frac{1}{c_{\mathrm{LS}}^{5/2}}\left(\frac{B_2(1+L)+\lambda_1}{\lambda_2} + \log(1/\epsilon)\right)^{3/2}\tilde{d}\right],$$

which gives the assertion of Proposition 2.

## C    PROOF OF THEOREM 1

In this section, we give the convergence rate analysis of our algorithm. We first show the convergence rate Algorithm 1 (the continuous limit version), then we prove Theorem 1 by bounding the discretization error. Before we begin the proofs, we need some auxiliary variables that interpolate between the continuous limit solution and the discretized one.

The key ingredient of our proof is to show the discrepancy between the solution with particle discretization and the one with the continuous limit is sufficiently small throughout each inner loop from $t = \tilde{n}T$ to $t = \tilde{n}(T+1)$. To rigorously prove this intuition, we introduce $\tilde{g}^{(t)}$ which follows the continuous limit update but is initialized from the discretized one $g^{(\tilde{n}T)}$ at the every starting point of the inner loop (see its definition in Algorithm 3). To fill in the gap between the continuous limit $\tilde{g}^{(t)}$ and the discretized version $g^{(t)}$, we also define $\bar{g}^{(t)}$ ($t = \tilde{n}T, \ldots, T(\tilde{n}+1)$) which lies in between (see Algorithm 4). Basically, $\bar{g}^{(t)}$ and $\tilde{g}^{(t)}$ are updated by the same rule, but are initialized in different ways. Indeed, the primal variables corresponding to these two sequences are updated as $p[\tilde{g}^{(t+1)}](\theta) \propto p[\tilde{g}^{(t)}](\theta) \times \exp\left(-\frac{1}{n\lambda_2}h_{i_t}(\theta)\delta\tilde{g}^{(t)}\right)$ and $\bar{p}^{(t+1)} \propto \bar{p}^{(t)}(\theta) \times \exp\left(-\frac{1}{n\lambda_2}h_{i_t}(\theta)\delta\bar{g}^{(t)}\right)$, respectively. However, different initial primal variables $\tilde{p}^{(nT)} = p[g^{(nT)}]$ and $\bar{p}^{(nT)} = p^{(nT)}$ are employed for the two schemes (remember that $p^{(nT)}$ is an approximation of $p[g^{(nT)}]$ by the sampling scheme). We denote difference of the updated coordinates before and after updates as

$$\delta g^{(t)} \overset{\mathrm{def}}{=} g_{i_t}^{(t+1)} - g_{i_t}^{(t)}, \; \delta\tilde{g}^{(t+1)} \overset{\mathrm{def}}{=} \tilde{g}^{(t+1)} - \tilde{g}^{(t)}, \; \delta\bar{g}^{(t)} = \bar{g}^{(t+1)} - \bar{g}^{(t)} \; (t = \tilde{n}T, \ldots, T(\tilde{n}+1)-1),$$

respectively.

At the same time, we define some variants of $r_j^{(t)}$, in order to fill the gap more finely between $\bar{g}^{(t)}$ and $g^{(t)}$. We define $\bar{r}_j^{(t)}$ as a direct discretization of $\bar{p}^{(t)}$ using the sampling point $\theta_j$ (see Algorithm 5). Let $\hat{r}_j^{(t)}$ be a numerator of $\bar{r}_j^{(t)}$ (see Algorithm 6). We expect that $\frac{\hat{r}_j^{(t)}}{\sum_{j=1}^M \hat{r}_{j'}^{(t)}}$ plays a role of interpolation between $\bar{p}^{(t)}$ and $\frac{r_j^{(t)}}{\sum_{j=1}^M r_{j'}^{(t)}}$.

In the convergence rate analysis, the smoothness of the dual objective plays an important role. The following lemma shows the smoothness of the dual of the regularization term.

---

**Algorithm 4** Definition of $\bar{g}^{(t)}$

---

1: $\bar{p}^{(\tilde{n}T)} \leftarrow p^{(\tilde{n}T)}$

2: $\bar{g}_{i_{\tilde{n}T}}^{(\tilde{n}T+1)} \leftarrow \underset{\bar{g}_{i_{\tilde{n}T}}}{\operatorname{argmax}} -l_{i_{\tilde{n}T}}^*(\bar{g}_{i_{\tilde{n}T}}) + \int h_{i_{\tilde{n}T}}(\theta)\bar{p}^{(\tilde{n}T)}(\theta)\mathrm{d}\theta(\bar{g}_{i_{\tilde{n}T}} - g_{i_{\tilde{n}T}}^{(\tilde{n}T)}) - \frac{1}{2n\lambda_2}|\bar{g}_{i_{\tilde{n}T}} - g_{i_{\tilde{n}T}}^{(\tilde{n}T)}|^2$

3: **for** $t = \tilde{n}T+1, \ldots, \tilde{n}(T+1) - 1$ **do**

4: $\quad \bar{p}^{(t)}(\theta) \leftarrow \dfrac{p^{(\tilde{n}T)}(\theta) \exp\left(-\frac{1}{n\lambda_2} \sum_{\tau=\tilde{n}T}^{t-1} h_{i_\tau}(\theta_j)\delta\bar{g}_{i_\tau}^{(\tau)}\right)}{\int p^{(\tilde{n}T)}(\theta') \exp\left(-\frac{1}{n\lambda_2} \sum_{\tau=\tilde{n}T}^{t-1} h_{i_\tau}(\theta')\delta\bar{g}_{i_\tau}^{(\tau)}\right)\mathrm{d}\theta'}$

5: $\quad \bar{g}_{i_t}^{(t+1)} \leftarrow \underset{\bar{g}_{i_t}}{\operatorname{argmax}} -l_{i_t}^*(\bar{g}_{i_t}) + \int h_{i_t}(\theta)\bar{p}^{(t)}(\theta)\mathrm{d}\theta(\bar{g}_{i_t} - \bar{g}_{i_t}^{(t)}) - \frac{1}{2n\lambda_2}|\bar{g}_{i_t} - \bar{g}_{i_t}^{(t)}|^2$

6: **end for**

---

**Algorithm 5** Definition of $\bar{r}_j^{(t)}$

---

1: $\bar{r}_j^{(\tilde{n}T)} \leftarrow \frac{1}{M} \quad (j = 1, \ldots, M)$

2: **if** $\tilde{n} > 1$ **then**

3: $\quad$ **for** $t = \tilde{n}T, \tilde{n}T+1, \ldots, \tilde{n}(T+1) - 2$ **do**

4: $\quad\quad \bar{r}_j^{(t+1)} \leftarrow \dfrac{\frac{1}{M} \exp\left(-\frac{1}{n\lambda_2} \sum_{\tau=\tilde{n}T}^{t} h_{i_\tau}(\theta_j)\delta\bar{g}_{i_\tau}^{(\tau)}\right)}{\int p^{(nT)}(\theta') \exp\left(-\frac{1}{n\lambda_2} \sum_{\tau=\tilde{n}T}^{t} h_{i_\tau}(\theta')\delta\bar{g}_{i_\tau}^{(\tau)}\right)\mathrm{d}\theta'}$

5: $\quad$ **end for**

6: **end if**

---

**Algorithm 6** Definition of $\hat{r}_j^{(t)}$

---

1: $\hat{r}_j^{(\tilde{n}T)} \leftarrow \frac{1}{M} \quad (j = 1, \ldots, M)$

2: **if** $\tilde{n} > 1$ **then**

3: $\quad$ **for** $t = \tilde{n}T, \tilde{n}T+1, \ldots, \tilde{n}(T+1) - 2$ **do**

4: $\quad\quad \hat{r}_j^{(t+1)} \leftarrow \dfrac{1}{M} \exp\left(-\dfrac{1}{n\lambda_2} \sum_{\tau=\tilde{n}T}^{t} h_{i_\tau}(\theta_j)\delta\bar{g}_{i_\tau}^{(\tau)}\right)$

5: $\quad$ **end for**

6: **end if**

---

**Lemma 4.** *The first and second order derivatives of the second term of the dual objective $D(g)$ can be evaluated as*

$$\left[\nabla_g \lambda_2 \log\left[\int q[g](\theta)\mathrm{d}\theta\right]\right]_i = -\frac{1}{n}\int h_i(\theta)p[g](\theta)\mathrm{d}\theta, \tag{11}$$

$$\left[\nabla_g^2 \lambda_2 \log\left[\int q[g](\theta)\mathrm{d}\theta\right]\right]_{i,j} =$$

$$\frac{1}{n^2\lambda_2}\left\{\int h_i(\theta)h_j(\theta)p[g](\theta)\mathrm{d}\theta - \int h_i(\theta)p[g](\theta)\mathrm{d}\theta \int h_j(\theta)p[g](\theta)\mathrm{d}\theta\right\}. \tag{12}$$

*In particular, the dual of the regularization term can be bounded by a quadratic function as follows:*
*(a) $L_1$-norm bound,*

$$\lambda_2 \log \left[ \int q[g + \delta g](\theta) \mathrm{d}\theta \right] \leq \lambda_2 \log \left[ \int q[g](\theta) \mathrm{d}\theta \right] - \frac{1}{n} \sum_{i=1}^{n} \int h_i(\theta) p[g](\theta) \mathrm{d}\theta \delta g_i + \frac{1}{2n^2 \lambda_2} \|\delta g\|_1^2, \tag{13}$$

*and (b) $L_2$-norm bound,*

$$\lambda_2 \log \left[ \int q[g + \delta g](\theta) \mathrm{d}\theta \right] \leq \lambda_2 \log \left[ \int q[g](\theta) \mathrm{d}\theta \right] - \frac{1}{n} \sum_{i=1}^{n} \int h_i(\theta) p[g](\theta) \mathrm{d}\theta \delta g_i + \frac{1}{2n \lambda_2} \|\delta g\|_2^2. \tag{14}$$

The $L_1$-norm bound is useful to derive the dual coordinate ascent method (Algorithms 1 and 2). On the other hand, the $L_2$-norm bound is also useful in the following convergence rate analysis.

*Proof.* First of all, it is easy to see that $\lambda_2 \log \left[ \int q[g](\theta) \mathrm{d}\theta \right]$ is two times differentiable by $g_i$. Since a direct calculation yields

$$\frac{\partial}{\partial g_i} \int q[g](\theta) \mathrm{d}\theta = -\frac{1}{n \lambda_2} \int h_i(\theta) \exp \left( \frac{-\frac{1}{n} \sum_{i=1}^{n} h_i(\theta)(g_i) - \lambda_1 \|\theta\|_2^2}{\lambda_2} \right) \mathrm{d}\theta, \tag{15}$$

we immediately obtain Eq. (13):

$$\frac{\partial}{\partial g_i} \lambda_2 \log \left[ \int q[g](\theta) \mathrm{d}\theta \right] = \frac{-\frac{1}{n} \int h_i(\theta) q[g](\theta) \mathrm{d}\theta}{\int q[g](\theta) \mathrm{d}\theta'} = -\frac{1}{n} \int h_i(\theta) p[g](\theta) \mathrm{d}\theta.$$

By differentiating the right hand side by $g_j$ again and using Eq. (15), we get the desired equality (12) for the Hessian.

Note that the right hand side of the equation (12) for the Hessian evaluation is given by the covariance between $h_i$ and $h_j$ with respect to the probability distribution $p[g](\theta)$. Hence, each element is bounded by $\frac{1}{n^2 \lambda_2}$ due to the boundedness assumption of $h_i$ (Assumption (A2)), which yields Eq. (13). As for the $L_2$-norm bound (14), it is easily obtained by the relation $\|\delta g\|_1 \leq \sqrt{n} \|\delta g\|_2$. □

## C.1 CONVERGENCE OF THE IDEAL UPDATE $\check{g}^{(t)}$ (ALGORITHM 1)

First, we prove the convergence of the ideal update $\check{g}^{(t)}$. Convergence analysis in the continuous limit is the basis for that of discretized updates; more concretely, we can use results and proofs for continuous limit in this section as a reference to derive Lemma 8, which is one of the key lemmas for Theorem 1 (convergent proof for discretized updates).

**Lemma 5.** *Suppose that $\ell_i$ is $1/\gamma$-smooth for all $i \in \{1 \ldots, n\}$ (Assumption (A1)). Consider the update of $\check{g}^{(t)}$ given in Algorithm 1 (continuous limit update), then, for any iteration $t$ and any $s \in [0, 1]$, it holds that*

$$\mathbb{E}_{i_t} \left[ D(\check{g}^{(t+1)}) - D(\check{g}^{(t)}) \right] \geq \frac{s}{n} (P(\check{p}^{(t)}) - D(\check{g}^{(t)})) - \left( \frac{s}{n} \right)^2 \frac{G^{(t)}}{2\lambda_2}, \tag{16}$$

*where $\check{p}^{(t)} = p[\check{g}^{(t)}]$,*

$$G^{(t)} \stackrel{\text{def}}{=} \left( 1 - \frac{\gamma(1-s)\lambda_2 n}{s} \right) \mathbb{E}_{i_t} \left[ (u_{i_t}^{(t)} - \check{g}_{i_t}^{(t)})^2 \right] \tag{17}$$

*and $u_i^{(t)} \stackrel{\text{def}}{=} \ell_i' \left( \int \check{p}^{(t)}(\theta) h_i(\theta) \mathrm{d}\theta \right)$. Here, the expectation was taken with respect to the choice of the coordinate $i_t$. Accordingly, we also have the same bound for the ideal update in Algorithm 2 ($g^{(\tilde{n}T)}$ and $\tilde{g}^{(t)}$ ($t = \tilde{n}T + 1, \ldots, \tilde{n}(T+1)$)), considering the definition of $\tilde{g}^{(t)}$.*

*Proof.* Since we will focus on the update of one step, $i_t$ is omitted as $i$ in the following. At each update, the improvement in the dual objective can be written as

$$n[D(\check{g}^{(t+1)}) - D(\check{g}^{(t)})]$$

$$
\begin{aligned}
= &- \ell_i^*(\check{g}_i^{(t+1)}) + \ell_i^*(\check{g}_i^{(t)}) - \lambda_2 \log \left[ \int q[\check{g}^{(t+1)}](\theta) \mathrm{d}\theta \right] + \lambda_2 \log \left[ \int q[\check{g}^{(t)}](\theta) \mathrm{d}\theta \right] \\
\geq &- \ell_i^*(\check{g}_i^{(t+1)}) + \int h_i(\theta) \check{p}^{(t)}(\theta) \mathrm{d}\theta \, (\check{g}_i^{(t+1)} - g_i^{(t)}) - \frac{1}{2n\lambda_2} |g_i - \check{g}_i^{(t)}|^2 + \ell_i^*(\check{g}_i^{(t)}) \\
\geq &- \ell_i^*(\check{g}_i^{(t)} + s(u_i^{(t)} - \check{g}_i^{(t)})) + s \int h(\theta) \check{p}^{(t)}(\theta) \mathrm{d}\theta \, (u_i^{(t)} - \check{g}_i^{(t)}) - \frac{s^2}{2n\lambda_2} |u_i^{(t)} - \check{g}_i^{(t)}|^2 + \ell_i^*(\check{g}_i^{(t)}),
\end{aligned}
\tag{18}
$$

where we used Eq. (13) in Lemma 4 for the first inequality, and the second inequality follows from the definition of $\check{g}^{(t+1)}$.

Since $\ell_i^*$ is $\gamma$-strongly convex ($\gamma = 0$ is also allowed), it holds that

$$
\ell_i^*(\check{g}_i^{(t)} + s(u_i^{(t)} - \check{g}_i^{(t)})) \leq s\ell_i^*(u_i^{(t)}) + (1-s)\ell_i^*(\check{g}_i^{(t)}) - \frac{\gamma}{2}s(1-s)|u_i^{(t)} - \check{g}_i^{(t)}|^2.
\tag{19}
$$

From the property of the Legendre transform, $u_i^{(t)} = \ell_i' \left( \int \check{p}^{(t)}(\theta) h_i(\theta) \mathrm{d}\theta \right)$ implies that

$$
\ell_i \left( \int \check{p}^{(t)}(\theta) h_i(\theta) \mathrm{d}\theta \right) + \ell_i^*(u_i^{(t)}) = \int h_i(\theta) \check{p}^{(t)}(\theta) \mathrm{d}\theta \, u_i^{(t)}.
\tag{20}
$$

Similarly, we also have the following relation from the duality of the regularization term:

$$
\begin{aligned}
&\lambda_1 \int \check{p}^{(t)}(\theta) \|\theta\|_2^2 \mathrm{d}\theta + \lambda_2 \int \check{p}^{(t)}(\theta) \log p^{(t)}(\theta) \mathrm{d}\theta + \lambda_2 \log \left[ \int q[\check{g}^{(t)}](\theta) \mathrm{d}\theta \right] \\
&= -\frac{1}{n} \sum_{i=1}^n \int h_i(\theta) \check{p}^{(t)}(\theta) \mathrm{d}\theta \check{g}_i^{(t)}.
\end{aligned}
\tag{21}
$$

Combining Eqs. (18), (19) and (20), we have that

$$
\begin{aligned}
&n[D(\check{g}^{(t+1)}) - D(\check{g}^{(t)})] \\
\geq &- s \left( \ell_i^*(u_i^{(t)}) - \int h_i(\theta) \check{p}^{(t)}(\theta) \mathrm{d}\theta \, u_i^{(t)} \right) + \frac{s}{2} \left( \gamma(1-s) - \frac{s}{n\lambda_2} \right) |u_i^{(t)} - \check{g}_i^{(t)}|^2 \\
&+ s \left( \ell_i^*(\check{g}_i^{(t)}) - \check{g}_i^{(t)} \int h_i(\theta) \check{p}^{(t)}(\theta) \mathrm{d}\theta \right) \\
= &s \left[ \ell_i \left( \int \check{p}^{(t)}(\theta) h_i(\theta) \mathrm{d}\theta \right) + \ell_i^*(\check{g}_i^{(t)}) - \int h_i(\theta) \check{p}^{(t)}(\theta) \mathrm{d}\theta \check{g}_i^{(t)} \right] \\
&+ \frac{s}{2} \left( \gamma(1-s) - \frac{s}{n\lambda_2} \right) |u_i^{(t)} - \check{g}_i^{(t)}|^2.
\end{aligned}
\tag{22}
$$

By taking expectation of both sides of (22) with respect to $i$, it follows that

$$
\begin{aligned}
&\mathbb{E} \left[ n[D(\check{g}^{(t+1)}) - D(\check{g}^{(t)})] \right] \\
\geq &s \left[ \frac{1}{n} \sum_{i=1}^n \ell_i \left( \int \check{p}^{(t)}(\theta) h_i(\theta) \mathrm{d}\theta \right) + \frac{1}{n} \sum_{i=1}^n \ell_i^*(\check{g}_i^{(t)}) - \frac{1}{n} \sum_{i=1}^n \int h_i(\theta) \check{p}^{(t)}(\theta) \mathrm{d}\theta \check{g}_i^{(t)} \right] \\
&+ \frac{s}{2n} \left( \gamma(1-s) - \frac{s}{n\lambda_2} \right) \sum_{i=1}^n |u_i^{(t)} - \check{g}_i^{(t)}|^2 \\
= &s \left[ \frac{1}{n} \sum_{i=1}^n \ell_i \left( \int \check{p}^{(t)}(\theta) h_i(\theta) \mathrm{d}\theta \right) + \frac{1}{n} \sum_{i=1}^n \ell_i^*(\check{g}_i^{(t)}) \right. \\
&\left. + \lambda_1 \int \check{p}^{(t)}(\theta) \|\theta\|_2^2 \mathrm{d}\theta + \lambda_2 \int \check{p}^{(t)}(\theta) \log \check{p}^{(t)}(\theta) \mathrm{d}\theta + \lambda_2 \log \left( \int q[\check{g}^{(t)}](\theta) \mathrm{d}\theta \right) \right] \\
&+ \frac{s}{2n} \left( \gamma(1-s) - \frac{s}{n\lambda_2} \right) \sum_{i=1}^n |u_i^{(t)} - \check{g}_i^{(t)}|^2
\end{aligned}
$$

$$= s(P(\check{p}^{(t)}) - D(\check{g}^{(t)})) - \frac{s^2}{n} \frac{G^{(t)}}{2\lambda_2}, \tag{23}$$

where we used Eq. (21) for the first equality. Dividing the both sides of Eq. (23) by $n$, we obtain the desired inequality (16). $\qquad\square$

**Lemma 6.** $\check{g}_i^{(t)}$ in Algorithm 1 is uniformly bounded by $B_2$, that is,

$$|\check{g}_i^{(t)}| \le B_2 \quad (\forall t = 0, 1, \dots, n T_{\text{end}}, \ \forall i \in \{1, \dots, n\}).$$

In the same way, $g_i^{(t)}$, $\tilde{g}_i^{(t)}$ and $\bar{g}_i^{(t)}$ in Algorithm 2 satisfy

$$|g_i^{(t)}|, \ |\tilde{g}_i^{(t)}|, \ |\bar{g}_i^{(t)}| \le B_2 \quad (\forall t = 0, 1, \dots, n T_{\text{end}}, \ \forall i \in \{1, \dots, n\}).$$

*Proof.* Let $p$ be an arbitrary probability distribution. Let us consider the update

$$z = \operatorname*{argmax}_{z' \in \mathbb{R}} \left\{ -\ell_i^*(z') + (z' - y) \int h_i(\theta) p(\theta) \mathrm{d}\theta \right\}, \tag{24}$$

for an arbitrary $i \in \{1, \dots, n\}$. Since $h_i$ is bounded by 1 (Assumption (A2)), the sub-differential of $\ell_i^*$ at $z$ satisfies that

$$\partial \ell_i^*(z) \cap [-1, 1] \ne \emptyset.$$

By the assumption (A3), i.e.,

$$B_2 = \sup_{i,x} \left\{ |\ell_i{}'(x)| \ \big| \ |x| \le 1 \right\} = \sup_{i,y} \left\{ |y| \ \big| \ \partial \ell_i^*(y) \cap [-1, 1] \ne \emptyset \right\} < \infty,$$

we have that $z$ in Eq. (24) is bounded by $B_2$.

Next, we consider its *proximal-version* update:

$$z = \operatorname*{argmax}_{z' \in \mathbb{R}} \left\{ -\ell_i^*(z') + \int h_i(\theta) p(\theta) \mathrm{d}\theta (z' - y) - \frac{1}{2n\lambda_2} |z' - y|^2 \right\}. \tag{25}$$

In this proximal-version update, we added the term $-\frac{1}{2n\lambda_2}(z - y)^2$ whose maximizer with respect to $z$ is given by $z = y$. Since the maximum point of the sum of two concave functions is located between the maximizers of them, $z$ in Eq. (25) satisfies that

$$|z| \le \max \left\{ B_2, |y| \right\}.$$

From this and the initialization $|\check{g}_i^{(0)}| \le B_2$, we have that $|\check{g}_i^{(t)}|$ in Algorithm 1 is uniformly bounded by $B_2$ by induction.

In the same way, $g_i^{(t)}$, $\tilde{g}_i^{(t)}$ and $\bar{g}_i^{(t)}$ in Algorithm 2 minimize the sum of the average of $h_i(\theta_j)$ and quadratic term as well. Note that the absolute value of the the average of $h_i(\theta_j)$ is bounded with 1, regardless of methods of averaging, that is, weighted particle for $g_i^{(t)}$ and probability distribution for $\tilde{g}_i^{(t)}$ and $\bar{g}_i^{(t)}$. Thus, $g_i^{(t)}$, $\tilde{g}_i^{(t)}$ and $\bar{g}_i^{(t)}$ are uniformly bounded with $B_2$, as well as $\check{g}^{(t)}i$ in Algorithm 1. $\qquad\square$

**Remark 4.** *In the Section C.2, we sometimes face the situation where we should upper-bound increment of the coordinates $\delta g^{(t)}$, $\delta \tilde{g}^{(t)}$ and $\delta \bar{g}^{(t)}$.*

*Lemma 6 directly implies that $\delta g^{(t)}$, $\delta \tilde{g}^{(t)}$ and $\delta \bar{g}^{(t)}$ are all bounded by $2B_2$ for all $t = 0, 1, \dots, \tilde{n} T_{\text{end}}$. Thus, for future convenience, we define*

$$C_1 \overset{\text{def}}{=} \exp \left( \frac{2\kappa B_2}{\lambda_2} \right)$$

*with which $\exp \left( \frac{2\kappa |\delta g^{(t)}|}{\lambda_2} \right)$, $\exp \left( \frac{2\kappa |\delta \tilde{g}^{(t)}|}{\lambda_2} \right)$, $\exp \left( \frac{2\kappa |\delta \bar{g}^{(t)}|}{\lambda_2} \right) \le C_1$ hold, where $\kappa$ is defined as $\kappa = \frac{\tilde{n}}{n}$.*

*Note that, however, we could have much tighter evaluations of $\delta g^{(t)}$, $\delta \tilde{g}^{(t)}$ and $\delta \bar{g}^{(t)}$, which are dependent of residual $D(g^*) - D(g^{(t)})$ or duality gap $P[p(g^{(t)})] - D(g^{(t)})$, since the dual problem is strongly convex due to the smoothness of primal problem.*

**Lemma 7.** *Under Assumptions (A2) and (A3), it holds that*

$$P(p[\check{g}^{(0)}]) - D(\check{g}^{(0)}) \le B_1 + B_2,$$

*for Algorithm 1 and*

$$P(p[g^{(0)}]) - D(g^{(0)}) \le B_1 + B_2,$$

*in Algorithm 2.*

*Proof.* By Assumptions (A2) and (A3), we can see that $\left| \ell_i \left( \int p^{(0)}(\theta) h_i(\theta) \mathrm{d}\theta \right) - \inf_x \ell_i(x) \right|$ is bounded by $B_1$. Since $\inf_x \ell_i(x) > -\inf$ (Assumption (A3)), we also have $\ell_i^*(0) = \sup_x \{0 \cdot x - \ell_i(x)\} = -\inf_x \ell_i(x) < \infty$. In Algorithm 1, every initial coordinate $\check{g}_i^{(0)}$ is bounded by $B_2$. Also, $g_i^{(0)}$ in Algorithm 2 is similarly bounded by $B_2$. Therefore, in the following, $g$ will be used without distinguishing between $g$ and $\check{g}$. Moreover, in a similar manner to Eq. (21) in Lemma 5, we have that

$$\lambda_1 \int p[g^{(t)}](\theta) ||\theta||_2^2 \mathrm{d}\theta + \lambda_2 \int p[g^{(t)}](\theta) \log p[g^{(t)}](\theta) \mathrm{d}\theta + \lambda_2 \log \left[ \int q[g^{(t)}](\theta) \mathrm{d}\theta \right]$$

$$= -\frac{1}{n} \sum_{i=1}^n \int h_i(\theta) p[g^{(t)}](\theta) \mathrm{d}\theta g_i^{(t)},$$

from the duality.

Combining these observations, we have that

$$P(p[g^{(0)}]) - D(g^{(0)})$$

$$= \frac{1}{n} \sum_{i=1}^n \ell_i \left( \int p^{(0)}(\theta) h_i(\theta) \mathrm{d}\theta \right) + \frac{1}{n} \sum_{i=1}^n \ell_i^*(g_i^{(0)}) + \lambda_1 \int p^{(0)}(\theta) ||\theta||_2^2 \mathrm{d}\theta$$

$$\quad + \lambda_2 \int p^{(0)}(\theta) \log p^{(0)}(\theta) \mathrm{d}\theta + \lambda_2 \log \left[ \int q[g^{(0)}](\theta) \mathrm{d}\theta \right]$$

$$= \frac{1}{n} \sum_{i=1}^n \ell_i \left( \int p^{(0)}(\theta) h_i(\theta) \mathrm{d}\theta \right) + \frac{1}{n} \sum_{i=1}^n \ell_i^*(g_i^{(0)}) - \frac{1}{n} \sum_{i=1}^n \int h_i(\theta) p^{(0)}(\theta) \mathrm{d}\theta \cdot g_i^{(0)}$$

$$\le \frac{1}{n} \sum_{i=1}^n \left[ \ell_i \left( \int p^{(0)}(\theta) h_i(\theta) \mathrm{d}\theta \right) - \inf_x \ell_i(x) \right] + \frac{1}{n} \sum_{i=1}^n |g_i^{(0)}|$$

$$\le \frac{1}{n} \sum_{i=1}^n B_1 + B_2 = B_1 + B_2,$$

where we used Lemma 6 in the last inequality. $\qquad\square$

**Theorem 2.** *Let $\epsilon_P(> 0)$ be a desired accuracy. To obtain an expected duality gap $\mathbb{E}\left[ P(\check{p}^{(t_{\mathrm{end}})}) - D(\check{g}^{(t_{\mathrm{end}})}) \right] \le \epsilon_P$ in Algorithm 1, it suffices to have the total number $T_{\mathrm{end}}$ of iterations as*

$$t_{\mathrm{end}} \ge \left( n + \frac{1}{\gamma\lambda_2} \right) \log \left( \left( n + \frac{1}{\gamma\lambda_2} \right) \frac{B_1 + B_2}{\epsilon_P} \right).$$

*Furthermore, if we randomly choose $(\check{p}^{(t)}, \check{g}^{(t)})$ from $t = t_{\mathrm{end}} - n + 1, \ldots, t_{\mathrm{end}}$ in Algorithm 1, then*

$$t_{\mathrm{end}} \ge \left( n + \frac{1}{\gamma\lambda_2} \right) \log \left( \left( 1 + \frac{1}{n\gamma\lambda_2} \right) \frac{B_1 + B_2}{\epsilon_P} \right) + n$$

*is sufficient to achieve $\epsilon_P$ duality gap in expectation.*

*Proof.* In Lemma 4, choosing $s = \frac{n\gamma\lambda_2}{1+n\gamma\lambda_2} \in [0, 1]$, then we have that

$$1 - \frac{(1-s)n\gamma\lambda_2}{s} \le 1 - 1 = 0.$$

Hence, we have $G^{(t)} \leq 0$ for all $t$. Thus, Lemma 5 implies that

$$\mathbb{E}\left[D(\check{g}^{(t+1)}) - D(\check{g}^{(t)})\right] \geq \frac{s}{n}\mathbb{E}\left[P(\check{p}^{(t)}) - D(\check{g}^{(t)})\right]. \tag{26}$$

Therefore, we have that

$$\frac{s}{n}\mathbb{E}\left[D(g^*) - D(\check{g}^{(t)})\right] \leq \frac{s}{n}\mathbb{E}\left[P(\check{p}^{(t)}) - D(\check{g}^{(t)})\right] \leq \frac{s}{n}\mathbb{E}\left[D(\check{g}^{(t+1)}) - D(\check{g}^{(t)})\right]$$
$$= \mathbb{E}\left[D(g^*) - D(\check{g}^{(t)})\right] - \mathbb{E}\left[D(g^*) - D(\check{g}^{(t+1)})\right].$$

This implies that

$$\mathbb{E}\left[D(g^*) - D(\check{g}^{(t+1)})\right] \leq \left(1 - \frac{s}{n}\right)\mathbb{E}\left[D(g^*) - D(\check{g}^{(t)})\right]. \tag{27}$$

By applying this inequality repeatedly, we obtain that

$$\mathbb{E}\left[D(g^*) - D(\check{g}^{(t)})\right] \leq \left(1 - \frac{s}{n}\right)^t \mathbb{E}\left[D(g^*) - D(\check{g}^{(0)})\right] \leq (B_1 + B_2)\exp\left(-\frac{st}{n}\right), \tag{28}$$

where the last inequality is obtained by $\mathbb{E}\left[D(g^*) - D(\check{g}^{(0)})\right] \leq \mathbb{E}\left[P(\check{p}^{(0)}) - D(\check{g}^{(0)})\right]$ and Lemma 7.

Moreover, making use of Eq. (26), the duality gap can be bounded as follows:

$$\mathbb{E}\left[P(\check{p}^{(t)}) - D(\check{g}^{(t)})\right] \leq \frac{n}{s}\mathbb{E}\left[D(\check{g}^{(t+1)}) - D(\check{g}^{(t)})\right] \leq \frac{n}{s}\mathbb{E}\left[D(g^*) - D(\check{g}^{(t)})\right]. \tag{29}$$

Hence, combining Eqs. (28) and (29) yields

$$\mathbb{E}\left[P(\check{p}^{(t)}) - D(\check{g}^{(t)})\right] \leq \frac{n(B_1 + B_2)}{s}\exp\left(-\frac{\gamma\lambda_2 t}{1 + n\gamma\lambda_2}\right).$$

Therefore, in order to have an $\epsilon_P$-approximated solution in expectation, it suffices to let

$$t_{\text{end}} \geq \left(n + \frac{1}{\gamma\lambda_2}\right)\log\left(\left(n + \frac{1}{\gamma\lambda_2}\right)\frac{B_1 + B_2}{\epsilon_P}\right).$$

Finally, we show the second part. By summing up Eq. (26) over $t = t_{\text{end}} - n + 1, \ldots, t_{\text{end}}$, we have that

$$\mathbb{E}\left[\frac{1}{n}\sum_{t=t_{\text{end}}-n+1}^{t_{\text{end}}}\left(P(\check{p}^{(t)}) - D(\check{g}^{(t)})\right)\right] \leq \frac{1}{s}\mathbb{E}\left[D(\check{g}^{(t_{\text{end}}+1)}) - D(\check{g}^{(t_{\text{end}}-n)})\right]$$
$$\leq \frac{1}{s}\mathbb{E}\left[D(g^*) - D(\check{g}^{(t_{\text{end}}-n)})\right]$$
$$\leq \frac{B_1 + B_2}{1 + \frac{1}{n\gamma\lambda_2}}\exp\left(-\frac{\gamma\lambda_2(t_{\text{end}} - n)}{1 + n\gamma\lambda_2}\right).$$

Note that although we do not actually calculate $\check{g}^{(t_{\text{end}}+1)}$, we can virtually define $\check{g}^{(t_{\text{end}}+1)}$ as if the algorithm continued.

Therefore, in order to have an $\epsilon_P$-approximated solution, it suffices to let

$$t_{\text{end}} \geq \left(n + \frac{1}{\gamma\lambda_2}\right)\log\left(\left(1 + \frac{1}{n\gamma\lambda_2}\right)\frac{B_1 + B_2}{\epsilon_P}\right) + n,$$

when we randomly choose $(\check{p}^{(t)}, \check{g}^{(t)})$ from $t = t_{\text{end}} - n + 1, \ldots, t_{\text{end}}$. □

## C.2 CONVERGENCE OF THE PARTICLE-VERSION UPDATE $g^{(t)}$ (ALGORITHM 2)

Next, we consider the convergence in Algorithm 2.

In order to prove Theorem 1 and give a convergence proof for Algorithm 2, we adopt the following strategy. First we prove Lemma 8, by rewriting Theorem 2 to allow the discretization error in each

series of inner loops. Then, the assumptions of Theorem 2 is verified, by Theorem 3. There are two major streams leading up to Theorem 3. The first is about the error originated from the sampling accuracy (Langevin algorithms), and the second is to evaluate the error accumulated during the re-weighting stage. The former is considered in Section C.2.1 and the latter is in Section C.2.2, both of which are combined in Section C.2.3 and finally Theorem 3 is proved.

Apart from the sources of the error, we bound the discrepancy of the duality gap between continuous limit and discretized updates $(P(g^{(t)}) - D(g^{(t)})) - (P(\tilde{g}^{(t)}) - D(\tilde{g}^{(t)}))$ with the following two parts,

$$\epsilon_A^{(t)} \overset{\text{def}}{=} \mathbb{E}\left[ D(\tilde{g}^{(t)}) - D(g^{(t)}) \right],$$

$$\epsilon_B^{(t)} \overset{\text{def}}{=} \mathbb{E}\left[ P(p[g^{(t)}]) - P(p[\tilde{g}^{(t)}]) \right],$$

each of which corresponds to the discrepancy in the primal problem and dual problem, respectively. As mentioned above, these two terms are attributed to the two sources, that is, sampling and re-weighting. These variables $\epsilon_A^{(t)}, \epsilon_B^{(t)}$ represent how much the two sequences $g^{(t)}$ and $\tilde{g}^{(t)}$ diverge during the updates of the inner-loop. Intuitively, in each inner-loop, the absolute values of $\epsilon_A^{(t)}$ and $\epsilon_B^{(t)}$ are increasing sequences between $t = \tilde{n}T + 1$ and $\tilde{n}(T + 1)$. Then, they are "reset" at $t = \tilde{n}(T + 1)$ according to the initialization rule of $\tilde{g}^{(\tilde{n}(T+1)+1)}$ and again will increase from $t = \tilde{n}(T + 1) + 1$ to $\tilde{n}(T + 2)$.

The following lemma states that under the assumption about $\epsilon_A^{(t)}, \epsilon_B^{(t)}$, Algorithm 2 achieves linear convergence with regard to the number of coordinate update. From now on, we fix $s = \frac{n\gamma\lambda_2}{1+n\gamma\lambda_2}$. Also, we define $\tilde{s} = \kappa \cdot \frac{s}{2} = \frac{\tilde{n}\gamma\lambda_2}{1+n\gamma\lambda_2}$, where $\kappa = \frac{\tilde{n}}{n}$.

**Lemma 8.** *Suppose that Assumptions 1 and 2 hold. If $\epsilon_A^{(\tilde{n}T)}$ is smaller than $C_0 e^{-\tilde{s}T}$ for every $T = 1, \ldots, T_{\text{end}}$ with a constant $C_0$ you can arbitrarily take, an expected duality gap $\mathbb{E}\left[ P(p[g^{(nT_{\text{end}})}]) - D(g^{(nT_{\text{end}})}) \right] \le \epsilon_P$ can be achieved by a total number $T_{\text{end}}$ of outer iterations given by*

$$T_{\text{end}} \ge 2\frac{n}{\tilde{n}}\left(1 + \frac{1}{n\gamma\lambda_2}\right)\log\left(\left(n + \frac{1}{\gamma\lambda_2}\right)\frac{(B_1 + B_2 + \frac{C_0}{1-\exp(-\tilde{s})})}{\epsilon_P}\right). \tag{30}$$

*Furthermore, if $\epsilon_A^{(t)}$ and $\epsilon_B^{(t)}$ are uniformly bounded by $\frac{\epsilon_P}{4}$ for all $t = \tilde{n}(T_{\text{end}} - \lceil\frac{1}{\kappa}\rceil) + 1, \ldots, nT_{\text{end}}$ in addition to the assumptions above, then it also suffices to set*

$$T_{\text{end}} \ge 2\frac{n}{\tilde{n}}\left(1 + \frac{1}{n\gamma\lambda_2}\right)\log\left(\left(2 + \frac{2}{n\gamma\lambda_2}\right)\frac{(B_1 + B_2 + \frac{C_0}{1-\exp(-\tilde{s})})}{\epsilon_P}\right) + \left\lceil\frac{1}{\kappa}\right\rceil, \tag{31}$$

*to achieve $\epsilon_P$-duality gap when we randomly choose output $(p^{(t)}, g^{(t)})$ from $t = \tilde{n}(T_{\text{end}} - \lceil\frac{1}{\kappa}\rceil) + 1, \ldots, \tilde{n}T_{\text{end}}$.*

**Remark 5.** *Comparing the speed of convergence with regard to coordinate update $t$, it is $\exp(-st/\tilde{n})$ for Theorem 2 and $\exp(-\tilde{s}T) \propto \exp(-\frac{s}{2}t/\tilde{n})$ for Lemma 8, which can be interpreted as follows. In the descretized version, Lemma 8 allows discretization error to appear, at the cost of a slight decrease in convergence speed. Indeed, you can choose arbitrary $\tilde{s}$ from the open interval $(0, \kappa s)$. Our choice of $\tilde{s} = \frac{\kappa s}{2}$ is just for simplicity. If you choose another $\tilde{s}$ much closer to $\kappa s$, $T_{\text{end}}$ can be smaller.*

**Remark 6.** *If you change the length of inner loop in the middle, the convergence rate after the change is obtained by simply replacing $B_1 + B_2$ with the duality gap at the time of the change.*

*Proof.* Since the update of $\tilde{g}^{(t)}$ is the same as that of $\check{g}^{(t)}$ in Algorithm 1 during the inner loop, it holds that

$$\mathbb{E}\left[ D(g^*) - D(\tilde{g}^{(\tilde{n}T+1)}) \right] \le \left(1 - \frac{s}{n}\right) \mathbb{E}\left[ D(g^*) - D(g^{(\tilde{n}T)}) \right],$$

for $T = 0, 1, \ldots, T_{\text{end}} - 1$. We also have

$$\mathbb{E}\left[ D(g^*) - D(\tilde{g}^{(t+1)}) \right] \le \left(1 - \frac{s}{n}\right) \mathbb{E}\left[ D(g^*) - D(\tilde{g}^{(t)}) \right],$$

for $t = \tilde{n}T + 1, \ldots, \tilde{n}(T+1) - 1$, in the same way as the derivation of Eq. (27).

By applying these inequalities above iteratively, we have that

$$\mathbb{E}\left[D(g^*) - D(\tilde{g}^{(\tilde{n}(T+1))})\right] \le \left(1 - \frac{s}{n}\right)^{\tilde{n}} \mathbb{E}\left[D(g^*) - D(g^{(\tilde{n}T)})\right]$$

$$\le e^{-2\tilde{s}}\mathbb{E}\left[D(g^*) - D(g^{(\tilde{n}T)})\right]. \tag{32}$$

Bounding $D(\tilde{g}^{(\tilde{n}(T+1))})$ by $D(g^{(\tilde{n}(T+1))}) - \epsilon_A^{(\tilde{n}(T+1))}$, we have that

$$\mathbb{E}\left[D(g^*) - D(g^{(\tilde{n}(T+1))})\right] \le e^{-2\tilde{s}}\mathbb{E}\left[D(g^*) - D(g^{(\tilde{n}T)})\right] + \epsilon_A^{(\tilde{n}(T+1))}.$$

Therefore, by the Gronwall's lemma ("vanilla version" of Lemma 18), we obtain that

$$\mathbb{E}\left[D(g^*) - D(g^{(\tilde{n}T)})\right] \le e^{-2\tilde{s}T}\mathbb{E}\left[D(g^*) - D(g^{(0)})\right] + e^{-2\tilde{s}T}\sum_{T'=1}^{T} e^{2\tilde{s}T'}\epsilon_A^{(nT')}.$$

Applying the assumption $\epsilon_A^{(nT)} \le C_0 e^{-\tilde{s}T}$, this implies that

$$\mathbb{E}\left[D(g^*) - D(g^{(\tilde{n}T)})\right] \le e^{-sT}\mathbb{E}\left[D(g^*) - D(g^{(0)})\right] + C_0\frac{e^{-\tilde{s}T} - e^{-2\tilde{s}T}}{1 - e^{-\tilde{s}}}$$

$$\le e^{-\tilde{s}T}\left(\mathbb{E}\left[P(g^{(0)}) - D(g^{(0)})\right] + \frac{C_0}{1 - e^{-\tilde{s}}}\right)$$

$$\le e^{-\tilde{s}T}\left(B_1 + B_2 + \frac{C_0}{1 - e^{-\tilde{s}}}\right), \tag{33}$$

where we used Lemma 7 for the last inequality.

Moreover, the same reasoning as Eq. (29) yields that

$$\mathbb{E}\left[P(p[g^{(\tilde{n}T)}]) - D(g^{(\tilde{n}T)})\right] \le \frac{n}{s}\mathbb{E}\left[D(g^*) - D(g^{(\tilde{n}T)})\right]. \tag{34}$$

Therefore, combining Eqs. (33) and (34), we can bound the duality gap as

$$\mathbb{E}\left[P(p[g^{(\tilde{n}T)}]) - D(g^{(\tilde{n}T)})\right] \le e^{-\tilde{s}T}\frac{n}{s}\left(B_1 + B_2 + \frac{C_0}{1 - e^{-\tilde{s}}}\right).$$

This inequality provides the sufficient number of iterations to attain the desired duality gap. In order to have an $\epsilon_P$-approximated solution, it is sufficient to let

$$T_{\text{end}} \ge 2\frac{n}{\tilde{n}}\left(1 + \frac{1}{n\gamma\lambda_2}\right)\log\left(\left(n + \frac{1}{\gamma\lambda_2}\right)\frac{B_1 + B_2 + \frac{C_0}{1 - \exp(-\tilde{s})}}{\epsilon_P}\right).$$

Next, we show the convergence of the averaging version. Similarly to Theorem 2, we have that for $k = 1, \ldots, \lceil\frac{1}{\kappa}\rceil$,

$$\mathbb{E}\left[\frac{1}{\tilde{n}}\sum_{t=\tilde{n}(T_{\text{end}}-k)+1}^{\tilde{n}(T_{\text{end}}-k+1)}\left(P(p[\tilde{g}^{(t)}]) - D(\tilde{g}^{(t)})\right)\right] \le \frac{1}{s}\mathbb{E}\left[D(g^*) - D(\tilde{g}^{(\tilde{n}(T_{\text{end}}-k)+1)})\right]$$

$$\le \frac{1}{s}e^{-\tilde{s}(T_{\text{end}}-\lceil 1/\kappa\rceil)}\left(B_1 + B_2 + \frac{C_0}{1 - e^{-\tilde{s}}}\right).$$

Note that, if Eq. (31) holds, we have that $\frac{1}{s}e^{-\tilde{s}(T_{\text{end}}-\lceil 1/\kappa\rceil)}\left(B_1 + B_2 + \frac{C_0}{1-e^{-\tilde{s}}}\right) \le \frac{\epsilon_P}{2}$. Hence, if $\epsilon_A^{(t)}$ and $\epsilon_B^{(t)}$ are uniformly bounded by $\frac{\epsilon_P}{4}$ for all $t = \tilde{n}\left(T_{\text{end}} - \lceil 1/\kappa\rceil\right) + 1, \ldots, \tilde{n}T_{\text{end}}$, it holds that

$$\mathbb{E}\left[\frac{1}{\lceil 1/\kappa\rceil\tilde{n}}\sum_{t=n(T_{\text{end}}-\lceil 1/\kappa\rceil)+1}^{nT_{\text{end}}}\left(P(p[g^{(t)}]) - D(g^{(t)})\right)\right]$$

$$\leq \sum_{k=1}^{\lceil 1/\kappa \rceil} \frac{1}{\lceil 1/\kappa \rceil} \mathbb{E}\left[ \frac{1}{\tilde{n}} \sum_{t=\tilde{n}(T_{\text{end}}-k)+1}^{\tilde{n}(T_{\text{end}}-k+1)} \left( P(p[\tilde{g}^{(t)}]) - D(\tilde{g}^{(t)}) \right) \right]$$

$$+ \frac{1}{\lceil 1/\kappa \rceil \tilde{n}} \sum_{t=\tilde{n}(T_{\text{end}}-\lceil 1/\kappa \rceil)+1}^{\tilde{n}T_{\text{end}}} \epsilon_A^{(t)} + \frac{1}{\lceil 1/\kappa \rceil \tilde{n}} \sum_{t=\tilde{n}(T_{\text{end}}-\lceil 1/\kappa \rceil)+1}^{\tilde{n}T_{\text{end}}} \epsilon_B^{(t)}$$

$$\leq \frac{\epsilon_P}{2} + \frac{\epsilon_P}{4} + \frac{\epsilon_P}{4} = \epsilon_P.$$

This concludes the proof. $\qquad\square$

In the following, we derive the sufficient conditions for the assumptions in Lemma 8 by considering the error induced by the sampling stage (Section C.2.1) and the error induced at the re-weighting stage (Section C.2.2), respectively. For that purpose, we first prepare lemmas that will be used in the both proofs.

The following Lemma 9 evaluates how much the difference of two solutions in the previous step incurs difference of them in the next update in Algorithm 2. This will be used to bound the accumulated error through each inner loop.

**Lemma 9.** *For arbitrary $i \in [n]$, consider the following two updates:*

$$z_1 = \underset{z_1' \in \mathbb{R}}{\operatorname{argmax}} \left\{ -\ell_i^*(z_1') + H_1(z_1' - y_1) - \frac{1}{2n\lambda_2}|z_1' - y_1|^2 \right\},$$

$$z_2 = \underset{z_2' \in \mathbb{R}}{\operatorname{argmax}} \left\{ -\ell_i^*(z_2') + H_2(z_2' - y_2) - \frac{1}{2n\lambda_2}|z_2' - y_2|^2 \right\}.$$

*Then, we have that*

$$|z_2 - z_1| \leq \frac{1}{\gamma + \frac{1}{n\lambda_2}} \left( |H_2 - H_1| + \frac{1}{n\lambda_2}|y_2 - y_1| \right).$$

*Proof.* From the definition of $z_1$, it holds that

$$-\ell_i^{*\prime}(z_1) + H_1 - \frac{1}{n\lambda_2}(z_1 - y_1) = 0, \tag{35}$$

and similarly,

$$-\ell_i^{*\prime}(z_2) + H_2 - \frac{1}{n\lambda_2}(z_2 - y_2) = 0,$$

where $\ell_i^{*\prime}(z)$ is an element in the sub-differential $\partial \ell_i^*(z)$. From (35), we have that

$$-\ell_i^{*\prime}(z_1) + H_2 - \frac{1}{n\lambda_2}(z_1 - y_2) = H_2 - H_1 + \frac{1}{n\lambda_2}(y_2 - y_1).$$

Since $\ell_i^*$ is $\gamma$-strongly convex, $-\ell_i^*(z) + H_2(z - y_2) - \frac{1}{2n\lambda_2}|z - y_2|^2$ is $\left( \gamma + \frac{1}{n\lambda_2} \right)$-strongly concave. Thus, we obtain that

$$|z_2 - z_1|$$
$$\leq \frac{1}{\gamma + \frac{1}{n\lambda_2}} \left| \nabla\left( -\ell_i^*(z) + H_2(z - y_2) - \frac{1}{2n\lambda_2}|z - y_2|^2 \right)\bigg|_{z=z_2} \right.$$
$$\left. - \nabla\left( -\ell_i^*(z) + H_2(z - y_2) - \frac{1}{2n\lambda_2}|z - y_2|^2 \right)\bigg|_{z=z_1} \right|$$
$$\leq \frac{1}{\gamma + \frac{1}{n\lambda_2}} \left| H_2 - H_1 + \frac{1}{n\lambda_2}(y_2 - y_1) \right|$$
$$\leq \frac{1}{\gamma + \frac{1}{n\lambda_2}} \left( |H_2 - H_1| + \frac{1}{n\lambda_2}|y_2 - y_1| \right),$$

where $\nabla f(z)|_{z=z'}$ is an element of the sub-differential of $f$ at $z = z'$. This concludes the proof. $\quad\square$

Now we are ready to get into the main proofs which evaluate the discretization error.

### C.2.1 CONVERGENCE OF AUXILIARY VARIABLES $\tilde{g}^{(t)}$ AND $\bar{g}^{(t)}$

We now fix some $T$, and consider the sequances $\{g^{(t)}\}_{t=\tilde{n}T}^{\tilde{n}(T+1)}$, $\{\bar{g}^{(t)}\}_{t=\tilde{n}T+1}^{\tilde{n}(T+1)}$, $\{\tilde{g}^{(t)}\}_{t=\tilde{n}T+1}^{\tilde{n}(T+1)}$.

**Lemma 10.** *Suppose that $\int \left| p^{(\tilde{n}T)}(\theta) - p[g^{(\tilde{n}T)}](\theta) \right| d\theta \leq \epsilon_C^{(\tilde{n}T)}$ holds in Algorithm 2. Under these assumptions, it holds that*

$$|\bar{g}_i^{(t)} - \tilde{g}_i^{(t)}| \leq C_2 \epsilon_C^{(\tilde{n}T)}, \tag{36}$$

*for $t = \tilde{n}T + 1, \ldots, \tilde{n}(T+1)$, where $C_2$ is defined as follows:*

$$C_2 \overset{\text{def}}{=} \frac{2C_1^2}{\gamma + \frac{1}{n\lambda_2}} \exp\left( \frac{\tilde{n}(2C_1^2 + 1)}{n\gamma\lambda_2 + 1} \right).$$

*Proof.* By Lemma 9, we have that

$$
\begin{aligned}
&|\bar{g}_{i_t}^{(t+1)} - \tilde{g}_{i_t}^{(t+1)}| \\
&\leq \frac{1}{\gamma + \frac{1}{n\lambda_2}} \left( \left| \int h_{i_t}(\theta)\bar{p}^{(t)}(\theta)d\theta - \int h_{i_t}(\theta)p[\tilde{g}^{(t)}](\theta)d\theta \right| + \frac{1}{n\lambda_2}|\bar{g}_{i_t}^{(t)} - \tilde{g}_{i_t}^{(t)}| \right) \\
&\leq \frac{1}{\gamma + \frac{1}{n\lambda_2}} \left( \int \left| \bar{p}^{(t)}(\theta)d\theta - p[\tilde{g}^{(t)}](\theta) \right| d\theta + \frac{1}{n\lambda_2}|\bar{g}_{i_t}^{(t)} - \tilde{g}_{i_t}^{(t)}| \right), \tag{37}
\end{aligned}
$$

so we evaluate each terms of Eq. (37).

First, we bound the first term of the right hand side of Eq. (37). Using Lemma 6, we have the following bounds:

$$\frac{1}{C_1} \leq \exp\left( -\frac{1}{n\lambda_2} \sum_{\tau=\tilde{n}T}^{t-1} h_{i_\tau}(\theta)\delta\bar{g}_{i_\tau}^{(\tau)} \right) \leq C_1, \quad \frac{1}{C_1} \leq \exp\left( -\frac{1}{n\lambda_2} \sum_{\tau=\tilde{n}T}^{t-1} h_{i_\tau}(\theta)\delta\tilde{g}_{i_\tau}^{(\tau)} \right) \leq C_1. \tag{38}$$

Using (38), we have that

$$
\begin{aligned}
&\int \left| p^{(\tilde{n}T)}(\theta) \exp\left( -\frac{1}{n\lambda_2} \sum_{\tau=\tilde{n}T}^{t-1} h_{i_\tau}(\theta)\delta\bar{g}_{i_\tau}^{(\tau)} \right) - p[g^{(\tilde{n}T)}](\theta) \exp\left( -\frac{1}{n\lambda_2} \sum_{\tau=\tilde{n}T}^{t-1} h_{i_\tau}(\theta)\delta\tilde{g}_{i_\tau}^{(\tau)} \right) \right| d\theta \\
&\leq C_1 \int \left| p^{(\tilde{n}T)}(\theta) - p[g^{(\tilde{n}T)}](\theta) \right| d\theta \\
&\quad + \int p[g^{(\tilde{n}T)}](\theta) \left| \exp\left( -\frac{1}{n\lambda_2} \sum_{\tau=\tilde{n}T}^{t-1} h_{i_\tau}(\theta)\delta\bar{g}_{i_\tau}^{(\tau)} \right) - \exp\left( -\frac{1}{n\lambda_2} \sum_{\tau=\tilde{n}T}^{t-1} h_{i_\tau}(\theta)\delta\tilde{g}_{i_\tau}^{(\tau)} \right) \right| d\theta \\
&\leq C_1 \left( \epsilon_C^{(\tilde{n}T)} + \frac{1}{n\lambda_2} \sum_{\tau=\tilde{n}T}^{t-1} |\bar{g}_{i_\tau}^{(\tau+1)} - \tilde{g}_{i_\tau}^{(\tau+1)}| \right). \tag{39}
\end{aligned}
$$

Thus, combining Eqs. (38) and (39) yields we have the following bound for the first term of Eq. (37):

$$
\begin{aligned}
&\int \left| \bar{p}^{(t)}(\theta) - p[\tilde{g}^{(t)}](\theta) \right| d\theta \\
&= \int \left| \frac{p^{(\tilde{n}T)}(\theta) \exp\left( -\frac{1}{n\lambda_2} \sum_{\tau=\tilde{n}T}^{t-1} h_{i_\tau}(\theta)\delta\bar{g}_{i_\tau}^{(\tau)} \right)}{\int p^{(\tilde{n}T)}(\theta') \exp\left( -\frac{1}{n\lambda_2} \sum_{\tau=\tilde{n}T}^{t-1} h_{i_\tau}(\theta')\delta\bar{g}_{i_\tau}^{(\tau)} \right) d\theta'} - \frac{p[g^{(\tilde{n}T)}](\theta) \exp\left( -\frac{1}{n\lambda_2} \sum_{\tau=\tilde{n}T}^{t-1} h_{i_\tau}(\theta)\delta\tilde{g}_{i_\tau}^{(\tau)} \right)}{\int p[g^{(\tilde{n}T)}](\theta') \exp\left( -\frac{1}{n\lambda_2} \sum_{\tau=\tilde{n}T}^{t-1} h_{i_\tau}(\theta')\delta\tilde{g}_{i_\tau}^{(\tau)} \right) d\theta'} \right| d\theta \\
&\leq \int \left| \frac{p^{(\tilde{n}T)}(\theta) \exp\left( -\frac{1}{n\lambda_2} \sum_{\tau=\tilde{n}T}^{t-1} h_{i_\tau}(\theta)\delta\bar{g}_{i_\tau}^{(\tau)} \right)}{\int p^{(\tilde{n}T)}(\theta') \exp\left( -\frac{1}{n\lambda_2} \sum_{\tau=\tilde{n}T}^{t-1} h_{i_\tau}(\theta')\delta\bar{g}_{i_\tau}^{(\tau)} \right) d\theta'} - \frac{p^{(\tilde{n}T)}(\theta) \exp\left( -\frac{1}{n\lambda_2} \sum_{\tau=\tilde{n}T}^{t-1} h_{i_\tau}(\theta)\delta\tilde{g}_{i_\tau}^{(\tau)} \right)}{\int p[g^{(\tilde{n}T)}](\theta') \exp\left( -\frac{1}{n\lambda_2} \sum_{\tau=\tilde{n}T}^{t-1} h_{i_\tau}(\theta')\delta\tilde{g}_{i_\tau}^{(\tau)} \right) d\theta'} \right| d\theta \\
&\quad + \int \left| \frac{p^{(\tilde{n}T)}(\theta) \exp\left( -\frac{1}{n\lambda_2} \sum_{\tau=\tilde{n}T}^{t-1} h_{i_\tau}(\theta)\delta\tilde{g}_{i_\tau}^{(\tau)} \right) - p[g^{(\tilde{n}T)}](\theta) \exp\left( -\frac{1}{n\lambda_2} \sum_{\tau=\tilde{n}T}^{t-1} h_{i_\tau}(\theta)\delta\tilde{g}_{i_\tau}^{(\tau)} \right)}{\int p[g^{(\tilde{n}T)}](\theta') \exp\left( -\frac{1}{n\lambda_2} \sum_{\tau=\tilde{n}T}^{t-1} h_{i_\tau}(\theta')\delta\tilde{g}_{i_\tau}^{(\tau)} \right) d\theta'} \right| d\theta \\
&\leq \frac{\left| \int p[g^{(\tilde{n}T)}](\theta') \exp\left( -\frac{1}{n\lambda_2} \sum_{\tau=\tilde{n}T}^{t-1} h_{i_\tau}(\theta')\delta\tilde{g}_{i_\tau}^{(\tau)} \right) d\theta' - \int p^{(\tilde{n}T)}(\theta') \exp\left( -\frac{1}{n\lambda_2} \sum_{\tau=\tilde{n}T}^{t-1} h_{i_\tau}(\theta')\delta\bar{g}_{i_\tau}^{(\tau)} \right) d\theta' \right|}{\int p[g^{(\tilde{n}T)}](\theta') \exp\left( -\frac{1}{n\lambda_2} \sum_{\tau=\tilde{n}T}^{t-1} h_{i_\tau}(\theta')\delta\tilde{g}_{i_\tau}^{(\tau)} \right) d\theta'}
\end{aligned}
$$

$$+ \frac{\int \left| p^{(\tilde{n}T)}(\theta) \exp\left( -\frac{1}{n\lambda_2} \sum_{\tau=\tilde{n}T}^{t-1} h_{i_\tau}(\theta) \delta \bar{g}_{i_\tau}^{(\tau)} \right) - p[g^{(\tilde{n}T)}](\theta) \exp\left( -\frac{1}{n\lambda_2} \sum_{\tau=\tilde{n}T}^{t-1} h_{i_\tau}(\theta) \delta \tilde{g}_{i_\tau}^{(\tau)} \right) \right| \mathrm{d}\theta}{\int p[g^{(\tilde{n}T)}](\theta') \exp\left( -\frac{1}{n\lambda_2} \sum_{\tau=\tilde{n}T}^{t-1} h_{i_\tau}(\theta') \delta \tilde{g}_{i_\tau}^{(\tau)} \right) \mathrm{d}\theta'}$$

$$\leq 2C_1 \left| p^{(\tilde{n}T)}(\theta) \exp\left( -\frac{1}{n\lambda_2} \sum_{\tau=\tilde{n}T}^{t-1} h_{i_\tau}(\theta) \delta \bar{g}_{i_\tau}^{(\tau)} \right) - p[g^{(\tilde{n}T)}](\theta) \exp\left( -\frac{1}{n\lambda_2} \sum_{\tau=\tilde{n}T}^{t-1} h_{i_\tau}(\theta) \delta \tilde{g}_{i_\tau}^{(\tau)} \right) \right|$$

$$\leq 2C_1^2 \left( \epsilon_C^{(\tilde{n}T)} + \frac{1}{n\lambda_2} \sum_{\tau=\tilde{n}T}^{t-1} |\bar{g}_{i_\tau}^{(\tau+1)} - \tilde{g}_{i_\tau}^{(\tau+1)}| \right). \tag{40}$$

where we used the boundedness of each term following Eq. (38) for the second from last inequality and (39) for the last.

Finally, we bound $|\bar{g}_{i_t}^{(t+1)} - \tilde{g}_{i_t}^{(t+1)}|$ by applying (40) to (37):

$$|\bar{g}_{i_t}^{(t+1)} - \tilde{g}_{i_t}^{(t+1)}|$$

$$\leq \frac{1}{\gamma + \frac{1}{n\lambda_2}} \left( 2C_1^2 \left( \epsilon_C^{(\tilde{n}T)} + \frac{1}{n\lambda_2} \sum_{\tau=\tilde{n}T}^{t-1} |\bar{g}_{i_\tau}^{(\tau+1)} - \tilde{g}_{i_\tau}^{(\tau+1)}| \right) + \frac{1}{n\lambda_2} |\bar{g}_{i_\tau}^{(\tau+1)} - \tilde{g}_{i_\tau}^{(\tau+1)}| \right)$$

$$\leq \frac{1}{\gamma + \frac{1}{n\lambda_2}} \left( 2C_1^2 \epsilon_C^{(\tilde{n}T)} + \frac{2C_1^2 + 1}{n\lambda_2} \sum_{\tau=\tilde{n}T}^{t-1} |\bar{g}_{i_\tau}^{(\tau+1)} - \tilde{g}_{i_\tau}^{(\tau+1)}| \right). \tag{41}$$

Moreover, we have that

$$|\bar{g}_{i_{\tilde{n}T}}^{(\tilde{n}T+1)} - \tilde{g}_{i_{\tilde{n}T}}^{(\tilde{n}T+1)}| \leq \frac{1}{\gamma + \frac{1}{n\lambda_2}} \left| \int h_{i_{\tilde{n}T}}(\theta) p^{(\tilde{n}T)}(\theta) \mathrm{d}\theta - \int h_{i_{\tilde{n}T}}(\theta) p[g^{(\tilde{n}T)}](\theta) \mathrm{d}\theta \right|$$

$$\leq \frac{1}{\gamma + \frac{1}{n\lambda_2}} \int |p^{(\tilde{n}T)}(\theta) - p[g^{(\tilde{n}T)}](\theta)| \mathrm{d}\theta \leq \frac{\epsilon_C^{(\tilde{n}T)}}{\gamma + \frac{1}{n\lambda_2}}. \tag{42}$$

Combining Eqs. (41) and (42), we can apply Gronwall's lemma ("summation version" of Lemma 18): we have that, for $t = \tilde{n}T, \tilde{n}T + 1, \ldots, \tilde{n}(T+1) - 1$,

$$|\bar{g}_{i_t}^{(t+1)} - \tilde{g}_{i_t}^{(t+1)}|$$

$$\leq \left( \frac{2C_1^2}{\gamma + \frac{1}{n\lambda_2}} + \frac{2C_1^2 + 1}{\left(\gamma + \frac{1}{n\lambda_2}\right)^2 n\lambda_2} \right) \left( 1 + \frac{2C_1^2 + 1}{n\gamma\lambda_2 + 1} \right)^{\tilde{n}-2} \epsilon_C^{(\tilde{n}T)}$$

$$\leq \frac{2C_1^2}{\gamma + \frac{1}{n\lambda_2}} \left( 1 + \frac{2C_1^2 + 1}{n\gamma\lambda_2 + 1} \right)^{\tilde{n}-1} \epsilon_C^{(\tilde{n}T)}$$

$$\leq \frac{2C_1^2}{\gamma + \frac{1}{n\lambda_2}} \exp\left( \frac{\tilde{n}(2C_1^2 + 1)}{n\gamma\lambda_2 + 1} \right) \epsilon_C^{(\tilde{n}T)},$$

where we used $C_1 \geq 1$ for the second inequality. Remember that we defined $C_2$ as $C_2 = \frac{2C_1^2}{\gamma + \frac{1}{n\lambda_2}} \exp\left( \frac{\tilde{n}(2C_1^2+1)}{n\gamma\lambda_2+1} \right)$. By the definition of $\tilde{g}^{(t)}$ and $\bar{g}^{(t)}$, $\bar{g}_i^{(t)} = \tilde{g}_i^{(t)}$ holds for $t = \tilde{n}T + 1, \ldots, \tilde{n}(T+1)$ unless the coordinate $i$ is chosen during the inner loop. Otherwise, we have $\left| \tilde{g}_i^{(t)} - \bar{g}_i^{(t)} \right| \leq C_2 \epsilon_C^{(nT)}$ for all $t \geq \tau + 1$ where $\tau$ is the first time at which the coordinate $i$ is chosen as we have seen above. Thus, Eq. (36) holds for all $t = \tilde{n}T + 1, \ldots, \tilde{n}(T+1)$ and $i = 1, \ldots, n$.

$\square$

### C.2.2 Particle approximation, and error analysis of $g^{(t)}$

From now on, we bound the error induced by particle sampling in the re-weighting stage. The following three Lemmas relate, in three steps, the update of the continuous limit to the approximation by particle weighting.

**Lemma 11.** *With probability $1 - \delta$, it holds that*

$$\left| \sum_{j=1}^{M} h_{i_t}(\theta_j)\bar{r}_j^{(t)} - \int h_{i_t}(\theta)\bar{p}^{(t)}(\theta)\mathrm{d}\theta \right| \leq 7C_1^2 \sqrt{\frac{1}{M} \log\left(\frac{2n}{\delta}\right)},$$

*uniformly over all $t = \tilde{n}T, \ldots, \tilde{n}(T+1) - 1$ and the choice of the coordinates $(i_\tau)_{\tau=\tilde{n}T}^{\tilde{n}(T+1)-1}$.*

*Proof.* For $t = \tilde{n}T + 1, \ldots, \tilde{n}(T+1) - 1$, let

$$\mathcal{H} := \Bigg\{ f_{\mathcal{I},a}(\cdot) = h_{i_t}(\cdot)\exp\left(-\frac{1}{n\lambda_2}\sum_{\tau=\tilde{n}T}^{t-1} h_{i_\tau}(\cdot)a_\tau\right) C_{\mathcal{I},a}^{-1}$$

$$| \; i_t \in [n], \; \mathcal{I} = (i_{\tilde{n}T}, \ldots, i_{t-1}) \in [n]^{t-\tilde{n}T}, \; |a_\tau| \leq 2B_2 \; (\tau \in \{\tilde{n}T, \ldots, t-1\}) \Bigg\},$$

where $C_{\mathcal{I},a} := \int p^{(\tilde{n}T)}(\theta')\exp\left(-\frac{1}{n\lambda_2}\sum_{\tau=\tilde{n}T}^{t-1} h_{i_\tau}(\theta')a_\tau\right)\mathrm{d}\theta'$ for $\mathcal{I} = (i_{\tilde{n}T}, \ldots, i_{t-1})$ and $a = (a_{\tilde{n}T}, \ldots, a_{t-1})$. We also define

$$\mathcal{H}_i := \Bigg\{ f_{\mathcal{I},a}(\cdot) = h_i(\cdot)\exp\left(-\frac{1}{n\lambda_2}\sum_{\tau=\tilde{n}T}^{t-1} h_{i_\tau}(\cdot)a_\tau\right) C_{\mathcal{I},a}^{-1}$$

$$| \; \mathcal{I} = (i_{\tilde{n}T}, \ldots, i_{t-1}) \in [n]^{t-\tilde{n}T}, \; |a_\tau| \leq 2B_2 \; (\tau \in \{\tilde{n}T, \ldots, t-1\}) \Bigg\}.$$

Then, we can see that $\mathcal{H} = \cup_{i\in[n]}\mathcal{H}_i$. Then, $\sum_{j=1}^{M} h_{i_t}(\theta_j)\bar{r}_j^{(t)}$ can be rewritten as

$$\sum_{j=1}^{M} h_{i_t}(\theta_j)\bar{r}_j^{(t)} = \frac{1}{M}\sum_{j=1}^{M} f_{\mathcal{I},a}(\theta_j).$$

In a similar way, we also have that

$$\int h_{i_t}(\theta)\bar{p}^{(t)}(\theta)\mathrm{d}\theta = \mathbb{E}_\theta[f_{\mathcal{I},a}(\theta)].$$

To check this, first note that $(Mh_{i_t}(\theta_j)\bar{r}_j^{(t)})_{j=1}^{M}$ are i.i.d. random variables for each $t = \tilde{n}T + 1, \ldots, \tilde{n}(T+1) - 1$. Indeed, $\bar{r}_j^{(t)}$ is determined by $\bar{g}^{(t)}$ which is updated by using a continuous limit and thus is independent of the choice of particles $(\theta_j)_{j=1}^{M}$. Their expectations can be evaluated as

$$\mathbb{E}_{\theta_j}\left[Mh_{i_t}(\theta_j)\bar{r}_j^{(t)}\right] = \int h_{i_t}(\theta) \frac{p^{(\tilde{n}T)}(\theta)\exp\left(-\frac{1}{n\lambda_2}\sum_{\tau=\tilde{n}T}^{t-1} h_{i_\tau}(\theta)\delta\bar{g}_{i_\tau}^{(\tau)}\right)}{\int p^{(\tilde{n}T)}(\theta')\exp\left(-\frac{1}{n\lambda_2}\sum_{\tau=\tilde{n}T}^{t-1} h_{i_\tau}(\theta')\delta\bar{g}_{i_\tau}^{(\tau)}\right)\mathrm{d}\theta'}\mathrm{d}\theta$$

$$= \int h_{i_t}(\theta)\bar{p}^{(t)}(\theta)\mathrm{d}\theta.$$

Moreover, they are uniformly bounded as

$$|Mh_{i_t}(\theta_j)\bar{r}_j^{(t)}| \leq |h_{i_t}(\theta_j)|\frac{\exp\left(-\frac{1}{n\lambda_2}\sum_{\tau=\tilde{n}T}^{t-1} h_{i_\tau}(\theta_j)\delta\bar{g}_{i_\tau}^{(\tau)}\right)}{\int p^{(\tilde{n}T)}(\theta')\exp\left(-\frac{1}{n\lambda_2}\sum_{\tau=\tilde{n}T}^{t-1} h_{i_\tau}(\theta')\delta\bar{g}_{i_\tau}^{(\tau)}\right)\mathrm{d}\theta'} \leq C_1^2. \tag{43}$$

The same results are also true for $t = \tilde{n}T$.

Note that it suffices to bound $Z = \sup_{f_{\mathcal{I},a}\in\mathcal{H}} |\frac{1}{M}\sum_{j=1}^{M} f_{\mathcal{I},a}(\theta_j) - \mathbb{E}_\theta[f_{\mathcal{I},a}(\theta)]|$ to obtain the assertion. For that purpose, we define the Rademacher complexity of a function class $\mathcal{H}'$ as

$$\mathcal{R}(\mathcal{H}') := \mathbb{E}\left[\sup_{f\in\mathcal{H}'}\frac{1}{M}\sum_{j=1}^{M}\varepsilon_j f(\theta_j)\right],$$

where $(\varepsilon_j)_{j=1}^{M}$ is an i.i.d. sequence of the Rademacher random variables ($P(\varepsilon_j = 1) = P(\varepsilon_j = -1) = 1/2$) and the expectation is taken with respect to both $(\varepsilon_j)_{j=1}^{M}$ and $(\theta_j)_{j=1}^{M}$. Then, the

Rademacher concentration inequality (Mohri et al., 2012, Theorem 3.1) tells that the uniform bound on a function class $\mathcal{H}'$ can be obtained as

$$P\left(\sup_{f'\in\mathcal{H}'}\left|\frac{1}{M}\sum_{j=1}^{M}(f(\theta_j)-\mathbb{E}_\theta[f(\theta)])\right|\geq 2\mathcal{R}(\mathcal{H}')+\sup_{f\in\mathcal{H}'}\|f\|_\infty\sqrt{\frac{1}{2M}\log\left(\frac{2}{\delta}\right)}\right)\leq\delta,\quad(44)$$

for $0<\delta<1$. Here, let $Z_i=\sup_{f\in\mathcal{H}_i}|\frac{1}{M}\sum_{j=1}^{M}f(\theta_j)-\mathbb{E}_\theta[f(\theta)]|$ and then we also see that $Z=\max_{i\in[n]}Z_i$. Hence, we have that

$$P(Z\geq s)\leq\sum_{i_t=1}^{n}P(Z_{i_t}\geq s),$$

for $s>0$. If we substitute $s\leftarrow\max_{i\in[n]}2\mathcal{R}(\mathcal{H}_i)+\sup_{f\in\mathcal{H}_i}\|f\|_\infty\sqrt{\frac{1}{2M}\log\left(\frac{2n}{\delta}\right)}$, then by Eq. (44) we can see that the right hand side can be bounded by $\delta$. Note that we have already seen that $\|f\|_\infty\leq C_1^2$ for any $f\in\mathcal{H}$ (see Eq. (43)). Therefore, we just need to bound $\mathcal{R}(\mathcal{H}_{i_t})$. By the contraction property of the Rademacher complexity ((Boucheron et al., 2013, Theorem 11.6) or (Ledoux & Talagrand, 1991, Theorem 4.12) and its proof) and remembering the fact that the Rademacher complexity of a function class is same as that of its convex hull (Mohri et al., 2012, Theorem 6.2), we obtain

$$\mathcal{R}(\mathcal{H}_{i_t})$$
$$\leq\mathbb{E}\left[\sup_{\mathcal{I},a}\frac{1}{M}\sum_{j=1}^{M}\varepsilon_j\exp\left(-\frac{1}{n\lambda_2}\sum_{\tau=\tilde{n}T}^{t-1}h_{i_\tau}(\theta_j)a_\tau-\log C_{\mathcal{I},a}\right)\right]\quad(\because|h_{i_t}(\cdot)|\leq 1)$$
$$\leq\frac{C_1}{\lambda_2}\mathbb{E}\left[\sup_{\mathcal{I},a}\frac{1}{M}\sum_{j=1}^{M}\varepsilon_j\left(-\frac{1}{n}\sum_{\tau=\tilde{n}T}^{t-1}h_{i_\tau}(\theta_j)a_\tau-\lambda_2\log C_{\mathcal{I},a}\right)\right]$$
$$\leq\frac{2C_1B_2}{\lambda_2}\mathbb{E}\left[\max_{i\in[n],s\in\{\pm1\},\xi:|\xi|\leq\log(C_1)}\frac{1}{M}\sum_{j=1}^{M}\varepsilon_j\left(sh_i(\theta_j)+\frac{\lambda_2}{2B_2}\xi\right)\right]$$
$$\leq\frac{4C_1B_2}{\lambda_2}\sqrt{\frac{2}{M}\log(2n)},$$

where we used the contraction properties in the first and second inequalities, we used the convex hull argument in the third inequality, and the last inequality is by Massart's Lemma (Mohri et al., 2012, Theorem 3.3). Summarizing these evaluations, we obtain that

$$P\left(Z\geq\frac{8C_1B_2}{\lambda_2}\sqrt{\frac{2}{M}\log(2n)}+C_1^2\sqrt{\frac{1}{2M}\log\left(\frac{2n}{\delta}\right)}\right)\leq\delta.$$

The same bound also holds for $t=\tilde{n}T$.

Then, by taking the uniform bound over $t=\tilde{n}T,\ldots,\tilde{n}(T+1)-1$ and simpifying the bound using $\tilde{n}\leq n$, we yield that

$$\left|\sum_{j=1}^{M}h_{i_t}(\theta_j)\bar{r}_j^{(t)}-\int h_{i_t}(\theta)\bar{p}^{(t)}(\theta)\mathrm{d}\theta\right|\leq 7C_1^2\sqrt{\frac{1}{M}\log\left(\frac{2n}{\delta}\right)}$$

uniformly over $t=\tilde{n}T,\ldots,\tilde{n}(T+1)-1$ and the choice of the sequence $(i_\tau)_{\tau=\tilde{n}T}^{\tilde{n}(T+1)-1}$ with probability $1-\delta$, which yields the assertion. □

**Lemma 12.** *With probability $1-\delta$, it holds that*

$$\left|\sum_{j=1}^{M}h_i(\theta_j)\frac{\hat{r}_j^{(t)}}{\sum_{j'=1}^{M}\hat{r}_{j'}^{(t)}}-\sum_{j=1}^{M}h_i(\theta_j)\bar{r}_j^{(t)}\right|\leq 7C_1^2\sqrt{\frac{1}{M}\log\left(\frac{2n}{\delta}\right)},$$

*uniformly over all $t=\tilde{n}T,\ldots,\tilde{n}(T+1)-1$, $i=1,\ldots,n$ and the choice of coordinates $(i_\tau)_{\tau=\tilde{n}T}^{\tilde{n}(T+1)-1}$.*

*Proof.* Let $C_p = \int p^{(\tilde{n}T)}(\theta) \exp\left(-\frac{1}{n\lambda_2} \sum_{\tau=\tilde{n}T}^{t-1} h_{i_\tau}(\theta)\delta\bar{g}^{(\tau)}\right) d\theta$. Then, noticing that $\hat{r}_j^{(t)} = \hat{r}_j^{(t)}/C_p$ and $\|h_i\|_\infty \le 1$, we have that

$$
\left| \sum_{j=1}^{M} h_i(\theta_j) \frac{\hat{r}_j^{(t)}}{\sum_{j'=1}^{M} \hat{r}_{j'}^{(t)}} - \sum_{j=1}^{M} h_i(\theta_j) \bar{r}_j^{(t)} \right|
$$

$$
= \left| \sum_{j=1}^{M} h_i(\theta_j) \frac{\hat{r}_j^{(t)}}{\sum_{j'=1}^{M} \hat{r}_{j'}^{(t)}} - \sum_{j=1}^{M} h_i(\theta_j) \frac{\hat{r}_j^{(t)}}{C_p} \right|
$$

$$
= \left| \sum_{j=1}^{M} h_i(\theta_j) \frac{\hat{r}_j^{(t)}}{\sum_{j'=1}^{M} \hat{r}_{j'}^{(t)}} \right| \left| 1 - \frac{\sum_{j=1}^{M} \hat{r}_j^{(t)}}{C_p} \right|
$$

$$
\le \left| 1 - \sum_{j=1}^{M} \bar{r}_j^{(t)} \right|.
$$

We apply the same argument as Lemma 11 to evaluate the far right hand side $\left| 1 - \sum_{j=1}^{M} \bar{r}_j^{(t)} \right|$ by replacing $h_{i_t}(\theta)$ with 1 in the definition of $\mathcal{H}$. Then, we obtain the assertion. $\qquad\square$

From now on, we define $\epsilon_D$ as

$$
\epsilon_D \stackrel{\text{def}}{=} 7\sqrt{\frac{1}{M} \log\left(\frac{4n}{\delta}\right)}.
$$

Then, we have the following bound.

**Lemma 13.** *We have that, with probability $1 - \delta$ with respect to realization of particles $(\theta_j)_{j=1}^{M}$,*

$$
|\bar{g}_i^{(t)} - g_i^{(t)}| \le C_2 C_1^2 \epsilon_D \tag{45}
$$

*for all $t = \tilde{n}T + 1, \ldots, \tilde{n}(T+1)$, $i = 1, \ldots, n$ and the choice of coordinates $(i_\tau)_{\tau=\tilde{n}T}^{\tilde{n}(T+1)-1}$ uniformly.*

*Proof.* From Lemma 11 and 12, with probability $1 - \delta$, we have that

$$
\left| \sum_{j=1}^{M} h_{i_t}(\theta_j) \bar{r}_j^{(t)} - \int h_{i_t}(\theta) \bar{p}^{(t)}(\theta) d\theta \right| \le C_1^2 \epsilon_D,
$$

and that

$$
\left| \sum_{j=1}^{M} h_{i_t}(\theta_j) \frac{\hat{r}_j^{(t)}}{\sum_{j'=1}^{M} \hat{r}_{j'}^{(t)}} - \sum_{j=1}^{M} h_{i_t}(\theta_j) \bar{r}_j^{(t)} \right| \le C_1^2 \epsilon_D,
$$

for all $t = \tilde{n}T, \ldots, \tilde{n}(T+1) - 1$. Therefore, by Lemma 8, we have that

$$
|g_{i_t}^{(t+1)} - \bar{g}_{i_t}^{(t+1)}|
$$

$$
\le \frac{1}{\gamma + \frac{1}{n\lambda_2}} \left( \left| \sum_{j=1}^{M} h_i(\theta_j) \frac{r_j^{(t)}}{\sum_{j'=1}^{M} r_{j'}^{(t)}} - \int h_i(\theta) \bar{p}^{(t)}(\theta) d\theta \right| + \frac{1}{n\lambda_2} |g_i^{(t)} - \bar{g}_i^{(t)}| \right)
$$

$$
\le \frac{1}{\gamma + \frac{1}{n\lambda_2}} \left( 2C_1^2 \epsilon_D + \left| \sum_{j=1}^{M} h_i(\theta_j) \left( \frac{r_j^{(t)}}{\sum_{j'=1}^{M} r_{j'}^{(t)}} - \frac{\hat{r}_j^{(t)}}{\sum_{j'=1}^{M} \hat{r}_{j'}^{(t)}} \right) \right| + \frac{1}{n\lambda_2} |g_i^{(t)} - \bar{g}_i^{(t)}| \right). \tag{46}
$$

The second term of the right hand side of Eq. (46) can be further bounded as

$$
\left| \sum_{j=1}^{M} h_i(\theta_j) \left( \frac{r_j^{(t)}}{\sum_{j'=1}^{M} r_{j'}^{(t)}} - \frac{\hat{r}_j^{(t)}}{\sum_{j'=1}^{M} \hat{r}_{j'}^{(t)}} \right) \right|
$$

$$\leq \sum_{j=1}^{M} \frac{\left| r_j^{(t)} - \hat{r}_j^{(t)} \right|}{\sum_{j'=1}^{M} r_{j'}^{(t)}} + \sum_{j=1}^{M} \left| \frac{\hat{r}_j^{(t)}}{\sum_{j'=1}^{M} r_{j'}^{(t)}} - \frac{\hat{r}_j^{(t)}}{\sum_{j'=1}^{M} \hat{r}_{j'}^{(t)}} \right|$$

$$\leq C_1 \sum_{j=1}^{M} \left| r_j^{(t)} - \hat{r}_j^{(t)} \right| + \sum_{j=1}^{M} \frac{\hat{r}_j^{(t)}}{\sum_{j'=1}^{M} \hat{r}_{j'}^{(t)}} \left| \frac{\sum_{j'=1}^{M} r_{j'}^{(t)} - \sum_{j'=1}^{M} \hat{r}_{j'}^{(t)}}{\sum_{j'=1}^{M} r_{j'}^{(t)}} \right|$$

$$\leq C_1 \sum_{j=1}^{M} \left| r_j^{(t)} - \hat{r}_j^{(t)} \right| + C_1 \left| \sum_{j'=1}^{M} r_{j'}^{(t)} - \sum_{j'=1}^{M} \hat{r}_{j'}^{(t)} \right|$$

$$\leq C_1 \sum_{j=1}^{M} \left| r_j^{(t)} - \hat{r}_j^{(t)} \right| + C_1 \sum_{j'=1}^{M} \left| r_{j'}^{(t)} - \hat{r}_{j'}^{(t)} \right|$$

$$\leq 2C_1 \sum_{j=1}^{M} \left| r_j^{(t)} - \hat{r}_j^{(t)} \right|$$

$$\leq 2C_1 \sum_{j=1}^{M} \left| \frac{1}{M} \exp\left( -\frac{1}{n\lambda_2} \sum_{\tau=\tilde{n}T}^{t} h_{i_\tau}(\theta_j) \delta g_{i_\tau}^{(\tau)} \right) - \frac{1}{M} \exp\left( -\frac{1}{n\lambda_2} \sum_{\tau=\tilde{n}T}^{t} h_{i_\tau}(\theta_j) \delta \bar{g}_{i_\tau}^{(\tau)} \right) \right|$$

$$\leq \frac{2C_1^2}{n\lambda_2} \sum_{\tau=\tilde{n}T}^{t-1} |g_{i_\tau}^{(\tau+1)} - \bar{g}_{i_\tau}^{(\tau+1)}| \tag{47}$$

for $t = \tilde{n}T, \ldots, \tilde{n}(T+1) - 1$. This and Eq. (46) yields

$$|g_{i_t}^{(t+1)} - \bar{g}_{i_t}^{(t+1)}|$$
$$\leq \frac{1}{\gamma + \frac{1}{n\lambda_2}} \left( 2C_1^2 \epsilon_D + \frac{2C_1^2 + 1}{n\lambda_2} \sum_{\tau=\tilde{n}T}^{t-1} |g_{i_\tau}^{(\tau+1)} - \bar{g}_{i_\tau}^{(\tau+1)}| \right)$$

for $t = \tilde{n}T + 1, \ldots, \tilde{n}(T+1) - 1$. Also, it holds that

$$|g_i^{(\tilde{n}T+1)} - \bar{g}_i^{(\tilde{n}T+1)}| \leq \frac{C_1^2}{\gamma + \frac{1}{n\lambda_2}} \epsilon_D,$$

from Lemma 11.

Therefore, applying Gronwall's lemma ("summation version" of Lemma 18) yields that for $t = \tilde{n}T, \ldots, \tilde{n}(T+1) - 1$,

$$|g_{i_t}^{(t+1)} - \bar{g}_{i_t}^{(t+1)}| \leq \left( \frac{2C_1^2}{\gamma + \frac{1}{n\lambda_2}} + \frac{2C_1^2 + 1}{\left( \gamma + \frac{1}{n\lambda_2} \right)^2 n\lambda_2} \right) \left( 1 + \frac{2C_1^2 + 1}{n\gamma\lambda_2 + 1} \right)^{\tilde{n}-2} C_1^2 \epsilon_D$$

$$\leq \frac{2C_1^2}{\gamma + \frac{1}{n\lambda_2}} \left( 1 + \frac{2C_1^2 + 1}{n\gamma\lambda_2 + 1} \right)^{\tilde{n}-1} C_1^2 \epsilon_D$$

$$\leq \frac{2C_1^2}{\gamma + \frac{1}{n\lambda_2}} \exp\left( \frac{\tilde{n}(2C_1^2 + 1)}{n\gamma\lambda_2 + 1} \right) C_1^2 \epsilon_D$$

$$\leq C_2 C_1^2 \epsilon_D,$$

which concludes the proof of Eq. (45). $\qquad\qquad\square$

### C.2.3 INTEGRATION OF TWO SOURCES OF DISCRETIZATION ERROR

From now on, we integrate the convergence analysis from Section C.2.1 and Section C.2.2. Then, we extend our analysis from that of each inner loop to that of the outer loop over $T = 0, \ldots, T_{\text{end}}$. So far, we have been arguing only within one inner loop, while we may choose a different number

$M$ of particles in each outer-loop $T$. Accordingly, we denote the number of particles at the $T$-th outer-iteration by $M^{(\tilde{n}T)}$ and add a subscript to $\epsilon_D$ as

$$\epsilon_D^{(\tilde{n}T)} \stackrel{\text{def}}{=} 7\sqrt{\frac{1}{M^{(\tilde{n}T)}} \log\left(\frac{4n(T_{\text{end}})}{\delta}\right)}.$$

Moreover, we assume that the bound of Lemma 13 holds for every $T = 0, \ldots, T_{\text{end}} - 1$. We denote by $\mathcal{E}$ this even. We know that $P(\mathcal{E}) \geq 1 - \delta$. In the following, we analyze the convergence under the event $\mathcal{E}$ where the bounds in Lemma 13 hold, so that $|\bar{g}_i^{(t)} - g_i^{(t)}| \leq (C_2\epsilon_C^{(\tilde{n}T)} + C_2C_1^2\epsilon_D^{(\tilde{n}T)}) \, (T = \lceil \frac{t}{\tilde{n}} \rceil - 1)$ holds owing to Lemma 10 and Lemma 13. Therefore, the expectation with respect to the choice of coordinates is taken under the condition of this event. We note that the bound in Lemma 13 holds uniformly over the choice of the coordinates and thus the distribution of the coordinates is not affected by being conditioned by the event.

In order to evaluate the effect of the discrepancy of coordinate updates on the difference of $\epsilon_A$, we present the following Lemma 14 on the local smoothness of dual space.

**Lemma 14.** *Suppose that $n \geq \frac{2B_2}{\lambda_2}$ and Assumption 1 holds. Then, $D(g)$ is $\frac{1}{n\lambda_2}(\lambda_2\gamma' + 1)$-smooth with respect to $g \in \mathbb{R}^n$ on the set $\mathcal{A}$ defined by*

$$\mathcal{A} = \left\{ g \in \mathbb{R}^n \;\middle|\; \min\left\{\min_t g_i^{(t)}, \min_t \tilde{g}_i^{(t)}\right\} \leq g_i \leq \max\left\{\max_t g_i^{(t)}, \max_t \tilde{g}_i^{(t)}\right\} \; (i = 1, \ldots, n) \right\},$$

*where $t$ takes $t = 0, 1, \ldots, \tilde{n}T_{\text{end}}$.*

*Proof.* Suppose that the coordinate $i$ is chosen at the $t$-th iteration. By the definition of $g^{(t)}$, we have that

$$|\ell_i^{*\prime}(g_i^{(t)})| \leq \sum_{j=1}^M |h_i(\theta_j)| \frac{r_j^{(t)}}{\sum_{j'=1}^M r_{j'}^{(t)}} + \frac{1}{n\lambda_2}|g_i^{(t)} - g_i^{(t-1)}|, \tag{48}$$

where $\ell_i^{*\prime}$ is an element of the sub-differential. The first term of the right hand side can be bounded by 1 as follows:

$$\sum_{j=1}^M |h_i(\theta_j)| \frac{r_j^{(t)}}{\sum_{j'=1}^M r_{j'}^{(t)}} \leq \max_j |h_i(\theta_j)| \sum_{j=1}^M \frac{r_j^{(t)}}{\sum_{j'=1}^M r_{j'}^{(t)}} \leq 1.$$

Also, Lemma 6 implies that the second term can be bounded by $\frac{2B_2}{n\lambda_2}$. Therefore, under the assumption $n \geq \frac{2B_2}{\lambda_2}$, we have that

$$|\ell_i^{*\prime}(g_i^{(t)})| \leq 2.$$

Applying the same argument, we also have that

$$|\ell_i^{*\prime}(\tilde{g}_i^{(t)})| \leq 2.$$

Therefore, for $i \in [n]$ that has already chosen in the algorithm until the $t$-th step, we see that $\ell_i^*$ is $\gamma'$-smooth on the interval between $g_i^{(t)}$ and $\tilde{g}_i^{(t)}$. For the coordinates $i$ that have not yet chosen, we have that $g_i^{(t)} = \tilde{g}_i^{(t)} = g_i^{(0)}$ and $|\ell_i^{*\prime}(g_i^{(0)})| \leq 1$. Therefore $\ell_i^*$ is $\gamma'$-smooth on the box $\mathcal{A}$. By summarizing this argument and applying Lemma 4, we see that $D(g)$ is $\frac{1}{n\lambda_2}(\lambda_2\gamma' + 1)$-smooth. □

Combining the above results, we obtain the evalution of $\epsilon_A$ and sufficient conditions for the first statement of Lemma 8.

**Lemma 15.** *Suppose that $n \geq \frac{2B_2}{\lambda_2}$ and fix an arbitrary $T$. if $\mathbb{E}[D(\tilde{g}^{(\tilde{n}T')}) - D(g^{(\tilde{n}T')})] \leq C_0 \exp\left(-\tilde{s}T'\right)$ is satisfied for all $T' \leq T$, then we have that*

$$\mathbb{E}[D(\tilde{g}^{(t)}) - D(g^{(t)})] \leq \frac{C_0}{2}\exp\left(-\tilde{s}(T+1)\right) + \frac{1}{(2-\mu)\mu}\left(\gamma' + \frac{1}{\lambda_2}\right)\left(C_2\epsilon_C^{(nT)} + C_2C_1^2\epsilon_D^{(nT)}\right)^2, \tag{49}$$

*for all $t = \tilde{n}T + 1, \ldots, \tilde{n}(T+1)$, where $\mu = \frac{2}{1 + \frac{2e^{\tilde{s}}}{C_0}\left(B_1 + B_2 + \frac{C_0}{1 - e^{-\tilde{s}}}\right)}$.*

*In particular, if*

$$\epsilon_C^{(\tilde{n}T)} \le C_3 \exp\left(-\frac{\tilde{s}T}{2}\right), \quad \epsilon_D^{(\tilde{n}T)} \le \frac{C_3}{C_1^2} \exp\left(-\frac{\tilde{s}T}{2}\right)$$

*are satisfied for all $T = 0, \ldots, T_{\text{end}}$, where*

$$C_3 \le \frac{\sqrt{C_0(2-\mu)\mu}}{C_2\sqrt{2e^{\tilde{s}}(\gamma' + 1/\lambda_2)}}, \tag{50}$$

*then it holds that for all $t = 0, 1, \ldots, \tilde{n}T_{\text{end}}$,*

$$\mathbb{E}[D(\tilde{g}^{(t)}) - D(g^{(t)})] \le C_0 \exp\left(-\tilde{s}T\right),$$

*where $T = \lceil \frac{t}{\tilde{n}} \rceil$.*

*Proof.* Under the assumptions that $n \ge \frac{2B_2}{\lambda_2}$, Lemma 14 holds. Fix some $T$ and we will show the first assertion for $t = \tilde{n}T + 1, \ldots, \tilde{n}(T+1)$.

Let $\mathcal{A}_t = \left\{s\tilde{g}^{(t)} + (1-s)g^{(t)} \mid s \in [0,1]\right\}$, then Lemma 15 implies that $D(g)$ is $\frac{1}{n\lambda_2}(\lambda_2\gamma' + 1)$-smooth on the set. Hence, for $\mu = \frac{2}{1 + \frac{2e^{\tilde{s}}}{C_0}\left(B_1 + B_2 + \frac{C_0}{1-e^{-\tilde{s}}}\right)}$, the Cauchy-Schwarz inequality yields

$$D(\tilde{g}^{(t)}) - D(g^{(t)})$$
$$\le \sup_{g \in \mathcal{A}_t} \|\nabla D(g)\|_2 \|\tilde{g}^{(t)} - g^{(t)}\|_2$$
$$\le \frac{1}{2}\left[\mu \frac{n\lambda_2}{\lambda_2\gamma' + 1}\left(\sup_{g \in \mathcal{A}_t} \|\nabla D(g)\|_2\right)^2 + \frac{1}{\mu}\frac{\lambda_2\gamma' + 1}{n\lambda_2}\|\tilde{g}^{(t)} - g^{(t)}\|_2^2\right]$$
$$\le \frac{1}{2}\left[\mu \sup_{g \in \mathcal{A}_t}[D(g^*) - D(g)] + \frac{1}{\mu}\left(\gamma' + \frac{1}{\lambda_2}\right)\left(C_2\epsilon_C^{(\tilde{n}T)} + C_2C_1^2\epsilon_D^{(\tilde{n}T)}\right)^2\right].$$

If $D(\tilde{g}^{(t)}) - D(g^{(t)}) \ge 0$, then we have that

$$D(\tilde{g}^{(t)}) - D(g^{(t)})$$
$$\le \frac{1}{2}\left[\mu\left(D(g^*) - D(g^{(t)})\right) + \frac{1}{\mu}\left(\gamma' + \frac{1}{\lambda_2}\right)\left(C_2\epsilon_C^{(\tilde{n}T)} + C_2C_1^2\epsilon_D^{(\tilde{n}T)}\right)^2\right].$$

Rearranging the terms we have that

$$D(\tilde{g}^{(t)}) - D(g^{(t)})$$
$$\le \frac{\mu}{2-\mu}\left(D(g^*) - D(\tilde{g}^{(t)})\right) + \frac{1}{(2-\mu)\mu}\left(\gamma' + \frac{1}{\lambda_2}\right)\left(C_2\epsilon_C^{(\tilde{n}T)} + C_2C_1^2\epsilon_D^{(\tilde{n}T)}\right)^2,$$

This holds as well when $D(\tilde{g}^{(t)}) - D(g^{(t)}) \le 0$.

Here, Lemma 5 implies that $\mathbb{E}[D(\tilde{g}^{(t)})]$ is monotonically non-decreasing within the same inner loop, and thus we have that $\mathbb{E}\left[D(g^*) - D(\tilde{g}^{(t)})\right] \le \mathbb{E}\left[D(g^*) - D(g^{(\tilde{n}T)})\right]$. Moreover, the right hand side $\mathbb{E}\left[D(g^*) - D(g^{(\tilde{n}T)})\right]$ can be further bounded by $\left(B_1 + B_2 + \frac{C_0}{1-e^{-\tilde{s}}}\right)\exp\left(-\tilde{s}T\right) = \frac{C_0}{2}\frac{2-\mu}{\mu}\exp(-\tilde{s}(T+1))$ by using the assumption and applying Eq. (33) of Lemma 8. Therefore,

$$\mathbb{E}\left[D(\tilde{g}^{(t)}) - D(g^{(t)})\right]$$
$$\le \frac{C_0}{2}\exp\left(-\tilde{s}(T+1)\right) + \frac{1}{(2-\mu)\mu}\left(\gamma' + \frac{1}{\lambda_2}\right)\left(C_2\epsilon_C^{(\tilde{n}T)} + C_2C_1^2\epsilon_D^{(\tilde{n}T)}\right)^2.$$

This concludes the first assertion.

We will show the second assertion by induction. It is obvious that this is true at initialization. Suppose that Eq. (50) holds for $t = 0, 1, \ldots \tilde{n}T$. Then the assumption of the first statement of Lemma 8 holds for $T' = 0, \ldots, T$, which leads to Eq. (49) for $t = \tilde{n}T + 1, \ldots, \tilde{n}(T + 1)$. Then we have that Eq. (50) also holds for $t = \tilde{n}T + 1, \ldots, \tilde{n}(T + 1)$, since we chose $C_3$ as

$$C_3 = \frac{\sqrt{C_0(2 - \mu)\mu}}{C_2\sqrt{2e^{\tilde{s}}(\gamma' + 1/\lambda_2)}},$$

so that the right hand side of (49) is bounded by $C_0 \exp(-\tilde{s}(T + 1))$. Thus, the proof is completed by induction.

$\square$

Although The proof for the first statement of Lemma 8 is almost done, we have some proof left for the second claim. The next two Lemmas measure the effect of discrepancy of coordinates in the dual space on the value of the main problem.

**Lemma 16.** $P(p[g])$ is $\frac{1}{n\lambda_2^2}(\gamma + 3\lambda_2 + 9B_2)$-smooth with respect to $g$ on the set $\mathcal{A}$.

*Proof.* We show the assertion by a direct calculation. Let $\mathbb{E}_{p[g]}[h_i] \overset{\text{def}}{=} \int h_i(\theta)p[g](\theta)\mathrm{d}\theta$. First, the second derivative of the loss function can be calculated as

$$\frac{\partial^2}{\partial g_j g_k}\ell_i\left(\mathbb{E}_{p[g]}[h_i]\right)$$

$$= \frac{1}{n\lambda_2}\frac{\partial}{\partial g_k}\ell_i'\left(\mathbb{E}_{p[g]}[h_i]\right)\left(\mathbb{E}_{p[g]}[h_i]\mathbb{E}_{p[g]}[h_j] - \mathbb{E}_{p[g]}[h_ih_j]\right)$$

$$= \frac{1}{n^2\lambda_2^2}\left[\ell_i''\left(\mathbb{E}_{p[g]}[h_i]\right)\left(\mathbb{E}_{p[g]}[h_i]\mathbb{E}_{p[g]}[h_j] - \mathbb{E}_{p[g]}[h_ih_j]\right)\left(\mathbb{E}_{p[g]}[h_i]\mathbb{E}_{p[g]}[h_k] - \mathbb{E}_{p[g]}[h_ih_k]\right)\right.$$

$$+ \ell_i'\left(\mathbb{E}_{p[g]}[h_i]\right)\cdot\left(\left(\mathbb{E}_{p[g]}[h_i]\mathbb{E}_{p[g]}[h_k] - \mathbb{E}_{p[g]}[h_ih_k]\right)\mathbb{E}_{p[g]}[h_j]\right.$$

$$\left.\left. + \mathbb{E}_{p[g]}[h_i]\left(\mathbb{E}_{p[g]}[h_j]\mathbb{E}_{p[g]}[h_k] - \mathbb{E}_{p[g]}[h_jh_k]\right) - \left(\mathbb{E}_{p[g]}[h_ih_j]\mathbb{E}_{p[g]}[h_k] - \mathbb{E}_{p[g]}[h_ih_jh_k]\right)\right)\right]$$

$$\leq \frac{1}{n^2\lambda_2^2}(\gamma + 3B_2),$$

where we used smoothness of $\ell_i$ for the last inequality.

The regularization term can be split into the following two terms:

$$\lambda_1\int p[g](\theta)\|\theta\|_2^2\mathrm{d}\theta + \lambda_2\int p[g](\theta)\log p[g](\theta)\mathrm{d}\theta$$

$$= -\frac{1}{n}\sum_{i=1}^{n}\int h_i(\theta)p[g](\theta)\mathrm{d}\theta g_i + \lambda_2\log\left[\int q[g](\theta)\mathrm{d}\theta\right]$$

We already confirmed $\frac{1}{n\lambda_2}$−smoothness of the second term in Lemma 4. As for the first term, we have that

$$-\frac{\partial^2}{\partial g_j g_k}\left(\frac{1}{n}\sum_{i=1}^{n}\mathbb{E}_{p[g]}[h_i]g_i\right)$$

$$= -\frac{1}{n\lambda_2}\frac{\partial^2}{\partial g_k}\left(\lambda_2\mathbb{E}_{p[g]}[h_j] + \frac{1}{n}\sum_{i=1}^{n}\left(\mathbb{E}_{p[g]}[h_i]\mathbb{E}_{p[g]}[h_j] - \mathbb{E}_{p[g]}[h_ih_j]\right)g_i\right)$$

$$= -\frac{1}{n^2\lambda_2^2}\left[2\lambda_2\left(\mathbb{E}_{p[g]}[h_j]\mathbb{E}_{p[g]}[h_k] - \mathbb{E}_{p[g]}[h_jh_l]\right) + \frac{1}{n}\sum_{i=1}^{n}\left(\left(\mathbb{E}_{p[g]}[h_i]\mathbb{E}_{p[g]}[h_k] - \mathbb{E}_{p[g]}[h_ih_k]\right)\mathbb{E}_{p[g]}[h_j]\right.\right.$$

$$\left.\left. + \mathbb{E}_{p[g]}[h_i]\left(\mathbb{E}_{p[g]}[h_j]\mathbb{E}_{p[g]}[h_k] - \mathbb{E}_{p[g]}[h_jh_k]\right) - \left(\mathbb{E}_{p[g]}[h_ih_j]\mathbb{E}_{p[g]}[h_k] - \mathbb{E}_{p[g]}[h_ih_jh_k]\right)\right)g_i\right]$$

$$\leq \frac{1}{n^2\lambda_2^2}(2\lambda_2 + 6B_2),$$

where we used Lemma 6 for the last inequality to bound $g_i$.

Combining the two inequalities and the result from Lemma 4, we obtain the assertion. $\square$

**Lemma 17.** *Suppose Eq. (50) of Lemma 15 and its assumption hold for $t = 0, 1, \ldots, \tilde{n}T_{\text{end}}$. Then, we have that*

$$\mathbb{E}\left[P(p[g^{(t)}]) - P(p[\tilde{g}^{(t)}])\right]$$

$$\leq \frac{2\gamma + 6\lambda_2 + 18B_2}{\lambda_2^{\frac{3}{2}}\sqrt{\gamma}}\sqrt{B_1 + B_2 + C_0 + \frac{C_0}{1 - e^{-\tilde{s}}}}\frac{\sqrt{C_0 e^{\tilde{s}}(2 - \mu)\mu}}{\sqrt{2(\gamma'\lambda_2 + 1)}}\exp(-\tilde{s}T),$$

*where $T = \left\lceil \frac{t}{n} \right\rceil$.*

*Proof.* By using the smoothness of $P(p[g])$ by Lemma 16, we have that

$$P(p[g^{(t)}]) - P(p[\tilde{g}^{(t)}])$$

$$\leq \sup_{g \in \mathcal{A}_t} \|\nabla P(p[g])\|_2 \|\tilde{g}^{(t)} - g^{(t)}\|_2$$

$$\leq \sqrt{\frac{n\lambda_2^2}{\gamma + 3\lambda_2 + 9B_2}\left\{\sup_g \|\nabla P(p[g])\|_2\right\}^2}\sqrt{\frac{\gamma + 3\lambda_2 + 9B_2}{n\lambda_2^2}}\|\tilde{g}^{(t)} - g^{(t)}\|_2$$

$$\leq \sqrt{\sup_g P(p[g]) - P(p[g^*])}\sqrt{\frac{\gamma + 3\lambda_2 + 9B_2}{\lambda_2^2}}\left(C_2\epsilon_C^{(\tilde{n}T)} + C_2C_1^2\epsilon_C^{(\tilde{n}T)}\right).$$

where $T = \left\lceil \frac{t}{n} \right\rceil - 1$. if $P(p[g^{(t)}]) - P(p[\tilde{g}^{(t)}]) \geq 0$, then we have that

$$P(p[g^{(t)}]) - P(p[\tilde{g}^{(t)}])$$

$$\leq \sqrt{P(p[g^{(t)}]) - P(p[g^*])}\sqrt{\frac{\gamma + 3\lambda_2 + 9B_2}{\lambda_2^2}}\left(C_2\epsilon_C^{(\tilde{n}T)} + C_2C_1^2\epsilon_C^{(\tilde{n}T)}\right)$$

$$\leq \frac{\gamma + 3\lambda_2 + 9B_2}{\lambda_2^2\sqrt{n}}\|g^{(t)} - g^*\|_2\left(C_2\epsilon_C^{(\tilde{n}T)} + C_2C_1^2\epsilon_C^{(\tilde{n}T)}\right).$$

Since $D$ is $\frac{\gamma}{n}$-strongly convex, we have that$\|g^{(t)} - g^*\|_2 \leq \sqrt{\frac{n}{\gamma}\left(D(g^*) - D(g^{(t)})\right)}$. Therefore, it holds that

$$P(p[g^{(t)}]) - P(p[\tilde{g}^{(t)}])$$

$$\leq \frac{\gamma + 3\lambda_2 + 9B_2}{\lambda_2^2\sqrt{\gamma}}\sqrt{\left(D(g^*) - D(g^{(t)})\right)}\left(C_2\epsilon_C^{(\tilde{n}T)} + C_2C_1^2\epsilon_C^{(\tilde{n}T)}\right).$$

which holds as well when $P(p[g^{(t)}]) - P(p[\tilde{g}^{(t)}]) \leq 0$.

By taking expectations, it yields that

$$\mathbb{E}\left[P(p[g^{(t)}]) - P(p[\tilde{g}^{(t)}])\right]$$

$$\leq \frac{\gamma + 3\lambda_2 + 9B_2}{\lambda_2^2\sqrt{\gamma}}\mathbb{E}\left[\sqrt{\left(D(g^*) - D(g^{(t)})\right)}\right]\left(C_2\epsilon_C^{(\tilde{n}T)} + C_2C_1^2\epsilon_C^{(\tilde{n}T)}\right)$$

$$\leq \frac{\gamma + 3\lambda_2 + 9B_2}{\lambda_2^2\sqrt{\gamma}}\sqrt{\mathbb{E}\left[D(g^*) - D(g^{(t)})\right]}\left(C_2\epsilon_C^{(\tilde{n}T)} + C_2C_1^2\epsilon_C^{(\tilde{n}T)}\right)$$

$$\leq \frac{\gamma + 3\lambda_2 + 9B_2}{\lambda_2^2\sqrt{\gamma}}\sqrt{\mathbb{E}\left[D(g^*) - D(\tilde{g}^{(t)}) + D(\tilde{g}^{(t)}) - D(g^{(t)})\right]}\left(C_2\epsilon_C^{(\tilde{n}T)} + C_2C_1^2\epsilon_C^{(\tilde{n}T)}\right)$$

$$\leq \frac{2\gamma + 6\lambda_2 + 18B_2}{\lambda_2^2\sqrt{\gamma}}\sqrt{B_1 + B_2 + \frac{C_0}{1 - e^{-\tilde{s}}} + C_0C_2C_3}\exp\left(-\tilde{s}T\right),$$

where we used Lemma 8 and Lemma 15 for the last inequality.

Furthermore, substituting $C_2C_3 = \frac{\sqrt{C_0(2-\mu)\mu}}{\sqrt{2e^{\tilde{s}}(\gamma'+1/\lambda_2)}}$, we have the assertion. $\qquad\square$

Finally, we will prove the Theorem 3, which gives sufficient conditions about sampling accuracy and number of particles for Theorem 1.

**Theorem 3.** *Suppose that Assumptions 1 and 2 hold. When we choose an Option (A), suppose that we set $T_{\text{end}}$ so that*

$$T_{\text{end}} \geq 2\frac{n}{\tilde{n}}\left(1 + \frac{1}{n\gamma\lambda_2}\right)\log\left(\left(n + \frac{1}{\gamma\lambda_2}\right)\frac{(B_1 + B_2 + \frac{C_0}{1-\exp(-\tilde{s})})}{\epsilon_P}\right), \tag{51}$$

*as in Eq. (30) in Lemma 8. Furthermore, when we choose an Option (B), where we choose a solution randomly from $t = \tilde{n}(T_{\text{end}} - \lceil\frac{1}{\kappa}\rceil), \ldots, \tilde{n}T_{\text{end}}$, suppose that we set $T_{\text{end}}$ so that*

$$T_{\text{end}} \geq 2\frac{n}{\tilde{n}}\left(1 + \frac{1}{n\gamma\lambda_2}\right)\log\left(\frac{C_q}{\epsilon_P}\right) + \left\lceil\frac{1}{\kappa}\right\rceil,$$

*where*

$$C_q = \max\left\{\left(2 + \frac{2}{n\gamma\lambda_2}\right)\left(B_1 + B_2 + \frac{C_0}{1 - \exp(-\tilde{s})}\right),\right. \tag{52a}$$

$$4C_0, \tag{52b}$$

$$\left.\left(\frac{(8\gamma + 24\lambda_2 + 72B_2)\sqrt{B_1 + B_2 + C_0 + \frac{C_0}{1-e^{-\tilde{s}}}}\sqrt{C_0 e^{\tilde{s}}(2-\mu)\mu}}{\lambda_2^{\frac{3}{2}}\sqrt{2\gamma(\gamma'\lambda_2 + 1)}}\right)\right\}. \tag{52c}$$

*Then, the condition imposed on $\epsilon_A^{(t)}$ and $\epsilon_B^{(t)}$ in Lemma 8 can be satisfied for both options under the event $\mathcal{E}$ if the following inequalities are satisfied:*

$$\begin{cases} n \geq \frac{2B_2}{\lambda_2}, \\ \epsilon_C^{(nT)} \leq C_3\exp\left(-\frac{\tilde{s}T}{2}\right), \\ \epsilon_D^{(nT)} \leq \frac{C_3}{C_1^2}\exp\left(-\frac{\tilde{s}T}{2}\right) \Leftrightarrow M^{(nT)} \geq \frac{49C_1^4}{C_3^2}\log\left(\frac{4nT_{\text{end}}}{\delta}\right)\exp\left(\tilde{s}T\right), \end{cases} \tag{53}$$

*for $T = 0, 1, \ldots, T_{\text{end}} - 1$. That is, under the conditions above, $\mathbb{E}\left[P(p[\hat{g}]) - D(\hat{g}))|\mathcal{E}\right] \leq \epsilon_P$ is satisfied for the solution $\hat{g}$ of either Option A or B.*

*Proof.* As for the Option (A), according to Lemma 15, Eqs. (51) and (53) are the sufficient conditions for the first statement of Lemma 8 in the event $\mathcal{E}$.

Next, we will show the convergence of Option (B). In the same way, Eqs. (52a) and (53) assure that $\epsilon_A^{(\tilde{n}T)}$ is smaller than $C_0 e^{-\tilde{s}T}$ for every $T = 1, \ldots, T_{\text{end}}$, according to Lemma 15. Furthermore, Eqs. (52b) and (52c) states that both $\epsilon_A^{(t)}$ and $\epsilon_B^{(t)}$ are uniformly bounded by $\frac{\epsilon_P}{4}$ for all $t = \tilde{n}(T_{\text{end}} - \lceil\frac{1}{\kappa}\rceil) + 1, \ldots, nT_{\text{end}}$. Note that the evaluation of $\epsilon_A^{(t)}$ follows from Lemma 15, and that of $\epsilon_B^{(t)}$ is derived in Lemma 17.

Therefore, all assertions are proved. $\qquad\square$

Theorem 3 essentially shows Theorem 1. More detailed explanations can be found in the following.

*Proof of Theorem 1.* Theorem 3 gives the sufficient conditions for Lemma 8. Thus, we have that, for Option (A), with Eqs. (51) and (53), an expected duality gap $\mathbb{E}\left[P(p[g^{(\tilde{n}T_{\text{end}})}]) - D(g^{(\tilde{n}T_{\text{end}})})|\mathcal{E}\right] \leq \epsilon_P$ can be achieved, conditioned by the event $\mathcal{E}$ whose probability is no less than $1 - \delta$. In the same way, for Option (B), with Eqs. (52a)-(52c) and (53), an conditional expectation of duality gap $\mathbb{E}\left[P(p[g^{(t'_{\text{end}})}]) - D(g^{(t'_{\text{end}})})|\mathcal{E}\right] \leq \epsilon_P$ can be achieved. The sufficient condition of $t_{\text{end}}$ is then obtained by noticing $t_{\text{end}} = \tilde{n}T_{\text{end}}$ and setting $\hat{C}_1 = C_3/7$, $\hat{C}_2 = C_1^2$ and $\hat{C}_3 = \frac{2C_q}{1+\frac{1}{n\gamma\lambda_2}}$.

Next, we evaluate the constants as follows. First, we note that $\hat{C}_2 = C_1^2$ by their definitions. Moreover,

$$C_3^{-1} = \frac{\sqrt{2e^{\tilde{s}}(\gamma'+1/\lambda_2)}}{\sqrt{C_0(2-\mu)\mu}}C_2 = O\left(\lambda_2^{-1/2}C_1^2\exp\left(\frac{\tilde{n}(2C_1^2+1)}{n\gamma\lambda_2+1}\right)\right) = O\left(\lambda_2^{-1/2}\hat{C}_2\exp\left(\frac{\tilde{n}(2\hat{C}_2+1)}{n\gamma\lambda_2+1}\right)\right)$$

where we used $C_2 = \frac{2C_1^2}{\gamma+\frac{1}{n\lambda_2}}\exp\left(\frac{\tilde{n}(2C_1^2+1)}{n\gamma\lambda_2+1}\right)$ and $C_1^2 = \hat{C}_2$. Finally, the evaluation of $\hat{C}_3$ can be obtained by bounding $C_q$ using $1/(1-\exp(-\tilde{s})) = O(\tilde{s}^{-1})$ and $1 \leq 1+\frac{1}{n\gamma\lambda_2} \leq 1+\frac{1}{2B_2\gamma} = O(1)$. $\qquad\square$

### C.2.4 DISCRETE GRONWALL'S LEMMA

In the proofs of Lemmas 8, 10 and 13, we used a discrete version of the Gronwall's lemma. We give its formal statement in the following lemma.

**Lemma 18** (Discrete Gronwall's lemma)**.**

*(1) (Vanilla version) Suppose that a sequence of real numbers $(u_n)$ satisfies*

$$u_{n+1} \leq au_n + b_{n+1} \quad (\forall n \geq 0),$$

*for a positive real $a > 0$ and a sequence of real numbers $(b_n)$. Then, it holds that*

$$u_n \leq a^n u_0 + a^n \sum_{k=1}^{n} a^{-k}b_k.$$

*(2) (Summation version) Suppose that a sequence of real numbers $(u_n)$ satisfies*

$$u_n \leq \sum_{k=0}^{n-1} au_k + B \quad (\forall n \geq 1),$$

*for a positive real $a > 0$ and a real number $B$. Then, it holds that*

$$u_n \leq (1+a)^{n-1}(B + au_0).$$

*Proof.* The assertion is just an application of the standard discrete version of the Gronwall's lemma. However, we give a proof for completeness.

The first assertion is a direct consequence of Lemma 5.1 of Mischler (2019). The second assertion is also proven by Lemma 5.2 of Mischler (2019). Indeed, Lemma 5.2 of Mischler (2019) yields that

$$u_n \leq B + \sum_{k=1}^{n-1} a(1+a)^{n-1-k}B + a(1+a)^{n-1}u_0.$$

The right hand side is equivalent to

$$B\left[1 + a\sum_{k=0}^{n-2}(1+a)^k\right] + a(1+a)^{n-1}u_0$$

$$=B\left[1 + a\frac{(1+a)^{n-1}-1}{1+a-1}\right] + a(1+a)^{n-1}u_0$$

$$=(1+a)^{n-1}(B+au_0),$$

which yields the second assertion. $\qquad\square$

## D ADDITIONAL EXPERIMENTS AND DETAILS OF EXPERIMENTAL SETTINGS

Here, we give additional experiments and more detailed explanations of our experimental settings. All experiments were conducted under Google Colaboratory. All codes of the experiments in the main text and the following additional experiments are provided in the supplementary material.

**Additional experiments for a single neuron teacher network** We consider a setting where the teacher network consists of a single neuron:

$$y_i = \sigma(w_1^{*\top} x_i) + \epsilon_i,$$

where the input dimensionality is $d = 10$, $w_1^* \in \mathbb{R}^d$ is generated from the uniform distribution on the unit sphere $\mathbb{S}^{d-1}$ in the $d$-dimensional Euclidean space, $x_i \in \mathbb{R}^d$ is distributed from a standard normal distribution on $\mathbb{R}^d$, $\epsilon_i \in \mathbb{R}$ obeys the Gaussian distribution with mean 0 and standard deviation 0.5 and the activation function $\sigma$ is ReLU. We generated $n = 100$ trainig data points from this model and trained a student network by using P-SDCA, PDA, and a naive SGD. Here, as the number of particles, we employed $M = 200$ and the student network is the tanh neural network ($h_\theta(x) = \tanh(w^\top x + b)$ with $\theta = (w, b)$). We employed the squared loss $\ell(f, y) = (y - u)^2/2$ for the loss function. We conducted experiments with three patterns of regularization parameters: $\lambda_1 = \lambda_2 = \lambda \in \{0.01, 0.001, 0.0001\}$. The naive SGD was trained to minimize the primal objective for the student network with $M = 200$ without the entropy regularization, i.e.,

$$\frac{1}{n} \sum_{i=1}^{n} \ell\left(\frac{1}{M} \sum_{m=1}^{M} h_{\theta_m}(x_i), y_i\right) + \frac{\lambda_1}{M} \sum_{m=1}^{M} \|\theta_m\|^2.$$

We employed the ULA for sampling the particles for PDA and P-SDCA where we ran the algorithm with 200 steps in each re-sampling stage. The step-sizes $\eta$ used for sampling (ULA) and SGD were set as $10^{-3}$.

We plotted the excess primal objective value against the number of gradient evaluations for each method in Figure 2. The excess primal objective value is computed by subtracting the minimum objective value throughout the optimization and all optimization methods from the current primal objective. Since we are approximating the primal variable by a finite particles, we cannot determine the entropy term exactly. To handle this issue, we employed the $k$-NN entropy estimator (Kozachenko & Leonenko, 1987; Brodersen, 2020) to estimate the entropy term from the finite particles. This is also applied to SGD to compute the primal objective with the regularization term for the solution obtained by SGD. From Figure 2, we can see that P-SDCA and PDA behave almost identically for large regularization parameter $\lambda$. In that regime, SGD once decreases the primal objective rapidly but increases it afterward. This is because SGD is minimizing an objective without the entropy regularization. For $\lambda = 10^{-2}$, P-SDCA and PDA does not decrease the excess primal objective below $10^{-2}$. This would be due to the instability of the entropy estimator. For small $\lambda$, we see that P-SDCA outperforms other methods after some iterations because it yields linear convergence. This observation supports our theoretical analyses.

**Additional experiments for a teacher network with multiple neurons** We consider a setting where the teacher network consists of multiple neurons:

$$y_i = \frac{1}{M_t} \sum_{m=1}^{M} \sigma(w_m^{*\top} x_i) + \epsilon_i,$$

where the width of the teacher network is $M_t = 10$, the input dimensionality is $d = 5$, the activation function of the teacher network is $\sigma(\cdot) = \tanh(\cdot)$, the noise is drawn from Normal distribution and

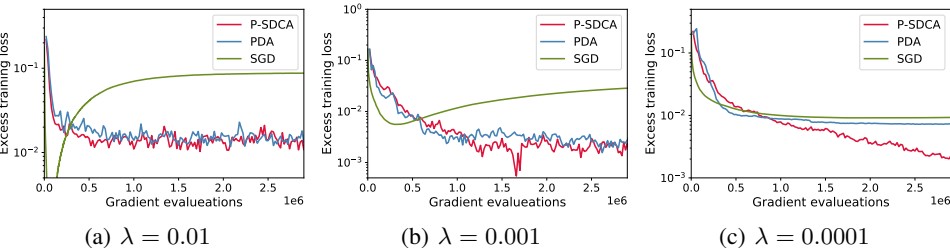

(a) $\lambda = 0.01$      (b) $\lambda = 0.001$      (c) $\lambda = 0.0001$

Figure 2: The number of gradient evaluations versus the excess primal objective for P-SDCA, PDA and SGD for a teacher network with a single ReLU neuron.

its Signal to Noise Ratio (SNR) is 7.0. For each method, we selected the optimal step size: $10^{-4}$ for ULA in P-SDCA and PDA, and $10^{-5}$ for SGD. In the following, "step size of P-SDCA or PDA" refers to the step size for ULA in the sampling scheme. In addition to that, we plotted SGD with its step size $10^{-4}$ to see how the step size affects the convergence of SGD. The purpose to include SGD as well as P-SDCA and PDA is to see how the entropy term will behave when we do not explicitly include the entropic regularization term into the objective function.

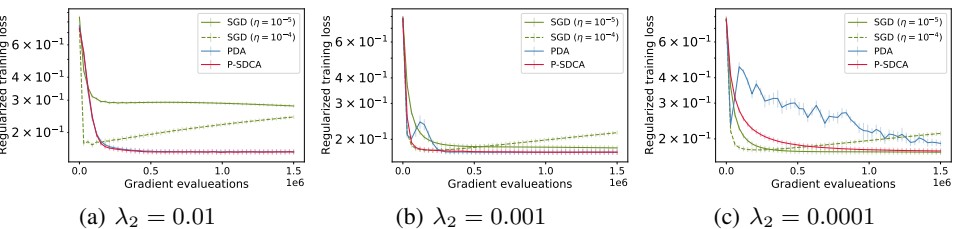

(a) $\lambda_2 = 0.01$        (b) $\lambda_2 = 0.001$        (c) $\lambda_2 = 0.0001$

Figure 3: The number of gradient evaluations versus the excess primal objective for P-SDCA, PDA and SGD for a teacher network with multiple tanh neurons.

We compared our method with SGD and PDA with the regularization parameters $\lambda_1 = 0.01$ (fixed) and $\lambda_2 = 0.01, 0.001, 0.0001$. We repeated the experiments 10 times and took the average of primal objective values, which includes regularization terms. The primal objective value is plotted against the number of gradient evaluations in Figure 3 where the error-bar represents the standard error over the 10 repetitions.

For $\lambda_2 = 0.01$, we can see that particle methods (P-SDCA and PDA) showed faster convergence than that of SGD. Next, as for $\lambda_2 = 0.001$, we can see that P-SDCA converged faster than SGD and PDA. Although PDA and SGD with step size $\eta = 10^{-5}$ also converged eventually, the behavior of PDA in the intermediate stage was quite unstable and SGD with $\eta = 10^{-5}$ showed quite slow convergence. These observations support the superiority of our method. Finally, as for $\lambda_2 = 0.0001$, SGD with step size $\eta = 10^{-5}$ was the fastest due to the small regularization parameter $\lambda_2$ for the particle methods which makes the convergence of them slower. However, we still see that P-SDCA outperformed PDA in terms of both convergence speed and stability. The convergence of SGD is strongly affected by its step size. In fact, SGD with larger step size ($\eta = 10^{-4}$) could decrease the objective rapidly at the early stage, but it eventually diverged. This would be partly for the same reason as in the single neuron setting, that is, due to the lack of the entropic regularization in the objective of SGD. In summary, we can see that P-SDCA properly minimizes the corresponding objective function in wide range of $\lambda_2$ and shows faster and more stable convergence than PDA as indicated by our theory.

**Experiments on MNIST dataset**    We conduct numerical experiments to illustrate the "feature learning" aspect of mean field neural network. For this purpose, we run the algorithm in a binary classification task that separates "2" and "4" in the MNIST dataset. We subsampled $n = 1000$ training examples and trained the two layer tanh network with width $M = 2500$. The step size for ULA in P-SDCA was set as $\eta = 10^{-5}$ and the regularization parameters were set as $\lambda_1 = 10^{-2}$, $\lambda_2 = 10^{-4}$.

To visualize the adaptivity of the feature map, we first plot the evolution of singular values of the first layer's weight matrix, $W = [w_1, w_2, \ldots, w_M] \in \mathbb{R}^{d \times M}$, until convergence, when the solution achieves 99.8% training accuracy. We can see that the singular values of the network trained by P-SDCA drastically changed during optimization. This indicates that the mean-field neural network does not "freeze" and can adaptively learn features during optimization by P-SDCA. We also plotted the same quantity for SGD updates in the NTK regime. In contrast, we observe only small change of the singular values. This is because the first layer's parameters do not change much during optimization in the NTK regime; in other words, SGD in the NTK regime does not perform adaptive feature extraction. In that sense, we expect that training in the mean field regime is effective in a setting where adaptive feature extraction is required.

In addition to the singular value evolution, we evaluate how well the extracted features are "aligned" to the label. For that purpose, we define a kernel function $k_W(x, x') = \sum_{m=1}^{M} \sigma(w_m^\top x) \sigma(w_m^\top x')$ for a weight matrix $W = [w_1, w_2, \ldots, w_M] \in \mathbb{R}^{d \times M}$. Then, we define the *kernel alignment* (Cristianini

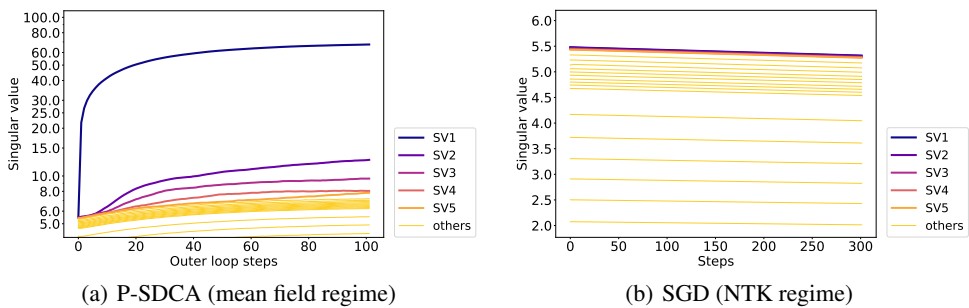

(a) P-SDCA (mean field regime)  (b) SGD (NTK regime)

Figure 4: Evolution of singular values of the first layer's weight matrix during optimization. The figure shows top 5 largest singular values in addition to the $k \times 10$-th largest singular values for $k = 2, 3, 4, \ldots, 9$ and $k \times 100$-th largest singular values for $k = 1, 2, 3, \ldots$.

et al., 2002) on the training data $(x_i, y_i)_{i=1}^n$ as

$$A(k_W) := \frac{\langle K_W, yy^\top \rangle_\mathrm{F}}{\sqrt{\langle K_W, K_W \rangle_\mathrm{F} \langle yy^\top, yy^\top \rangle_\mathrm{F}}}$$

where $\langle A, B \rangle_\mathrm{F} := \sum_{i=1}^n \sum_{j=1}^n A_{ij} B_{ij}$, $K_W = (k_W(x_i, x_j)_{i=1, j=1}^{n,n})$ and $y = (y_1, \ldots, y_n)^\top$. We also define the kernel alignment on the test data in the same manner. We can see that the kernel alignment represents how strongly the kernel function defined by the features after the first layer is aligned to the target signal.

The kernel alignment of the solutions obtained by each method (P-SDCA, SGD in NTK regime, SGD in mean field regime) is depicted in Figure 5. We can see that P-SDCA (and SGD in the mean field regime) properly improves the kernel alignment, which means that P-SDCA can extract informative features adaptively depending on the data. On the other hand, SGD in the NTK regime does not increase the kernel alignment due to the parameters almost "frozen" at initialization. This experiment highlights how well the mean field neural networks can execute feature learning.

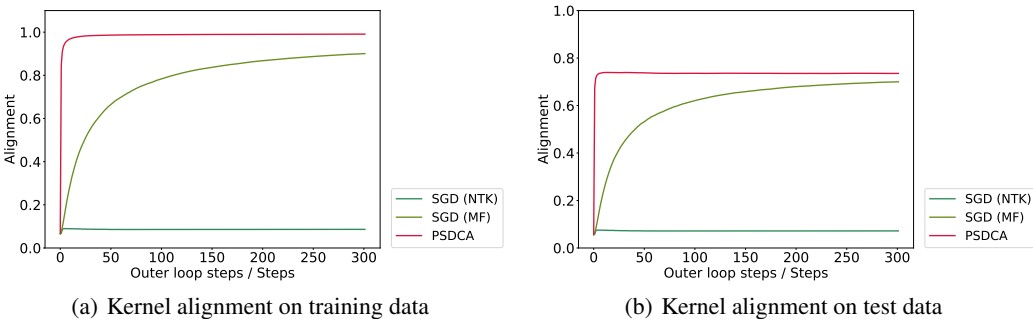

(a) Kernel alignment on training data  (b) Kernel alignment on test data

Figure 5: Evolution of kernel alignment of the extracted features during optimization.

