# OpenReview forum: "Particle Stochastic Dual Coordinate Ascent: Exponential convergent algorithm for mean field neural network optimization"
_ICLR.cc/2022/Conference — ICLR 2022 Poster_

### Official Review · Reviewer_ny66 · 2021-11-01

**Correctness:** 3
**Technical Novelty And Significance:** 4
**Empirical Novelty And Significance:** Not applicable
**Recommendation:** 8
**Confidence:** 3

**Main Review:**

### Strengths
The paper is well-written, technically sound and I have no doubts about the correctness of the results (though I did not check the proofs in the appendix carefully).

The presented dual particle optimization scheme appears to be novel and might be useful for other problems on probability density than mean-field neural networks.

### Weaknesses
A) Motivation
* Since the main contribution is an algorithm (and its analysis) for Eq. (1), it would be great to motivate this problem a bit more for the uninitiated reader. For example, is it expected to be preferable over standard neural network training (eg. due to the regularization or convexity) or is the goal rather to gain a better understanding on standard training through it? In the experiments, the entropic regularization seems to help generalization, so perhaps this could be mentioned as a motivation (with additional explanations).

B) Clarity
* It would be helpful if the paper could provide an explicit definition of the set P of probability density functions on R^{\tilde d} and a proof of Proposition 1. I am confused about this point, since in order for Theorem 15.23 of Bauschke et al. (2011) to be applicable in Proposition 1, it seems we have to be in a Hilbert space setting. But I am unsure whether the set of probability density functions considered here is a subset of a Hilbert space.  As a remark, in case the Hilbert space setting does not suffice, note that the original work by Rockafellar "Duality and stability in extremum problems involving convex functions", Pacific J. Math, 1967 already proves the duality theorem in the more general setting of paired topological vector spaces.

* It was a bit tedious to follow the proof of Lemma 1 (Lemma 3 in the appendix on page 19). I can see that the elements of the Hessian are bounded by one, but it would be nice if the paper could add a clarifying sentence how this yields Eq. (13).  Also, the sentence "Note that the bound (12) of the Hessian in the form of the covariance matrix regarding h_i" did not make sense to me, and Eq. (12) appears not to be a bound but rather an equality.

C) Experimental evaluation
* Would it be possible to run the method on a small binary classification example, e.g. on MNIST (2 vs 4)? The current experiment in Figure 1 suggests that the particle method generalizes much better than SGD (mean field / NTK), and it would be interesting to confirm this on a small real world dataset rather than in the teacher/student setup. Perhaps one could even show the learned 28x28 image-filters to illustrate the "feature learning" aspect of the mean-field algorithm (opposed to the NTK one).


**Summary Of The Paper:**

The problem of training an infinitely wide neural network with one hidden layer can be written as an optimization problem in the space of probability distributions, where each atom in the distribution corresponds to a hidden neuron. The paper proposes to solve instead the dual problem (of an entropy regularized variant), which in the case of finite data is a finite-dimensional optimization problem.  The main contribution is the conception and analysis of an efficient dual coordinate ascent scheme to solve the dual, which in theory and practice requires fewer iterations to find an epsilon-accurate solution than existing methods.


**Summary Of The Review:**

The paper proposes and studies an interesting optimization method for problems in the space of probability density functions. The main application considered are mean-field neural networks, but  I can see how this algorithm will be very useful also for other problems over probability densities which are ubiquitous in ML.

There are several smaller issues mostly regarding the technical clarity and motivation in the paper (see main review). If these are clarified/addressed in a revision, I am also willing to increase my score.

---

> ### Author Response · Authors · 2021-11-22
> **Reply to reviewer ny66**
>
> Thank you for the thoughtful review which helped us improve the manuscript significantly. We address the technical comments below.
>
> **1. Motivation of the algorithm**
> Our goal is to propose a more efficient algorithm for optimizing mean field neural networks with faster convergence guarantee, rather than to gain a better understanding on standard neural network training. More generally, our motivation is to extend classical methods in convex (stochastic) optimization into the space of measures.
> We have added a short comment on this point in the introduction.
>
>
> **2. Clarity**
> - **It would be helpful if the paper could provide an explicit definition of the set P of probability density functions.**
> Thank you for the valuable comment -- indeed Rockafellar (1967) best fits our setting. In the revision, we have added a complete proof of Proposition 1 in Appendix A and modified the statement accordingly.
> Specifically, to apply the result of Rockafellar (1967), we notice that the set of density functions $\mathcal{P}$ is a (closed) convex subset of $L^1(\mathbb{R}^d)$ and it is known that $L^\infty(\mathbb{R}^d)$, the set of essentially bounded functions, is the topological dual of $L^1(\mathbb{R}^d)$.
> Therefore, $L^1(\mathbb{R}^d)$ with $L_1$-norm and $L^\infty(\mathbb{R}^d)$ with the weak$^*$ topology are topological dual to each other.
> Then, Theorems 3 and 8 of Rockafellar (1967) can be applied to establish the proposition.
> -  **The sentence "Note that the bound (12) of the Hessian in the form of the covariance matrix regarding $h_i$" did not make sense to me.**
> Thank you for the close reading. The statement contains a typo and is indeed unclear. We have modified this sentence to: "Note that the right hand side of the equation (12) for the Hessian evaluation is given by the covariance between $h_i$ and $h_j$ with respect to the probability distribution $p\[g\](\theta)$".
>
>
> **3. Experimental evaluation**
> Following your suggestion, we have conducted an additional experiment on the MNIST dataset in Appendix D to illustrate the feature learning aspect of our mean field algorithm. Figure 4 shows the evolution of singular values of the first-layer weight matrix during the optimization.
> We can see that the ratio of singular values drastically changes in the first few steps, which is a sign of feature (representation) learning of mean-field neural networks.
> In contrast, for model initialized according to the NTK scaling and optimized by SGD, we observe that the change is relatively small. This is likely because the model is close to kernel regime, so that it achieves good training error with small change in the features.
> We also plotted the "kernel alignment" between the label and the features trained by each method in Figure 5.
> This also shows that P-SDCA optimization in the mean field regime can properly learn features aligned to the target label signal.
> In contrast, SGD in the NTK regime does not improve the kernel alignment. These results demonstrate the presence of feature learning in our proposed method.
> Finally, to illustrate the potential benefit of entropy regularization, we included visualization of a 1D input regression problem (see Figure 6). We observe that due to the negative entropy regularization, the solution obtained by P-SDCA is smoother than the SGD solution (which does not involve the entropy regularization).
>
> We would be happy to clarify any further concerns/questions in the discussion period.

---

> > ### Comment · Reviewer_ny66 · 2021-11-23
> > **Thank you for the clarifications!**
> >
> > Thank you for adding a rigorous proof of Proposition 1; it appears correct to me. The experiments on MNIST and entropy regularization are also nice to see.  My concerns are addressed, and I will raise my score accordingly.
> >
> > Typos:
> > * In the proof of proposition 1, (L^1(\R^d) = L^2(R^d, \mu)) should be L^1(\R^d) = L^1(\R^d, \mu).
> > * "This indicates that the negative entropy regularization can as a smoothness penalty, which may be useful to avoid overfitting in a noisy dataset." -> "This indicates that the negative entropy regularization case serve as a smoothness penalty".

---

> > > ### Author Response · Authors · 2021-11-23
> > > **Thank you very much for checking the revision.**
> > >
> > > Thank you very much for quickly checking the revision.
> > > We have fixed the typos that you kindly pointed out, and uploaded the revised paper.
> > > We greatly appreciate your valuable feedbacks.

---

### Official Review · Reviewer_pSnQ · 2021-11-02

**Correctness:** 4
**Technical Novelty And Significance:** 3
**Empirical Novelty And Significance:** Not applicable
**Recommendation:** 6
**Confidence:** 4

**Main Review:**

Thanks for the clarification. I'd like to keep my score.

======================

The paper is clear in its exposition, despite involving many parameters. In the context of MF two-layer neural nets, the idea is new. The proofs seem correct. I also appreciate that the authors have made some honest remark concerning $\lambda_2$, which is clearly a deficiency in this approach.

I do not have much to complain, though a few things could be considered:
- An explanation of the role of the re-weighting stage is needed. It is said to reduce the number of resampling times $\tilde{n}$ down from $n$ and hence brings down the complexity. It would be good to be more quantitative about this. Would the result be $O((n/\lambda_2\gamma)\log(1/\epsilon_P)$ as mentioned in page 7?
- It is said that Mei et al 2018 has the error growing exponentially with time. I think here the same also applies. One can take a different view that is to fix a target error to argue against this exponential dependency, but the same view applies to Mei et al 2018 as well.

**Summary Of The Paper:**

The paper considers optimization of MF two-layer neural nets (the infinite-width version with entropic regularization). Specifically the paper proposes to do dual coordinate ascent, which allows for exponential convergence rate, together with a particle approximation scheme. The paper shows that in the discrete-time setting, an improved convergence rate is obtained (w.r.t. desired error $\epsilon_P$, the number of data points $n$, the regularization $\lambda_2$).

**Summary Of The Review:**

The paper proposes an interesting variant of dual coordinate ascent that achieves improvements in the convergence rate. There is not much to criticize since the paper has optimized the complexity to a good extent, except for the inherent difficulty with the approach that depends crucially on $\lambda_2$.

---

> ### Author Response · Authors · 2021-11-22
> **Reply to reviewer pSnQ**
>
> Thank you for the helpful feedback and positive evaluation. We address the technical comments below.
>
> **1. An explanation of the role of the re-weighting stage is needed. It would be good to be more quantitative about this.**
>
> We would like to direct the reviewer to Theorem 1 and the following remark and discussions, where we elaborated how the selection of the resampling interval affects the runtime complexity. In particular, the number of outer loops does not depend on $\tilde{n}$; but by employing the re-weighting scheme, we do not need to sample particles in every outer iteration. This is beneficial because the sampling step is computationally demanding and thus we want to skip it as much as possible.
> However, there is no free lunch in the sense that large $\tilde{n}$ requires higher accuracy of sampling and thus the computational complexity of the inner loop becomes large. The quantitative evaluation of how $\tilde{n}$ impacts the sampling complexity is given in Theorem 1 and Proposition 2 through $\hat{C}_1$ and $\hat{C}_2$ (see also Remark 1). We can see that $\tilde{n} = n\lambda_2$ would be a reasonable choice, since it avoids exponential computational cost with respect to $n$ and $\lambda_2$.
>
>
> **2. It is said that Mei et al 2018 has the error growing exponentially with time. I think here the same also applies.**
>
> Thank you for bringing up an important point; we make the following clarification.
> As mentioned in the main text, Mei et al. (2018) evaluated the discrepancy between a discrete time scheme and its continuous time counterpart, and provided a bound of the discrepancy as $O(e^{C k\eta}(1/\sqrt{M} + \eta))$, where $\eta$ is the step size, $k$ is the number of steps and $M$ is the number of particles. Since $T = k\eta$ should diverge to $\infty$ for exact global convergence, we have an exponential dependency on $T$.
> On the other hand, our P-SDCA *does not* have a continuous time counterpart, so we do not have an exponentially growing coupling error produced by time discretization. One may note that if ULA is employed as the MCMC method, then the time discretization error also appears; but importantly, the sampling error will be "forgotten" in an exponential order (this can be seen from the proof of Lemma 7), that is, the effect of such error in an intermediate iteration exponentially decreases to the final accuracy. This is in stark contrast to the exponential growth in the discretization error.
> Finally, we remark that Mei et al. (2018) also considers the noisy gradient descent algorithm,  which includes the negative entropy term in our problem formulation. Hence we believe that (part of) the analysis in Mei et al. (2018) would also suffer from the same issue of exponential dependency on $\lambda_2$ (which is a problem different than the time discretization error).
>
> We would be happy to clarify any further concerns/questions in the discussion period.

---

### Official Review · Reviewer_CmPk · 2021-11-03

**Correctness:** 4
**Technical Novelty And Significance:** 3
**Empirical Novelty And Significance:** 3
**Recommendation:** 6
**Confidence:** 4

**Main Review:**

This paper concerns optimizing mean-field neural networks. The problem is inherently challenging because the infinite width limit leads to a functional formulation of the representation and, consequently, the optimization must be carried out over the space of probability measures by solving a PDE. Particle-based approaches alleviate some of the challenges, but, as the authors note, the original mean-field papers deal with discretization simply by taking the limit $\Delta t\to 0$. Here, the authors propose to instead work directly with a space-time discretized dynamics.

The authors examine a particular variant of the mean-field objective function in which an entropic regularization term is added.  However, unlike these previous works, the authors take advantage of Fenchel duality to convert the problem into a maximization problem in $\mathbb{R}^n$ as opposed to an infimum over $p\in \mathcal{P}$.

The proposed algorithm relies on first sampling parameters from a proxy probability measure $\rho^{(t)}(\theta)d\theta$ that is close in total variation distance from the target measure $\rho[g^{(t)}]$. How is it ensured that the proxy is actually close? This strikes me as a challenging thing to verify.

Because the algorithm requires sampling a high-dimensional distribution, this step is performed infrequently and the particles are reweighted iteratively. The sampling with ULA or MALA is the basis of the convergence results, using a LS inequality. Of course, the exponential decay rate will be unfavorable if the target distribution is metastable. There's no real discussion of the nature of the parameter distribution. Is there some reason we should expect that it's easy to sample?

**Summary Of The Paper:**

This paper presents an optimization alogrithm for mean-field shallow neural networks. The algorithm optimizes the parameter measure using a particle-based implementation of the stochastic dual coordinate ascent algorithm. The authors establish convergence results, namely an exponential convergence rate owing to the convexity of the dual problem.

**Summary Of The Review:**

The paper outlines an algorithm and carefully analyzes it. The algorithm appears to be effective in a simple synthetic example. However, the assumption of efficient sampling of the parameter distribution requires substantial additional justification.

---

> ### Author Response · Authors · 2021-11-22
> **Reply to reviewer CmPk**
>
> Thank you for the thoughtful comments and insightful questions. We address the technical points below.
>
> **1. How is it ensured that the proxy is actually close? This strikes me as a challenging thing to verify.**
>
> This is a good point. While our theoretical analysis ensures the rapid convergence of the sampling algorithm (in TV distance) due to the verified LSI, as you noticed, it is a computationally challenging to empirically compute the TV distance from the particles.
> This being said, for practical usage, we don't need to exactly verify that the sampling distribution is actually close to the target distribution. This is because the algorithm works "without" such an exact guarantee and the theoretical discrepancy merely appears in the final algorithm accuracy. Thus, to improve the overall accuracy of the algorithm, we just need to increase the number of particles and number of iterations in the inner loop, which can be changed during optimization.
>
>
> **2. Is there some reason we should expect that the target distribution is easy to sample?**
>
> The sampling efficiency is due to our regularized empirical risk minimization setting (and that each neuron $h_\theta$ is bounded and smooth), which leads to a dimension independent bound of the log-Sobolev constant -- please refer to Proposition 2 for guarantee for sampling from the target distribution.
> One drawback of this evaluation is that there is an exponential dependency on $\lambda_2$, which renders our convergence analysis ineffective under vanishing regularization; as commented in Remark 2, we believe that this is unavoidable in the most general setting.
> As future work, we intend to investigate conditions under which such dependence can be avoided. For instance, it is natural to speculate that when the initial distribution and the true (target) distribution both have good isoperimetry, then the sampling complexity of our method in the intermediate iterations might also be independent of $\lambda_2$.
>
> We would be happy to clarify any further concerns/questions in the discussion period.

---

### Official Review · Reviewer_M8tQ · 2021-11-03

**Correctness:** 3
**Technical Novelty And Significance:** 3
**Empirical Novelty And Significance:** Not applicable
**Recommendation:** 6
**Confidence:** 2

**Main Review:**

Comments:
This paper introduces techniques to directly solve the infinite-dimensional mean-field model for two-layer NN. It allows finite step size and improves the convergence rate to linear by solving the dual problem and assuming primal smoothness. My main comments are as follows:

* Proposition 1 needs more careful treatment. The authors claim the Fenchel–Rockafellar strong duality holds without formal justification. Note that, the strong duality holds when certain domain inclusion condition [Eq. 15.33, Bauschke 2011] is verified, which is not trivial for Eq. 2. One should not think the domain of l_i^* would be the whole R as the loss could be a logistic loss, which is not coercive (see [Theorem 11.8(d), R1].

* Intuitively, the update 4+5 in Algorithm 1 is very similar to running one-step SGD for the i-th data point. After computing step 4, the "weights" are updated by step 5. Is there any connection with a primal algorithm given this similarity?

* On stepsize for P-SDCA: it seems the step size length in the Algorithm is n*lambda_2, which is from the lower estimation in Lemma 1. However, on Page 40, additional experimental results, they said "we selected the optimal step size ... for P-SDCA and PDA". If n=100 as on Page 39, then the step sizes for Fig 3(a-c) would be actually different. This should be made clear.

---
[R1] Rockafellar, R.T. and Wets, R.J.B., 2009. Variational analysis (Vol. 317). Springer Science & Business Media.

**Summary Of The Paper:**

This work solves the entropy regularized mean-field model for a two-layer neural network. As the Fenchel–Rockafellar dual of that problem is a finite dimension problem, they consider an SDCA-type scheme to solve the dual problem, which still contains an integral term and they use the Langevin iterate procedure to solve the subproblem approximately. In contrast to the existing analysis for SGD in the mean-field regime, the new algorithm has a fixed length dual step size and has a fast linear convergence rate.

**Summary Of The Review:**

This work introduces a dual algorithm to solve the infinite-dimensional mean-field training problem. The update step is solved approximation by estimating an integral term in every iteration. The new technique is interesting and the new linear convergence rate improves the prior result.

---

> ### Author Response · Authors · 2021-11-22
> **Reply to reviewer M8tQ**
>
> Thank you for the valuable feedback which helped us improve the manuscript. We address the technical comments below.
>
> **1. Proposition 1 needs more careful treatment.**
>
> Thank you for pointing out an important issue. Due to the space limitations, we omitted the technical details for this proposition, but we agree with your suggestion. Accordingly, we have modified the statement of the proposition to make it rigorous, and also added a complete proof in Appendix A.
> Indeed, Theorem 15.23 of Bauschke (2011) holds for more general vector spaces with a topological dual. Therefore, we do not restrict ourselves to a Hilbert space. This is rigorously justified by [R1]. In our setting, the set of density functions $\mathcal{P}$ is a (closed) convex subset of $L^1(\mathbb{R}^d)$ and it is known that $L^\infty(\mathbb{R}^d)$, the set of essentially bounded functions, is the topological dual of $L^1(\mathbb{R}^d)$. Therefore, $L^1(\mathbb{R}^d)$ with $L_1$-norm and $L^\infty(\mathbb{R}^d)$ with the weak$^*$ topology are topological dual to each other.
> Therefore, Theorems 3 and 8 of [R1] can be applied to establish the proposition. As for the coerciveness, the duality theorem does not require that the loss function is coercive. Instead, it requires the loss functions is a proper lower semi-continuous convex function.
>
> [R1] Rockafellar: "Duality and stability in extremum problems involving convex functions". Pacific Journal of Mathematics, Vol. 21, No. 1, 167--187, 1967.
>
> **2. Connection to a primal algorithm**
>
> Indeed, our method (P-SDCA) randomly selects one data point and updates the solution using the selected data point, which resembles the stochastic gradient descent. Although SDCA updates the dual variable instead of the primal, the method can be seen as a different implementation of stochastic gradient descent [R2]. However, the convergence rate for SDCA is exponential with respect to the number of iterations while SGD can achieve only a polynomial order convergence ($1/\sqrt{T}$ or $1/T$). This difference comes from the fact that SDCA has a "variance reduction" effect. Due to this variance reduction, we don't need to decrease the step-size to 0, in contrast to SGD. See [R2] for more details.
>
> [R2] Johnson and Zhang: "Accelerating Stochastic Gradient Descent using Predictive Variance Reduction". Advances in Neural Information Processing Systems 26 (NIPS 2013), 2013.
>
>
> **3. Choice of optimal step-size**
>
> We would like to clarify a potential misunderstanding on the choice of "step-size". In particular, for P-SDCA we do not need to determine a step-size for $g^{(t)}$. Instead, the "optimal step size" we mentioned refers to the step-size of the unadjusted Langevin algorithm (ULA) for sampling the particles in the inner loop -- this corresponds to line 3 of Algorithm 2 (see also the description of ULA in page 6). We have added one sentence in the revision to clarify the meaning of the "optimal step size" in page 40.
>
> We would be happy to clarify any further concerns/questions in the discussion period.

---

### Author Response · Authors · 2021-11-22
**General comments on the revision**

We appreciate the reviewers' valuable feedback, which were helpful for us to improve the manuscript. We reply to the reviewers’ feedback as separate comments below, and we have revised the paper accordingly. The main points of the revision are listed as follows. The revised parts are indicated by red color in the new version of the paper.

1. In Appendix A we provided a rigorous proof of Proposition 1 based on Rockafellar (1967) which gave the Fenchel's duality theorem on general vector spaces.
1. In Appendix D we added numerical experiments on the MNIST dataset to demonstrate the "feature learning" aspect of mean field neural network optimized by our P-SDCA method. In particular, we observed that the network optimized by P-SDCA largely changes its internal layer's features, and significantly improves the *kernel alignment* to the target label.
1. We fixed typos and improved the clarity of the writing.

If there are further questions/comments/suggestions, we would be happy to address them in the discussion period.

Best regards,
Authors.

---

### Decision · Program_Chairs · 2022-01-20

**Decision:**

Accept (Poster)

**Comment:**

This is a solid paper and considers the problem of training a wide neural network with a single hidden layer. This can be framed as an optimization problem in the space of probability distributions with a suitable entropy regularization, where each atom in the distribution corresponds to a hidden neuron. The dual of this problem (for finite data) is a finite-dimensional optimization problem and the paper proposes a particle based coordinate ascent scheme.
The paper provides some convergence rate results. After the rebuttal, the authors have also included more experimental/numerical results.

The authors have answered the concerns raised by the reviewers and overall, the paper can be accepted:
The presented approach appears to be sufficiently novel and might be useful in other settings.
The presentation is clear and easy to follow for such a technical paper; the paper is well organized.
The limitations of the approach are clearly stated (dependence on the regularization parameter for entropy term that may be hard to select)